# PANDA: A PRETRAINED FORECAST MODEL FOR CHAOTIC DYNAMICS

**Jeffrey Lai**[*]
The Oden Institute
UT Austin
jlai@utexas.edu

**Anthony Bao**[*]
ECE Department
UT Austin
abao@utexas.edu

**William Gilpin**[†]
Department of Physics
UT Austin
gilpin@chaos.utexas.edu

## ABSTRACT

Chaotic systems are intrinsically sensitive to small errors, challenging efforts to construct predictive data-driven models of real-world dynamical systems such as fluid flows or neuronal activity. Prior efforts comprise either specialized models trained on individual time series, or foundation models trained on vast time series databases with little underlying dynamical structure. Motivated by dynamical systems theory, we present *Panda*, *P*atched *A*ttention for *N*onlinear *Dyn*A*mics. We train *Panda* on a novel synthetic, extensible dataset of $2 \times 10^4$ chaotic dynamical systems that we discover using an evolutionary algorithm. Trained purely on simulated data, *Panda* exhibits emergent properties: zero-shot forecasting of unseen chaotic systems preserving both short-term accuracy and distributional measures, nonlinear resonance patterns in attention heads, and effective prediction of real-world experimental time series. Despite having been trained only on low-dimensional ordinary differential equations, *Panda* spontaneously develops the ability to predict partial differential equations without retraining. We also demonstrate a neural scaling law for differential equations, underscoring the potential of pretrained models for probing abstract mathematical domains like nonlinear dynamics.

## 1 INTRODUCTION

Nonlinear dynamical systems test the limits of scientific machine learning (SciML). When an approximate model is constructed of a chaotic nonlinear system, any small error grows exponentially over time, precluding long-term forecasting. This intrinsic property underscores the practical difficulty of accurately forecasting systems like weather fronts, neural activity, or economic markets (Li et al., 2022; Mikhaeil et al., 2022; Price et al., 2025).

Recent empirical studies show surprising progress on the classical problem of forecasting chaos, including the ability to predict these systems well-beyond the classical predictability timescale for nonlinear systems (Gilpin, 2021; 2023; Pathak et al., 2018). These approaches construct *local* forecast models trained on past observations of a single dynamical system, and then forecast future, unseen states of the same system. For dynamical systems, this represents an in-domain generalization task, because future timepoints are drawn from the same underlying differential equations. This problem thus reduces to learning the numerical propagator for the true underlying governing equations.

However, a frontier in SciML is out-of-domain generalization (Göring et al., 2024; Wang et al., 2022):

*Can a dynamics model effectively forecast unseen dynamical systems?*

This task requires a *global* forecast model, which combines training on a large body of background knowledge with local adaptation to generate meaningful forecasts of unseen systems (Sen et al., 2019). Moreover, what kind of data is required to train a forecasting model for dynamical systems in order to achieve generalization? A global nonlinear forecast model has intrinsic theoretical interest in SciML, which has long questioned the degree to which complexity can be "transformed out" i.e. whether the predictability of a system is determined by its intrinsic properties or by the choice of measurement coordinates (Brunton et al., 2022; Mezić, 2013).

---

[*]Equal contribution. [†]Corresponding author.

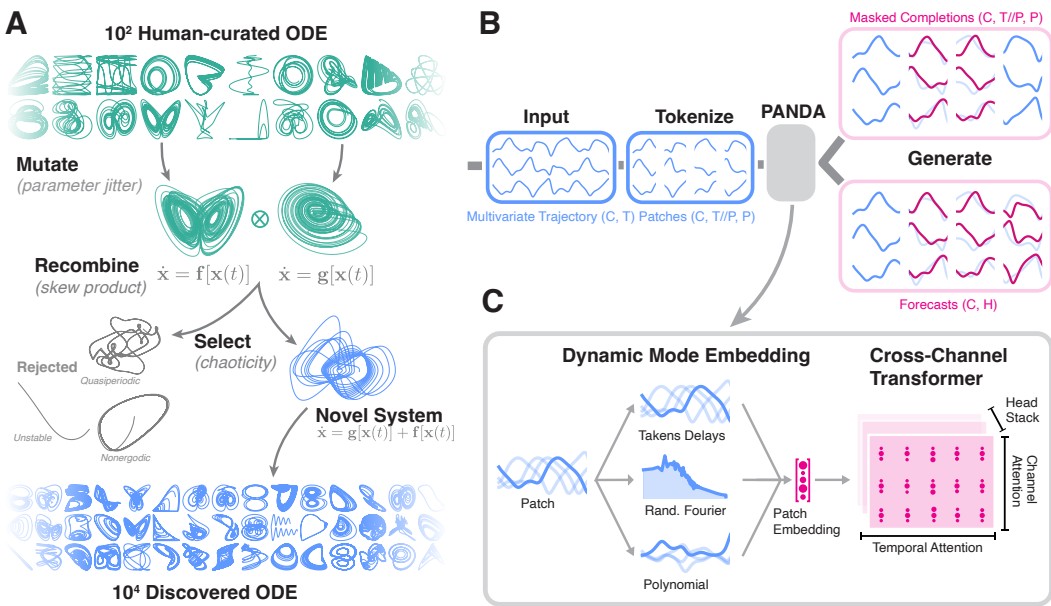

Figure 1: **A large-scale chaotic dynamics dataset and dynamics-informed forecast model.** (A) Evolutionary creation of a large dataset of chaotic ODEs through mutation and recombination of known systems. (B) Patch model architecture with forecasting and masked completion output modes. (C) The dynamics-informed time series embedding and attention modules.

To address these questions, we introduce *Panda*[1] — *P*atched *A*ttention for *N*onlinear *Dyn*amics. Our key contributions are as follows:

1. We introduce a framework for generating novel chaotic dynamical systems, allowing us to create a dataset of $\sim 2 \times 10^4$ ODEs, algorithmically-discovered based on evolutionary recombination of 129 chaotic systems such as the Lorenz attractor, double pendulum, etc.

2. We pretrain a global forecast model for nonlinear dynamics purely on chaotic trajectories integrated from our dataset. Our model exhibits competitive zero-shot forecasts for real systems including mechanical motion of C. Elegans, electronic circuits, and turbulent flows.

3. We demonstrate the effectiveness of features motivated by dynamical systems theory: (a) masked pretraining for dynamical continuity (b) channel attention for dynamical coupling, (c) kernelized patch embeddings based on dynamic mode decomposition.

4. Despite being trained only on low-dimensional ODEs, *Panda* develops emergent ability to zero-shot forecast high-dimensional PDEs.

## 2 RELATED WORK

**Machine learning for dynamical systems.** Machine learning models for dynamical systems (MLDS) leverage as inductive biases the unique properties of dynamical systems, relative to traditional time series. These include: (1) *Strong channel coupling:* The evolution of system variables is governed by deterministic differential or difference equations, implying coupled functional dependencies among variables rather than statistically correlations. Several MLDS approaches perform large-scale multivariate dynamical modeling, or infer interactions networks among measurement channels (Bhaskar et al., 2024; Brunton et al., 2022; Chen et al., 2018; Li et al., 2022). (2) *Invariant statistical measures:* Ergodic dynamical systems possess invariant probability measures supported on non-wandering sets, such as limit cycles or strange attractors, resulting in well-defined long-term statistical distributions for all observables. Recent works incorporate these properties as inductive biases in modern methods in MLDS settings (Cheng et al., 2025; Koppe et al., 2019; Pedersen et al., 2025).

---

[1]Code available: https://github.com/abao1999/panda

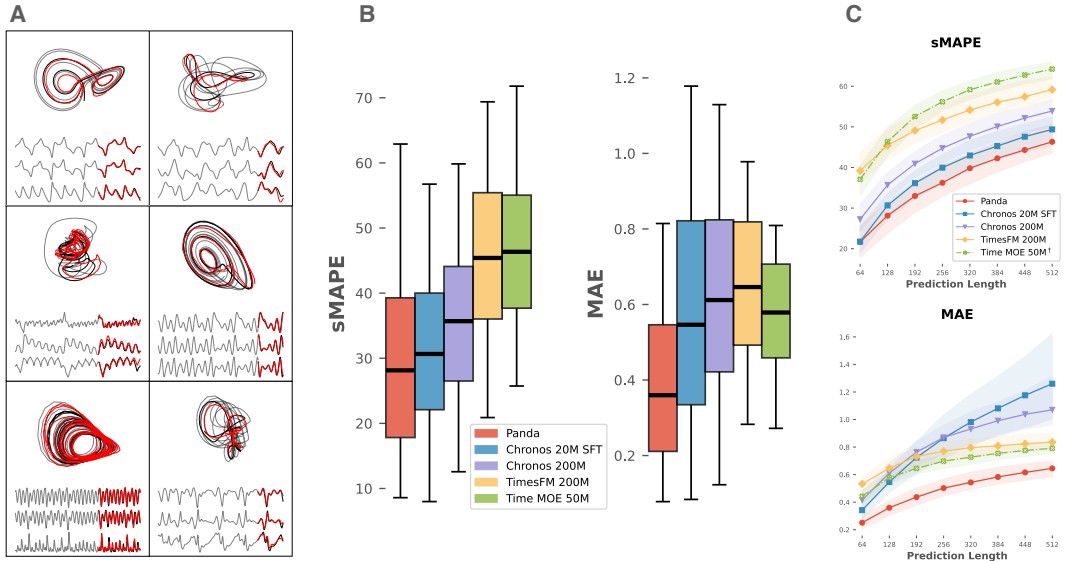

Figure 2: *Panda* **zero-shot forecasts unseen nonlinear dynamics.** (A) Example zero-shot forecasts on novel chaotic skew-systems. (B) sMAPE and MAE of *Panda* compared to zero-shot time series models over a 128 timepoint prediction horizon. (C) Error versus forecast horizon. Error ranges correspond to median and semi-interquartile range across $9.3 \times 10^3$ held-out dynamical systems, 6 forecasts per system. Note: † indicates some NaNs present in forecasts (more examples in Appendix C; dataset description in Section 3). See Table 6 in Appendix D for statistical significance tests.

**Discovering new dynamical systems.** Small datasets of dynamical systems have previously been curated from the published literature (Gilpin, 2021; 2023; La Cava et al., 2021). Several pretrained models, particularly for partial differential equations (PDE), generate new equations for training by randomly-perturbing parameters or initial conditions from known systems (Chen et al., 2024; Herde et al., 2024; Sun et al., 2025; Tripura & Chakraborty, 2023). Others construct *de novo* systems by combining terms from a fixed function library (Ziegler et al., 2024), or leveraging language models to create candidate symbolic expressions (d'Ascoli et al., 2023; Du et al., 2024). However, these approaches do not address the harder task of sampling based on whether a system exhibits a unique dynamical attractor. Richer sampling requires *post–hoc* analysis of candidate dynamics, akin to intrinsically-motivated discovery previously used in domains such as cellular automata and coupled oscillators (Crutchfield & Mitchell, 1995; Falk et al., 2024; Kumar et al., 2024; Reinke et al., 2020). Some foundation models generate synthetic time series using stochastic dynamics like Gaussian processes (Ansari et al., 2024; Das et al., 2024), or simulated physics environments (Lin et al., 2025; Wang et al., 2024).

**Pretrained models for SciML.** Pretrained foundation models for dynamics enable transfer learning and zero-shot inference. One study trains transformers across diverse PDEs to create a shared multiphysics embedding space (McCabe et al., 2024). Another study proposes supervised pretraining to enable out-of-domain generalization for scientific foundation models, and derives scaling laws for transfer learning on PDEs (Subramanian et al., 2023). Several recent studies evaluate the zero-shot performance of time series and language models in MLDS, and observe performance only comparable to standard time series tasks (Liu et al., 2024; Zhang & Gilpin, 2025b). Several studies apply pretrained transformers to control or symbolic equation discovery tasks (Becker et al., 2023; d'Ascoli et al., 2023; Lee et al., 2023; Zhang et al., 2024). One work generates pretraining data by randomizing the parameters of four named ODE (Song et al., 2024), similar to the first step of our evolutionary algorithm described below, with a small founder pool. Another work samples systems from a fixed function space, selecting based on total variation over time (Ziegler et al., 2024), while another study uses latent ODE as a prior for zero-shot imputation (Seifner et al., 2025). A contemporaneous work to our study, *DynaMix*, is a multivariate mixture-of-experts model for zero-shot dynamical systems reconstruction (Hemmer & Durstewitz, 2025), built from Almost-Linear RNN experts and trained on the founder pool for our dataset (Brenner et al., 2024). Our work is distinguished by (1) a rich data generation process, which discovers novel chaotic flows with diverse properties, and (2) a multivariate patched-based architecture which demonstrates emergent forecasting capabilities like zero-shot PDE inference.

## 3  DATASET

**Evolutionary search.** We discover $2 \times 10^4$ novel chaotic ODEs (schematic in Fig. 1A, example systems in Appendix A). **1. Founding population:** We start from a human-curated dataset of 129 previously-published low-dimensional chaotic systems (Gilpin, 2021), consisting of curated ODEs from the literature (e.g. the Lorenz equations or blinking vortex flow) of the form $\dot{\mathbf{x}} = \mathbf{f}_\theta(\mathbf{x}, t)$. The default parameters of each system $\theta$ and initial conditions $\mathbf{x}(0)$ were hand-tuned to the chaotic regime, and the integration timescales were standardized based on calculations of invariant mathematical properties of the underlying equations, such as the largest Lyapunov exponent. **2. Mutation:** We randomly sample pairs of systems $\mathbf{f}_a$, $\mathbf{f}_b$. For each ODE's default system parameters, we add random Gaussian noise, $\theta'_a \sim \mathcal{N}(\theta_a, \sigma)$, $\theta'_b \sim \mathcal{N}(\theta_b, \sigma)$. **3. Recombination:** We combine the mutated parents using an additive skew-product coupling:

$$\dot{\mathbf{x}} = \mathbf{f}_a(\mathbf{x}, t) \tag{1}$$
$$\dot{\mathbf{y}} = \kappa_b \mathbf{f}_b(\mathbf{y}, t) + \kappa_a \, \mathbf{f}_a(\mathbf{x}, t) \tag{2}$$

This coupling between flows is asymmetric, and thus we refer to $\mathbf{f}_a$ as the driver and $\mathbf{f}_b$ as the response. In general, skew-product coupling maps can be symmetric and nonlinear, but may be harder to integrate as a result. This particular recombination scheme, for appropriate scale factors, preserves chaoticity because the response system either synchronizes to the chaotic driver or continues to exhibit chaotic dynamics (Gilpin, 2025; Pecora & Carroll, 1990). For the scale factors, we compute the inverse RMS norm $\kappa = 1/\sqrt{\mathbb{E}||f(\mathbf{x}, t)||^2}$ for each individual flow over a representative trajectory. **4. Selection:** We integrate trajectories from multiple initial conditions using a 5th order implicit Runge-Kutta integrator (see Appendix A), and use a suite of *attractor tests* to cull systems that fail to exhibit chaotic behavior. First, transient systems that converge to a fixed point or diverge to infinity are filtered. Then, we apply the chaos 0-1 test, which distinguishes quasiperiodic dynamics from true chaos (Falconer et al., 2007). We also apply a near-recurrence test to reject limit cycles, a power spectrum test to reject trajectories with only a few distinct sharp peaks, and the data-driven Rosenstein estimator (Rosenstein et al., 1993) to ensure a positive maximum Lyapunov exponent. Finally, we filter for stationarity using the Kwiatkowski-Phillips-Schmidt-Shin (KPSS) (Kwiatkowski et al., 1992) and augmented Dickey–Fuller (ADF) (Dickey & Fuller, 1979) statistical tests.

**Augmentations.** On top of the integrated trajectories, we expand the training data by applying dynamics-inspired augmentations that preserve the property that the transformed timeseries arise from a closed nonlinear dynamical system. Our augmentations are: *Random time-delay embedding* $x_i(t) \to x_i(t - \tau_i)$, $\tau_i \sim \mathcal{U}(1, d_{embed})$. This augmentation produces dynamics diffeomorphic to the original trajectory due to Takens' embedding theorem (Packard et al., 1980; Takens, 1981). Given $X \in \mathbb{R}^{C \times T}$ and $d \sim \mathcal{U}(d_{min}, d_{max})$, *Convex combinations* take random linear combinations of coordinates with coefficients sampled from a Dirichlet distribution: $X \leftarrow MX \in \mathbb{R}^{d \times T}$; $M \in \mathbb{R}^{d \times C}$, $M_{i,:} \sim \text{Dir}(\alpha \mathbf{1}_C)$. *Affine transforms* implement $X \leftarrow AX + b$, $[A\ b] \in \mathbb{R}^{d \times (c+1)}$, $[A\ b]_{ij} \sim \mathcal{N}(0, \sigma^2)/\sqrt{d}$. We set $d_{min} = 3$, $d_{max} = 10$, and $d_{embed} = 10$ for our experiments.

**Standardization.** For all trajectories, we apply instance-normalization to standardize the scales per channel. For integration, we standardize the integration horizon and granularity based on the number of timepoints (4096) and dominant timescale; note, however, that the numerical integrator ultimately dictates the stepsizes (Gilpin, 2021; Rosenstein et al., 1993). We observe no decrease in the range of invariant properties (maximum Lyapunov exponents, fractal dimension) across generations, suggesting that the starting population is sufficiently large and diverse (see Appendix A more details).

**Held-out systems.** For our zero-shot test metrics, we evaluate on an unseen set of $9.3 \times 10^3$ systems. We form the test set by holding out a random subset of 20 systems from the 129 founding system population and ensure that none of these systems or their descendants (systems where the parent is the driver or response) appear in the training set. We then evolve these systems into the test set and include all skew product systems descended from these held-out systems.

## 4  MODEL ARCHITECTURE

Dynamical systems differ from traditional time series, and so we introduce a novel architecture motivated by dynamical systems theory (Fig. 1B). Time series foundation models with causal decoders that tokenize time series on a per-observation basis tend to *parrot* motifs from their context,

leading to over-confident predictions on out-of-domain tasks (Olsson et al., 2022; Zhang & Gilpin, 2025a;b). Parroting is a useful emergent inductive bias when modeling invariant properties in long forecasts is prioritized over accuracy — otherwise known as forecasting the *climate*. However, we opt for an encoder-only, fixed prediction horizon forecast model that maximizes short-term pointwise accuracy, known as predicting the *weather* in SciML.

*Panda* generalizes PatchTST, a transformer for univariate forecasting that tokenizes timeseries on a per-patch basis (Nie et al., 2022). Section 5.2 shows that univariate-only architectures underperform on dynamical systems, motivating channel attention. Moreover, patching admits an inductive bias for dynamical systems due to Takens' theorem which states that time-delayed copies of a low-dimensional measurement of a dynamical system result in a multivariate time series that preserves key topological features of the true attractor (Packard et al., 1980; Takens, 1981).

**Patching.** We tokenize a length $T$ trajectory $\mathcal{T} \in \mathbb{R}^{C \times T}$ by patching it into a token sequence of size $P$ patches with stride $S$ so that in general, $\mathcal{T}_{P,S} \in \mathbb{R}^{C \times (\lfloor \frac{T-P}{S} \rfloor + 1) \times P}$. We choose stride $S = P$ so that the token sequences are $\mathcal{T}_P \in \mathbb{R}^{C \times (T/P) \times P}$. We choose $P = 16$, unless stated otherwise.

**Dynamics Embedding.** We lift the patched multivariate timeseries to a higher-dimensional embedding space ($d_{\text{model}}$) by concatenating each patch token $\mathcal{P} \in \mathbb{R}^{C \times P}$ with random polynomial and random Fourier features. For random polynomial features with degree $d$, we sample a fixed index set $\mathcal{I} \subset \{1, \ldots, P\}^d$ of $|\mathcal{I}| = N_{\text{poly}}$ (number of features) $d$-tuples such that for $I \in \mathcal{I}$: $\Phi_{c,i}(\mathcal{P}) := \Pi_{j=1}^d \mathcal{P}_{c,I_j} = \mathcal{P}_{c,I_1} \cdot \ldots \cdot \mathcal{P}_{c,I_d}$. The random Fourier features sample parameters $W \in \mathbb{R}^{P \times (N_{\text{rff}}/2)}$, $b \in \mathbb{R}^{N_{\text{rff}}/2}$ such that $W_{ij}, b_i \sim \mathcal{N}(0, \sigma^2)$ and $\mathcal{F}(\mathcal{P}) := [\sin(PW + b) \quad \cos(PW + b)] \in \mathbb{R}^{C \times N_{\text{rff}}}$, where $b$ added across channels (Rahimi & Recht, 2007). The overall patch embedding is $\mathcal{E}(\mathcal{P}) := [P \quad \Phi(\mathcal{P}) \quad \mathcal{F}(\mathcal{P})] \in \mathbb{R}^{C \times (P + N_{\text{poly}} + N_{\text{rff}})}$. We use degrees $d \in \{2, 3\}$ and choose $N_{\text{poly}}$ and $N_{\text{rff}}$ such that $d_{\text{model}} = P + N_{\text{poly}} + N_{\text{rff}} = 512$. The use of polynomial and Fourier features as a lifted dynamics embedding is motivated by prior approximations of the Koopman operator via extended dynamic mode decomposition (eDMD) (Kutz et al., 2016; Williams et al., 2015) and next-generation reservoir computers, which use polynomial features to forecast chaotic systems (Gauthier et al., 2021). See Appendix B for our choices of hyperparameters.

**Temporal Attention.** We mix information over the temporal dimension by taking the channel dimension as a batch dimension and performing self-attention with $p$-RoPE (Barbero et al., 2025) (a modification of rotary positional encoding, RoPE (Su et al., 2023)) over the $T/P$ univariate patches of dimension $d_{\text{model}}$. For all experiments, we use a RoPE wavelength of 500 and $p = 75\%$.

**Multivariate Attention.** Several time series foundation models are univariate, and thus, channel-independent; they solely employ temporal attention for information mixing (Nie et al., 2022). However, chaotic dynamical systems exhibit strong channel coupling. We demonstrate this empirically for the electronic circuits dataset in Fig. 4D, where we show the benefit of channel attention as the coupling strength increases. We interleave channel attention layers without positional encoding after each temporal attention layer. Each layer transposes the token sequence, treating the token dimension as a batch dimension and the channels as a set before self-attention $\text{ChannelAttention}(\mathcal{T}_P) := \text{SelfAttention}(\mathcal{T}_P^\top)$, $\mathcal{T}_P^\top \in \mathbb{R}^{T/P \times C \times d_{\text{model}}}$. Temporal attention is followed by a feedforward residual network, GeLU activations (Hendrycks & Gimpel, 2016), and RMSNorm (Zhang & Sennrich, 2019). In the prediction head, processed tokens are aggregated along the sequence dimension $T/P$ and mapped with a linear layer into a length $H$ channel-wise forecast. The architecture is further described in Appendix B.

## 5 RESULTS

### 5.1 *Panda* ZERO-SHOT FORECASTS UNSEEN NONLINEAR DYNAMICS

To evaluate the quality of the generated forecasts, we measure (1) short-term forecast accuracy via mean squared error (MSE), mean absolute error (MAE), symmetric mean absolute percentage error (sMAPE), and Spearman correlation, as well as (2) attractor reconstruction accuracy via correlation dimension, spectral Hellinger distance, and Kullback-Leibler (KL) divergence from the ground-truth attractor. For brevity, we report only the sMAPE and MAE (short-term), and KL divergence and spectral Hellinger distance (global) in the main text; the other metrics show similar results and are

included in the Appendix D and Appendix C. We compute all metrics for forecasts generated from zeroshot (held-out) systems never seen during training. Specifically, these are $N_{\text{test}} = 9.3 \times 10^3$ unique skew-product dynamical systems found using the methodology described in section 3. We additionally include results for scaling up model size and training dataset size in Appendix K.

**Comparison to baseline models.** We train *Panda* with *21M* parameters and evaluate against several time series foundation models of comparable or larger scale: *Chronos 20M*, a causal univariate model which was recently shown to produce competitive forecasts of chaos systems (Ansari et al., 2024; Zhang & Gilpin, 2025b). *Chronos 20M SFT*: Chronos supervised-finetuned on our entire chaotic systems dataset (Section 3). *Time MOE 50M*: A 50M parameter univariate model based on sparse mixture of experts (Shi et al., 2024). *TimesFM 200M*: A patch-based 200M parameter decoder-only univariate model (Das et al., 2024). *DynaMix*: A parameter-efficient recurrent multivariate forecasting model pretrained on 3D dynamical systems (Hemmer & Durstewitz, 2025). For univariate models, each dimension is forecast independently.

Across $9.3 \times 10^3$ held-out systems, we find *Panda* outperforms the baselines across a variety of prediction horizons and error metrics (Fig. 2). While we train our model exclusively on $d = 3$-dimensional dynamical systems, the evaluation set includes arbitrary dimension systems, indicating that channel attention enables multivariate generalization. Moreover, we use window autoregression to extend our evaluation forecast horizon well beyond the forecast horizon used during training. Our model maintains its performance advantage, indicating that it learns an effective dynamical propagator independent of a single timescale. In Appendix D, we show that our results are robust to the choice of metric (see Fig. 15).

**Ablations.** We also ablate several features of *Panda*, in order to verify the contributions of our dynamics-based architectural choices. These include **(1)** Channel Attention, **(2)** Dynamics Embedding, and **(3)** Masked Language Modeling (MLM) Pretraining.

We observe a significant improvement due to channel attention and MLM pretraining (See Section E for example zero-shot completions). However, the combined effect of the MLM with the dynamics embedding appears to be more complex: with no MLM, the dynamics embedding helps, but with MLM, it reduces performance. Moreover, the dynamics embedding improves the error on autoregressive rollout, whereas MLM reduces performance on rollout. We conclude that using the dynamics embedding with polynomial features (*PolyEmbed*) gives us the best model for long prediction horizons.

We include additional forecast metrics in Fig. 16 in Appendix D. We continue the discussion and evaluation of MLM on the completions task in Appendix E. In particular, we compare the correlation dimension of the completions against that of the ground truth trajectories (Fig. 18) and show a strong match. Furthermore, we investigate the effect of patch size on *Panda*'s performance in Appendix J.

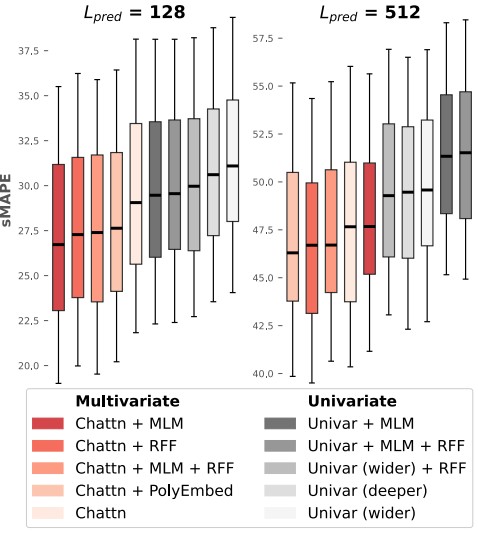

Figure 3: Ablations of key architectural features of *Panda*: MLM pretraining, channel attention (*Chattn*), and components of the dynamics embedding (*RFF* denotes random Fourier features and *PolyEmbed* includes polynomial features).

## 5.2 *Panda* ZERO-SHOT FORECASTS EXPERIMENTAL DATA

We next show that *Panda* generalizes to experimental time series from real-world dynamical systems. These experimental datasets have nonstationarity, missing values, noise, and other complexities not seen during training. Following prior works, we select systems in which the experimental data is known to have an underlying dynamical process generating it: the positions and momenta of the tips of the two rods in an experimental recording of a double pendulum (Asseman et al., 2018), the leading independent components of body posture from a light microscopy video of *C. elegans* worms crawling on agar gel (Ahamed et al., 2021), and voltage recordings from networks of 28 randomly

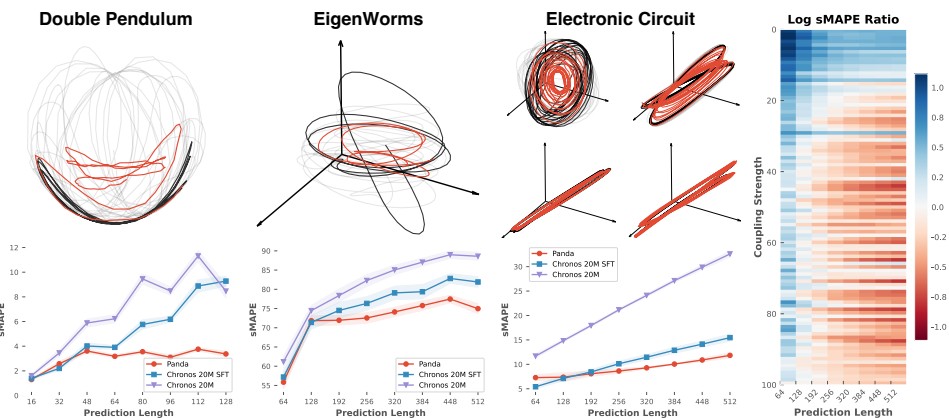

Figure 4: Zero-shot forecasts of experimental data from (a) Double Pendulum (Asseman et al., 2018), (b) Eigenworms (Ahamed et al., 2021), and (c) Electronic Circuit (Vera-Ávila et al., 2020). (d) Relative change in forecast error for *Panda* compared to Chronos-SFT (as measured in $\log(\text{sMAPE}_{Panda}/\text{sMAPE}_{Chronos\text{-}SFT})$), showing the advantage of our approach as the coupling strength between variables increases, for various prediction horizons.

.

connected electrical oscillators (Vera-Ávila et al., 2020). In all cases, the zero-shot performance of *Panda* outperforms Chronos-SFT (Fig. 4a).

For the circuit dataset in particular, we find that as the experimental coupling strength increases, the relative advantage of *Panda* over Chronos-SFT increases (red regions), particularly at long prediction horizons—leading to a visible Pareto front between the two models (Fig. 4b). This finding underscores the importance of channel attention for capturing nonlinear couplings typical in real world dynamical systems.

### 5.3 *Panda* EXHIBITS A DYNAMICAL SYSTEMS SCALING LAW

We create eight independent pretraining datasets that are subsets of the $2 \times 10^4$ unique systems generated using our methodology in Section 3. Across these eight datasets, we maintain a constant number of total timepoints while taking, at one extreme, a single trajectory (one initial condition) from each unique system, and at the other extreme, several trajectories (multiple initial conditions) from only a few unique systems. These datasets thus allow us to measure how dynamical diversity (unique systems versus initial conditions) affects generalization. We repeat our zero-shot evaluations on our set of $9.3 \times 10^3$ held-out systems for each model trained on the eight datasets.

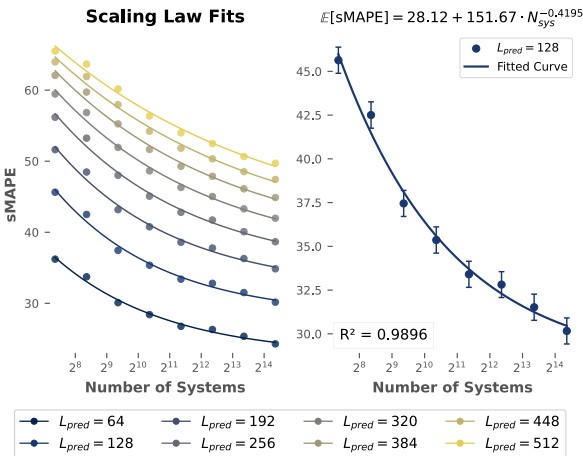

Figure 5: Scaling laws in zero-shot forecast error as the number of unique dynamical systems increases. The total amount of training timepoints is held constant.

In particular, let $N_{ics}$ be the number of sampled initial conditions and $N_{sys}$ the number of unique systems. Keeping $N_{ics} \times N_{sys}$ fixed, our eight dataset splits are constructed as $\{(N_{sys} \approx 2 \times 10^4, N_{ics} = 1), (N_{sys} \approx 10^4, N_{ics} = 2), \ldots, (N_{sys} \approx 156, N_{ics} = 128)\}$, where each subsequent split uses a strict subset of the systems of the previous split, but with double the number of sampled initial conditions $N_{ics}$.

We observe clear scaling of zero-shot performance on unseen systems with the number of new dynamical systems encountered. We emphasize that this scaling law is distinct from traditional neural scaling laws for total training data, because we hold the number of timepoints constant while

controlling the diversity of the data (Kaplan et al., 2020). These results highlight the advantages of scaling with diverse synthetic data. This finding accords with classical nonlinear dynamics theory: additional on-attractor trajectories continuously produce new information about that particular attractor's measure (a result of Pesin's theorem), but beyond a certain point they fail to provide new topological information about winding, voids, etc (Gilmore, 1998; Pesin, 1977). The distinction between these information types partly explains the gap between in-domain and out-of-domain generalization in MLDS (Göring et al., 2024).

## 5.4 *Panda* EXHIBITS EMERGENT PDE FORECASTING CAPABILITY

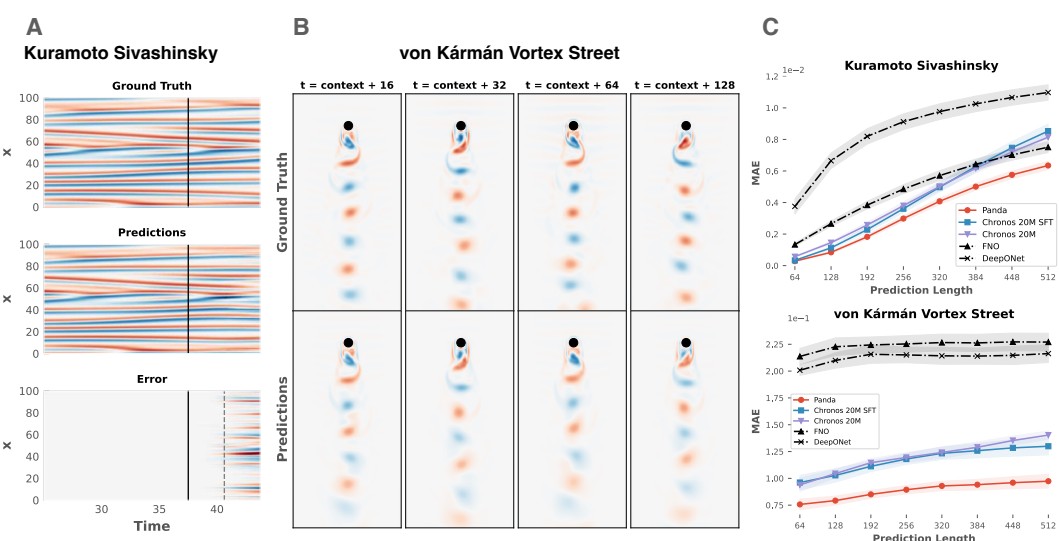

Figure 6: (A) Zero-shot forecasts of the Kuramoto-Sivashinsky equation. The time axis $t = 25$ to $t \approx 44$ contains 768 timepoints (512 context + 256 prediction): solid black line marks end of context window, and dashed gray line marks length 128 prediction horizon. (B) Zero-shot forecasts of the von Kármán vortext street. (C) The horizoned MAE (with standard errors bars) compared to baselines. We show point-wise MAE, due to sMAPE's saturation at the upper bound. We include two baselines, Fourier Neural Operators and DeepONet (black traces) fully-trained on the context (see Appendix H).

Partial differential equations (PDEs) are dynamical systems on continuous domains, with diverse applications in weather prediction or materials science (Kochkov et al., 2024). Conceptually, PDEs may be seen as coupled ordinary differential equations evolving in an infinite-dimensional space. We apply our trained model to the problem of forecasting two weakly-turbulent PDEs representing standard SciML benchmarks: the Von-Karman vortex street (VKVS) describes the unsteady motion of flow past a cylinder, and the Kuramoto-Sivashinksy (KS) models a laminar flame front (Cvitanović et al., 2010). More details on the PDE evaluation setup can be found in Appendix H.

Surprisingly, *Panda* outperforms baselines in zero-shot forecasting these systems (Fig. 6), *despite having never encountered PDE during training*. Unlike baselines, our model predicts nonlinear phenomena like merging of flame fronts in the KS equation or vortex pinchoff in the VKVS. While prior works require specially-trained models to forecast chaotic PDEs (Pathak et al., 2018), our zero-shot approach does not require extensive in-distribution training data, highlighting the advantages of cross-channel attention and multivariate training in generalization.

## 5.5 *Panda* DEVELOPS INTERPRETABLE INTERNAL REPRESENTATIONS OF COMPLEX DYNAMICS

To probe the role of channel attention in *Panda*, we feed two-tone sinusoids into the model and measure the response. The frequencies $f_1, f_2$ are each swept out over the range $[2\pi, 5\pi]$. Let $\tilde{A}$ denote the attention rollout (Abnar & Zuidema, 2020) of the temporal attention matrices. Since $\tilde{A}$ is the product of row-stochastic matrices, $\tilde{A}$ remains row-stochastic. Thus, we can measure the response

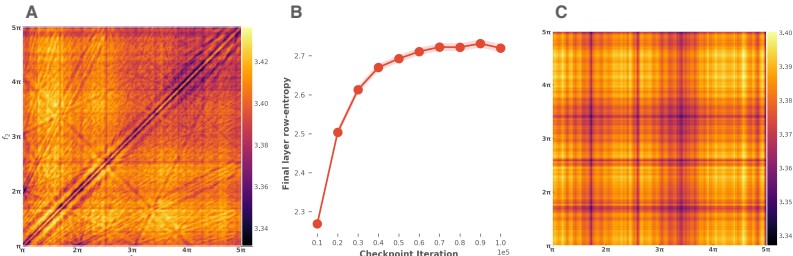

Figure 7: (A) Nonlinear resonance structure measured by average row-wise entropy of temporal attention rollout matrices. (B) Mean row-wise entropy of the final layer during training. (C) Lack of nonlinear resonance structure in the temporal attention rollout entropy for the *univariate* ablation.

from "shaking" the model at frequency mixtures $f_1$, $f_2$ by measuring the average of the rowwise entropies of $\tilde{A}$ (c.f. Fig. 7A). The attention maps exhibit complex, multiscale structure indicating nonlinear resonance, a phenomenon in dynamical systems where a physical system (such as a kicked rotor or forced pendulum) exhibits gain with nonlinear dependence on the input frequencies. As a control, the frequency response of an equally trained univariate model does not exhibit the same nonlinear multiscale structure (Fig. 7C).

We next analyze *Panda*'s attention maps to probe its underlying forecast strategy. The attention maps largely concentrate mass away from the diagonals, which indicates that *Panda* effectively uses the context. In contrast, a model implementing a purely local rule (like a numerical integrator) would exhibit predominant diagonal structure, indicating that *Panda* performs more complex operations than few-step integration. For example, some attention maps form recurrence maps, which encode large-scale attractor geometry in classical nonlinear dynamics (Donner et al., 2010; Gilpin, 2025). Other layers show banding and circulant structure (Fig. 8), consistent with global integral transforms like Fourier series.

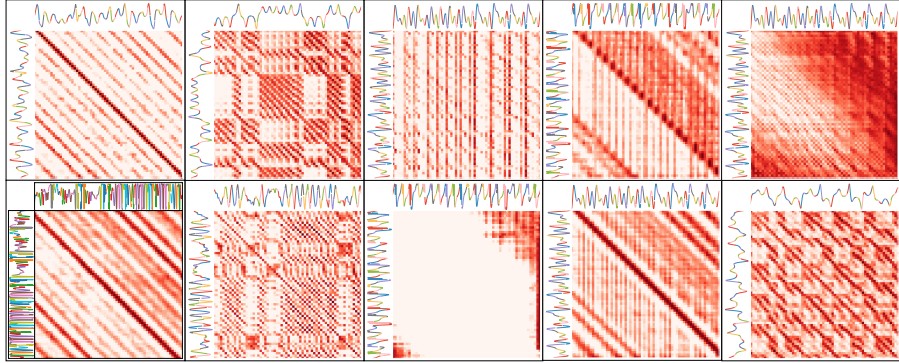

Figure 8: Temporal attention maps from *Panda* on context from different chaotic systems, showing Toeplitz, block, selector, and hybrid/combined structures (left to right). Appendix G further discusses spatiotemporal coupling and cross-channel maps (Fig. 22).

### 5.6 LONG-TERM BEHAVIOR OF PREDICTIONS

Divergence of error is inevitable when forecasting chaos with finite precision. Eventually, the prediction error grows to the point where a point forecast is useless, but invariant and geometric properties of the chaotic system can still be estimated in the long horizon regime. We quantify the utility of long horizon forecasts by measuring the geometry of forecasts much farther than $4\times$ the training prediction horizon. Specifically, we compute: the maximum Lyapunov exponents (Appendix F.6); the forward KL divergence (Table 11) between the attractor and the predictions (Hess et al., 2023); and the spectral Hellinger distance (Table 12), an $f$-divergence between power spectra of the attractor and predictions (Mikhaeil et al., 2022). We compare long-term forecast metrics in Table 1, we report mean $\pm$ std. dev. across all test systems, averaged over 5 context windows for

prediction horizons $L_{\text{pred}} \in \{512, 1024, 2048, 3072\}$. We report additional distributional metrics and computed invariant quantities in Appendix F.

We find that Panda outperforms transformer-based TSFM like Chronos and TimesFM, and that its performance remains competitive even when forecasts during testing are extended up to $8\times$ the prediction horizon used for training. We attribute this finding partly to the pretraining data imposing an inductive bias for sustained long-term dynamics, particularly because Chronos-SFT improves over Chronos across forecast horizons and metrics. However, most transformer-based TSFM trained using a loss tied to short-term forecasts, including Panda, eventually regress to the mean at sufficiently-long enough forecast horizons (Appendix M). We can quantify this failure mode by computing the distributional metrics on the tail forecasts in Appendix F.5. However, Chronos avoids this outcome because it was trained using cross-entropy on a tokenized time series, leading it to parrot sequences from context at long horizons (Fig. 31, 33, 32). This performs well on long-term metrics because ergodic dynamical systems have frequent repeated motifs known as unstable periodic orbits (Auerbach et al., 1987). A contemporaneous model to our work, *DynaMix* uses a mixture of recurrent neural networks, allowing it to better capture long term geometry than transformer-based models (Hemmer & Durstewitz, 2025). Because our experiment setting and dataset structure differs from this model's original setting, we highlight here comparisons to transformer-based TSFMs.

| | $D_{KL}$ (**Ground Truth**($L_{\text{pred}}$)$\|$**Model Prediction**($L_{\text{pred}}$)) | | | | |
|---|---|---|---|---|---|
| Model | $L_{\text{pred}} = 512$ | $L_{\text{pred}} = 1024$ | $L_{\text{pred}} = 2048$ | $L_{\text{pred}} = 3072$ | $L_{\text{pred}} = 3584$ |
| **Panda** | $\mathbf{3.93 \pm 3.51}$ | $\mathbf{4.72 \pm 3.64}$ | $\mathbf{5.63 \pm 3.71}$ | $6.14 \pm 3.68$ | $6.39 \pm 3.90$ |
| Chronos 20M SFT | $4.72 \pm 5.00$ | $5.09 \pm 4.90$ | $5.62 \pm 4.86$ | $\mathbf{5.93 \pm 4.84}$ | $\mathbf{6.05 \pm 5.34}$ |
| Chronos 20M | $5.99 \pm 5.07$ | $6.19 \pm 4.85$ | $6.51 \pm 4.76$ | $6.76 \pm 4.74$ | $6.94 \pm 5.41$ |
| Chronos 200M | $5.12 \pm 5.25$ | $5.49 \pm 5.22$ | $6.05 \pm 5.30$ | $6.36 \pm 5.28$ | $6.47 \pm 5.67$ |
| $\Delta\%\,(\uparrow)$ | $+16.7\%$ | $+7.3\%$ | $+0.0\%$ | $-3.5\%$ | $-5.6\%$ |
| | **Average** $H^2(S_{\text{Ground Truth } (L_{\text{pred}})}\|S_{\text{Model Prediction } (L_{\text{pred}})})$ | | | | |
| **Panda** | $\mathbf{0.25 \pm 0.14}$ | $\mathbf{0.25 \pm 0.12}$ | $\mathbf{0.25 \pm 0.11}$ | $\mathbf{0.25 \pm 0.11}$ | $\mathbf{0.26 \pm 0.12}$ |
| Chronos 20M SFT | $0.29 \pm 0.17$ | $0.29 \pm 0.16$ | $0.29 \pm 0.16$ | $0.30 \pm 0.16$ | $0.30 \pm 0.18$ |
| Chronos 20M | $0.37 \pm 0.16$ | $0.36 \pm 0.16$ | $0.37 \pm 0.16$ | $0.38 \pm 0.16$ | $0.38 \pm 0.17$ |
| Chronos 200M | $0.28 \pm 0.16$ | $0.28 \pm 0.15$ | $0.29 \pm 0.15$ | $0.30 \pm 0.15$ | $0.30 \pm 0.17$ |
| $\Delta\%\,(\uparrow)$ | $+10.7\%$ | $+10.7\%$ | $+13.8\%$ | $+16.7\%$ | $+13.3\%$ |

Table 1: KL divergence (top) and Hellinger Distance (bottom) between ground truth and model predictions. $\Delta\%$ denotes percent gain of *Panda* over the best baseline. See Table 13 for per-system differences, and Appendix F.1 for implementation details.

## 6 CONCLUSION AND FUTURE DIRECTIONS

Our work demonstrates the feasibility of pretrained models in discovering generalizable properties of dynamical systems, mathematical objects of intrinsic interest to the SciML and forecasting communities. Our model's emergent ability to predict higher-dimensional partial differential equations, and the scaling of its performance with the diversity of dynamical systems, show that its generalization signal stems from unique properties of dynamics relative to time series.

A limitation of our work stems from our focus on low-dimensional dynamical systems. We argue that low-dimensional dynamics are the building block for higher-dimensional systems like weather fronts or spiking neurons, because they capture essential properties like bifurcations that become more complex in extended systems. A future variant of our approach for high-dimensional dynamics could exploit the structure of coupling such as sparsity or blocks typical in these systems by allowing the channel attention layers to receive custom attention masks. Another limitation is the degradation of rollout performance from MLM pretraining. Future work will investigate the question of what pretraining task is most natural for modeling dynamical systems. We believe this is a basic question that necessitates further progress in SciML.

## 7 ACKNOWLEDGEMENTS

JL was supported by the UT CSEM Fellowship. AB was supported by the UT PGEF Fellowship and the Basdall Gardner Memorial Fellowship. WG was supported by NSF DMS 2436233 and NSF CMMI 2440490. This project has been made possible in part by Grant No. DAF2023-329596 from the Chan Zuckerberg Initiative DAF, an advised fund of Silicon Valley Community Foundation. Financial support for this publication results from grant CS-CSA-2026-075 from Research Corporation for Science Advancement. Computational analyses were performed using the Biomedical Research Computing Facility at UT Austin, Center for Biomedical Research Support (RRID: SCR_021979). The authors acknowledge the Texas Advanced Computing Center (TACC) at The University of Texas at Austin for providing computational resources.

## 8 REPRODUCIBILITY STATEMENT

Python code is available at https://github.com/abao1999/panda. Weights for the original model are available at https://huggingface.co/GilpinLab/panda, and the larger forecast checkpoint https://huggingface.co/GilpinLab/panda-72M and MLM checkpoint https://huggingface.co/GilpinLab/panda_mlm-66M.

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

## A  GENERATION OF A NOVEL CHAOTIC SYSTEMS DATASET

### A.1  SKEW-PRODUCT SYSTEMS

We algorithmically discover skew-product systems following the methodology described in Section 3. Here, we present a subset of 30 of these systems, out of a total of $2 \times 10^4$ in our training set.

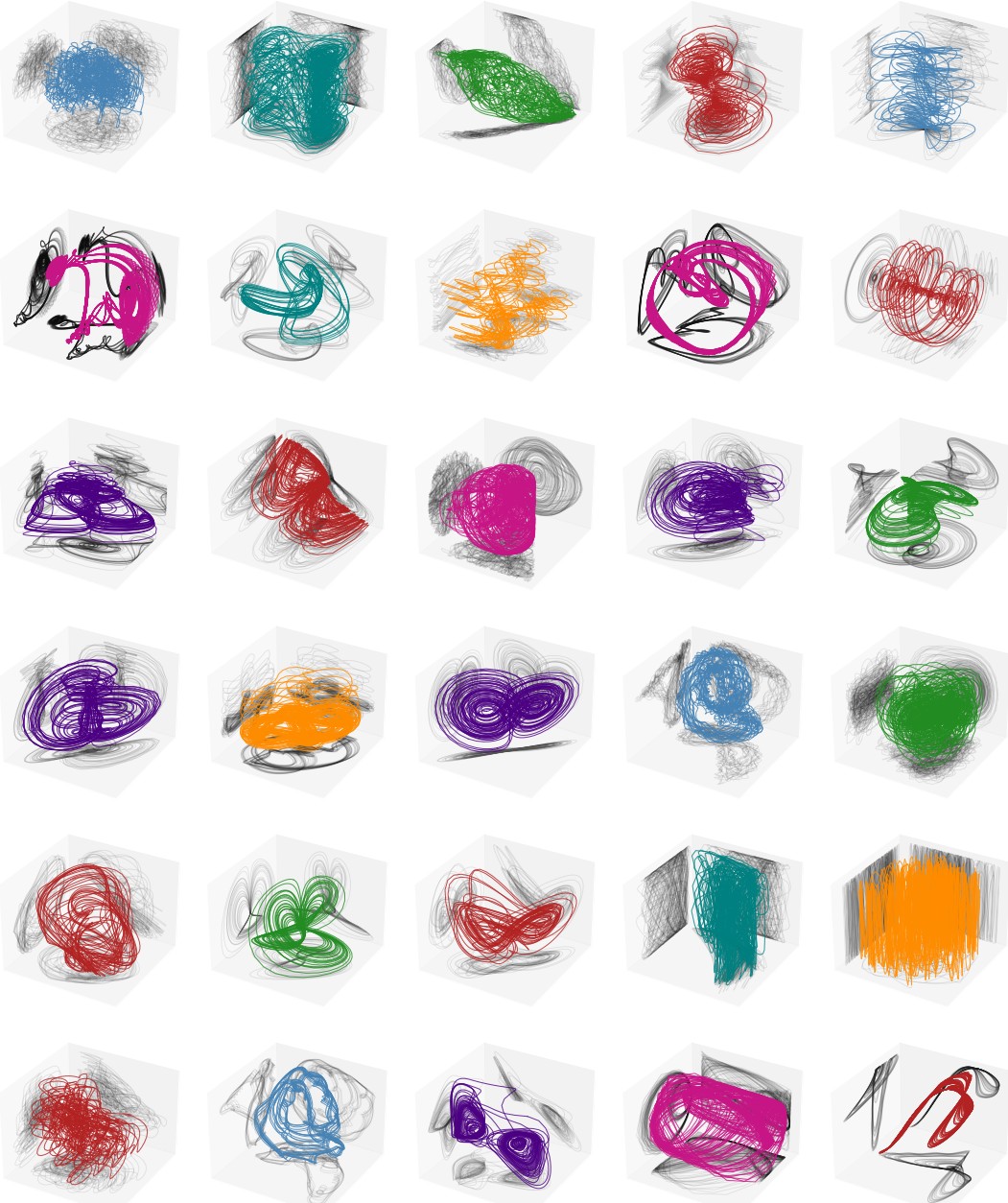

Figure 9: Examples of novel chaotic skew-product systems discovered via evolutionary search. Shaded regions correspond to two-dimensional projections onto the corresponding axes.

Our starting point is a hand-curated, crowdsourced public dataset of 129 chaotic low-dimensional dynamical systems from the nonlinear dynamics literature (Gilpin, 2021; 2023; Zhang & Gilpin, 2025b). Each entry comprises a set of coupled ordinary differential equations with dimensionality between three and ten. The parameters and initial conditions for each system have been hand-tuned into the chaotic regime, based on values used in previously-published studies.

## A.2 MUTATION OF BASE SYSTEMS

We also generate new instances of the base 129 chaotic systems by perturbing the ODE parameters.

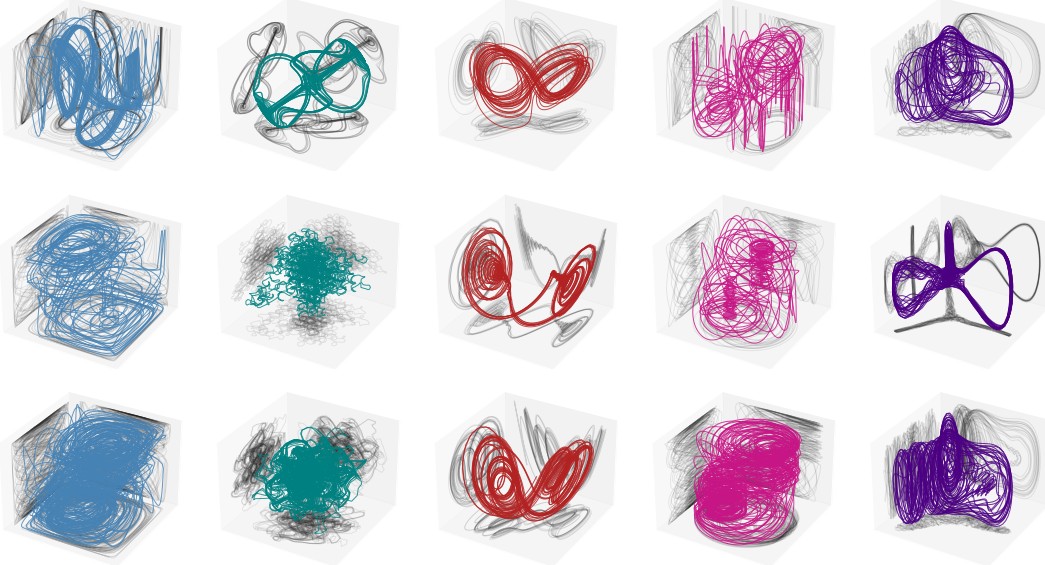

Figure 10: Examples of parameter perturbations of base systems. **Top row** Unperturbed original systems. **Botton rows** Parameter perturbations of the top row systems.

## A.3 NUMERICAL INTEGRATION

For all ODEs, we integrate trajectories of 4096 timesteps with an integration time-span dictated by a system's characteristic timescale based off the dominant modes in the power spectrum. We thus call this timescale the *period* of the system and set the integration time-span to be $[0, N_p \times \varphi]$ where $N_p$ is the number of periods, and $\varphi$ is the "period" measured from integrating test trajectories of the base system; for skew systems we take the period to be the maximum period between the driver and response systems. For all experiments in the main text, we take $N_p = 40$, but use a larger mixed-period dataset in our scaled up experiments (Appendix K).

The numerical integration timestep is controlled via adaptive step-sizing from the *Radau* solver, a 5th order implicit Runge Kutta Scheme. For high quality trajectory data, we integrate using a relative tolerance $1 \times 10^{-9}$ and an absolute tolerance of $1 \times 10^{-10}$. The initial conditions for discovered systems are obtained by integrating a test trajectory at a lower tolerance (rtol $= 1e-6$, atol $= 1e-7$) and sampling a point from the coarse trajectory which approximates starting at a point *on attractor*.

## A.4 ATTRACTOR SELECTION

The only general way to identify properties about chaotic dynamical systems is to integrate them. This fundamental fact makes the system discovery process described in Section 3 very expensive at scale. To effectively reduce the number of incoming candidates for chaoticity selection and validation, we employ callbacks during integration that will immediately kill the process and prune that system candidate. Specifically, we terminate integration whenever the step size falls below $10^{-10}$, any bounded (non-driving dimension) coordinate exceeds $10^4$ in value, and whenever the integration time exceeds 5 minutes. The surviving systems will finish integration and move on to the chaoticity selection phase (see the overview of our selection for chaoticity in Section 3).

### A.5 DATASET PROPERTIES

We verify that our integrated trajectories exhibit chaotic dynamics by measuring the average number of Lyapunov times in various prediction horizons. A chaotic flow separates nearby initial conditions according to $|\delta(t)| \approx \exp(\lambda_1 t)|\delta(0)|$ where $\delta$ is the time dependent separation and $\lambda_1$ is the maximum Lyapunov exponent. Thus, a Lyapunov time is defined to be $T_{\text{Lyap}} := 1/\lambda_1$. Given an arbitrary time series with timestep $\delta t$, the Lyapunov times per $N$ timepoints is then $\lambda_1 \times \delta t \times N$. Since we rely on an implicit integrator with adaptive step-sizing, we compute the average timestep over the integration timespan and estimate the maximum Lyapunov exponent using the Rosenstein estimator (Rosenstein et al., 1993) to compute the distribution of Lyapunov times per horizon length in Fig. 12a where it is clear that we are predicting in the chaotic regime most of the time.

To ensure consistency between the founder population and offspring, we featurize all pretraining time series using the same procedure as previous works reporting chaotic systems datasets (Gilpin, 2021; 2023). For each channel of a D-dimensional multivariate time series (4096 timepoints, 100 points per dominant Fourier period) we compute a vector of 749 standard time-shift invariant time series features like wavelet modes, signal power, reversion rate, etc. using the `tsfresh` library (Christ et al., 2018). We average the $D$ feature vectors for each system to produce a single channel-permutation invariant feature vector for each skew-product system. We then project all $2 \times 10^4$ pretraining skew-product systems into 2D using UMAP, a nonlinear embedding algorithm that preserves the local neighbors of each point from the high dimensional space (distances, however, are not necessarily preserved) (McInnes et al., 2018). We next featurize and embed the 135 parent systems from the founder population into the same space. We observe broad dispersion of the parent systems across the child population, implying the absence of mode collapse or strong distribution shift between the parents and offspring (Fig. 11). We interpret this result as the absence of strong founder or bottleneck effects in the offspring generation.

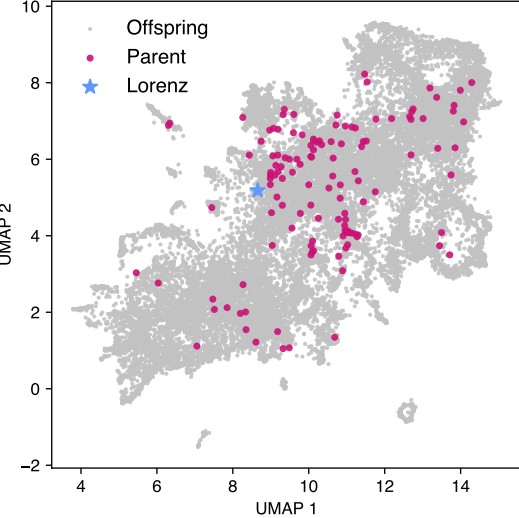

Figure 11: A low-dimensional embedding of the $2 \times 10^4$ skew-product systems (gray), as well as the 135 founding parent systems (magenta) from which these offspring systems are evolved. The well-known chaotic Lorenz attractor is starred on the plot.

Additionally, for all trajectories in the test set, we measure the empirical stiffness score defined as $S := \log_{10} (\max_t |\Delta_t|/\mathbb{E}_t|\Delta_t|)$, where $\Delta_t$ is the finite difference (forward or backward) at time $t$. Fig. 12b shows that most test systems have at least an order of magnitude scaling between the largest observation jump compared to the average change *per channel*. This distribution suggests that the dataset generation algorithm generates stiff systems and reinforces the fact that the integrated trajectories exhibit non-trivial dynamics.

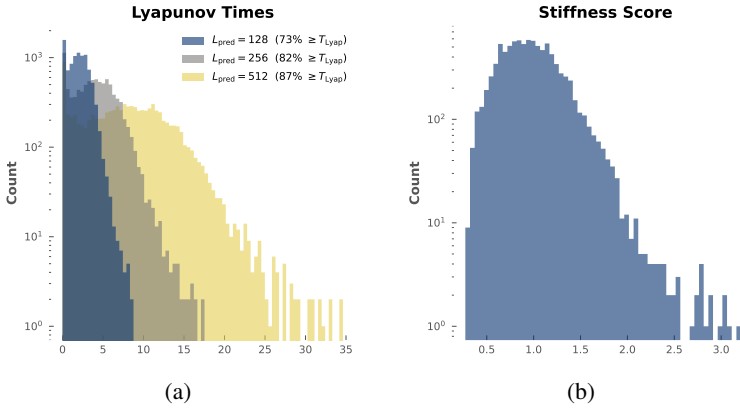

(a)       (b)

Figure 12: Dynamical properties of systems in the test set. (a) Distribution of Lyapunov times within $L_{\mathrm{pred}}$ timepoints; annotated with the proportion of systems which exceed 1 Lyapunov time in the horizon. (b) Distribution of sitffness scores (log-ratio of largest delta compared to the average delta).

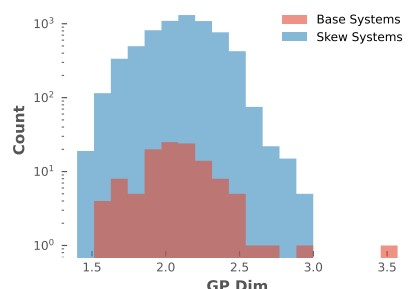

Figure 13: Distribution of correlation dimension (Grassberger-Procaccia) of skew systems and their founder (base) systems.

| System | GP Dim (mean $\pm$ std) |
|---|---|
| Base Systems | $2.09 \pm 0.27$ |
| Skew Systems | $2.11 \pm 0.23$ |

Table 2: Correlation dimension for Base (Founder) systems and Skew (Children) systems.

As shown in Fig. 13, our skew-product generation preserves the distribution of the correlation dimension, an invariant quantity used as a proxy for fractal dimension. This suggests that our dataset **does not** suffer from a "founder effect" that would kill off diversity. Table 3 further presents a comparison of the Kullback Leibler divergence between the invariant measures of the attractors, for skew systems: with the same parents; with different parents; with one parent shared; and between parent (Base) and child (Skew) systems.

| Metric | mean $\pm$ std | $N_{\mathbf{combos}}$ |
|---|---|---|
| $D_{KL}(\text{Skew} \,||\, \text{Response})$ | $5.35 \pm 5.81$ | 10,000 |
| $D_{KL}(\text{Skew} \,||\, \text{Driver})$ | $8.46 \pm 6.58$ | 10,000 |
| $D_{KL}(\text{Skew} \,||\, \text{Non-Parent})$ | $9.01 \pm 6.64$ | 10,000 |
| $D_{KL}(\text{Skew Intra})$ | $3.39 \pm 5.11$ | 10,000 |
| $D_{KL}(\text{Skew Inter})$ | $7.07 \pm 6.58$ | 10,000 |
| $D_{KL}(\text{Base Intra})$ | $2.54 \pm 4.13$ | 6,000 |
| $D_{KL}(\text{Base Inter})$ | $8.24 \pm 6.48$ | 6,000 |

Table 3: $D_{KL}$ between skew systems and: 1) param perts of response; 2) param perts of driver; 3) param perts of non-parent system in the founder pool. (Skew Intra) $D_{KL}$ between param perts of skew systems with the same parents. (Skew Inter) $D_{KL}$ between param perts of skew systems with different parents. (Base Intra) $D_{KL}$ between parameter perturbations of the same founder system. (Base Inter): $D_{KL}$ between parameter perturbations of different founder systems.

Table 4: Model Architecture

| Parameter | Value |
|---|---|
| Context length | 512 |
| Prediction length | 128 |
| Hidden layers | 8 |
| $d_{\mathrm{model}}$ | 512 |
| FFN dimension | 512 |
| Attention heads | 8 |
| Activation | gelu |
| Pre-norm | True |
| Normalization | RMSNorm |
| Init std | 0.02 |

Table 5: Model Architecture (Continued)

| Parameter | Value |
|---|---|
| Patch length / stride | 16 / 16 |
| Rope percent | 0.75 |
| Max wavelength | 500 |
| Poly features | 120 |
| Poly degrees | 2 |
| RFF count | 256 |
| RFF scale | 1.0 |

## B  TRAINING DETAILS

A technical difficulty of training a global multivariate model is forming batches of trajectories with mixed channel dimensions. We look to dynamical systems theory and note that it is well known that at least 3 coupled dynamical variables are necessary for a system to exhibit deterministic chaos in continuous-time (Strogatz, 2018). To this end, we fix the dimensions of each input trajectory to 3 *only during training* by randomly sampling 3 channels from each multivariate trajectory to enable efficient batching. During inference time, we process the full multivariate trajectories. For the 21.3M parameter *Panda* model we use $d_{\mathrm{model}} = d_{\mathrm{ffn}} = 512$, $N_{\mathrm{heads}} = N_{layers} = 8$, $N_{\mathrm{poly}} = 120$ with degree 2, and $N_{\mathrm{rff}} = 256$. For the 41.5M parameter model (Appendix K), we use $d_{\mathrm{model}} = d_{\mathrm{ffn}} = 640$, $N_{\mathrm{heads}} = N_{layers} = 10$, $N_{\mathrm{poly}} = 156$ with degree 2, and $N_{\mathrm{rff}} = 312$. And for the 71.5M parameter *Panda-72M* (Appendix K), we use $d_{\mathrm{model}} = d_{\mathrm{ffn}} = 768$, $N_{\mathrm{heads}} = N_{layers} = 12$, $N_{\mathrm{poly}} = 188$ with degree 2, and $N_{\mathrm{rff}} = 376$. Additionally, data augmentations (Section 3) are uniformly randomly applied to 20% of the training trajectories.

We use a patch size (and patch stride) of 16. All models are trained with a context length of 512, which corresponds to 32 patches, and use a non-causal transformer encoder with 8 layers, each with $d_{\mathrm{model}} = 512$ and 8 heads. Each attention block maps a (`batch size, channels, patches, hidden`) sized hidden state $H$ via:

$$H \leftarrow H + \mathrm{RopeTemporalAttention} \circ \mathrm{RMSNorm}(H)$$
$$H \leftarrow H + \mathrm{ChannelAttention} \circ \mathrm{RMSNorm}(H)^{\top}$$
$$H \leftarrow H + \mathrm{FFN} \circ \mathrm{RMSNorm}(H)$$

Where the transpose is applied to the channel and patch (sequence) dimension.

For models optimized with masked language modeling (MLM) style pretraining (masking and reconstructing intermediate patch tokens), a linear head is used to infill masked patches. For the forecasting model, a prediction head aggregates the encoder hidden states via a mean along the sequence (patch) dimension and a linear layer maps this representation to a *fixed-length* 128 forecast for **all** models. All models are trained with MSE loss and the AdamW optimizer with a maximum learning rate $1 \times 10^{-3}$ on a cosine schedule with a 10% warmup. Additionally, we train with gradient norm clipping at a value of 1.0. See Tables 4, 5 for comprehensive details about model architecture.

The 20M *Panda MLM* models are trained for 200K iterations ($\sim$52 wallclock hours across 4 GPUs or $\sim$208 GPU hours) with a batch size of 1024 with 50% of tokens randomly masked out each batch.

The forecast models are trained for 100K iterations with a batch size of 1024 and are optionally initialized with a pretrained encoder from an MLM model (Section 5.1). The 20M parameter forecasting checkpoints are trained for $\sim$26 wallclock hours or $\sim$104 GPU hours. The Chronos-SFT models use considerably more memory during training - permitting a batch size of 160 for 300K training iterations which required $\sim$48 wallclock hours or $\sim$192 GPU hours.

## C    FORECASTS

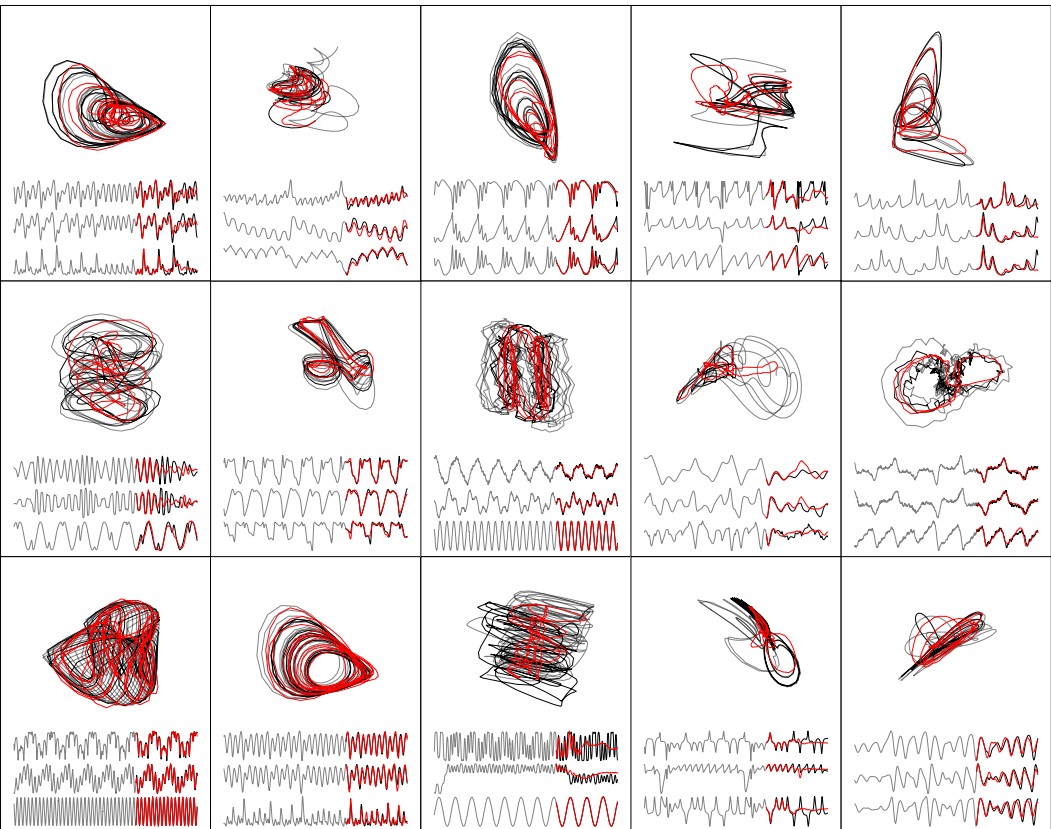

Figure 14: Examples of zero-shot forecasts ($L_{\text{pred}} = 256$) on held-out chaotic dynamical systems.

For additional forecasts, see Appendix L (Figs. 28, 29, and 30). All forecasts plotted are with prediction length $L_{\text{pred}} = 256$.

## D    ADDITIONAL FORECAST METRICS

Table 6 shows statistical significance testing of *Panda* metrics against other baselines. Note that we do not report results for TimeMoE due to the presence of NaNs, and instead test against a 200M Chronos baseline in greedy decoding and probabilistic mode. *Panda* clearly achieves lower error across the board; the gap closes with Chronos 20M SFT but still remains statistically significant.

In Fig. 2 we presented the sMAPE and MAE comparison for *Panda* versus our baseline models. We now present more zero-shot forecast metrics, but using the probabilistic forecasting mode for Chronos and Chronos-SFT. When finetuning *Chronos 20M* on our dataset (i.e. *Chronos 20M SFT*), we used the default top-k and top-p (nucleus sampling) and temperature settings. We use these same settings, *top-k* = 50, *top-p* = 1.0, *temperature* = 1.0 for the Chronos probabilistic forecasting, aggregating our metrics over 10 sample forecasts per context window per system.

Table 7 highlights the efficiency of *Panda* against Chronos. This $\approx$ 60x speedup is largely due to the fact that Chronos represents time series tokens as quantized individual points, whereas *Panda* relies on patches.

Table 6: Wilcoxon Signed Ranked test for Panda errors vs. baseline errors (Holm–Šidák adjusted $p$-values)

| Model | Prediction | MSE | | MAE | | sMAPE | |
|---|---|---|---|---|---|---|---|
| | Horizon | $p$-value | statistic | $p$-value | statistic | $p$-value | statistic |
| Chronos 20M SFT | $L = 128$ | $5.96 \times 10^{-48}$ | 13 484 | $9.01 \times 10^{-51}$ | 12 153 | $6.35 \times 10^{-3}$ | 49 421 |
| | $L = 512$ | $8.24 \times 10^{-50}$ | 12 275 | $3.07 \times 10^{-54}$ | 10 432 | $3.34 \times 10^{-2}$ | 51 252 |
| Chronos 200M | $L = 128$ | $5.84 \times 10^{-49}$ | 12 778 | $6.88 \times 10^{-57}$ | 9182 | $1.20 \times 10^{-28}$ | 23 770 |
| | $L = 512$ | $5.33 \times 10^{-48}$ | 13 233 | $1.45 \times 10^{-56}$ | 9323 | $1.36 \times 10^{-7}$ | 41 310 |
| Chronos 20M SFT Prob | $L = 128$ | $6.93 \times 10^{-41}$ | 16 851 | $3.09 \times 10^{-33}$ | 21 198 | $1.30 \times 10^{-5}$ | 44 464 |
| | $L = 512$ | $3.83 \times 10^{-41}$ | 16 513 | $1.18 \times 10^{-35}$ | 19 657 | $3.26 \times 10^{-1}$ | 54 730 |
| Chronos 200M Prob | $L = 128$ | $2.91 \times 10^{-49}$ | 12 577 | $1.22 \times 10^{-56}$ | 9290 | $5.69 \times 10^{-31}$ | 22 353 |
| | $L = 512$ | $2.37 \times 10^{-38}$ | 18 024 | $1.83 \times 10^{-49}$ | 12 541 | $7.73 \times 10^{-11}$ | 37 624 |
| TimesFM 200M | $L = 128$ | $4.05 \times 10^{-55}$ | 9863 | $1.67 \times 10^{-69}$ | 3863 | $4.89 \times 10^{-77}$ | 972 |
| | $L = 512$ | $2.50 \times 10^{-38}$ | 18 103 | $1.59 \times 10^{-59}$ | 7949 | $2.66 \times 10^{-70}$ | 3588 |
| Time MOE 50M | $L = 128$ | $1.03 \times 10^{-41}$ | 14 499 | $9.66 \times 10^{-58}$ | 7320 | $1.68 \times 10^{-76}$ | 170 |
| | $L = 512$ | $2.58 \times 10^{-29}$ | 21 152 | $3.14 \times 10^{-52}$ | 9639 | $3.13 \times 10^{-75}$ | 589 |
| DynaMix | $L = 128$ | $1.46 \times 10^{-23}$ | 26 724 | $3.03 \times 10^{-35}$ | 19 399 | $1.86 \times 10^{-40}$ | 16 533 |
| | $L = 512$ | $4.54 \times 10^{-8}$ | 39 970 | $7.00 \times 10^{-22}$ | 27 602 | $1.83 \times 10^{-27}$ | 23 592 |

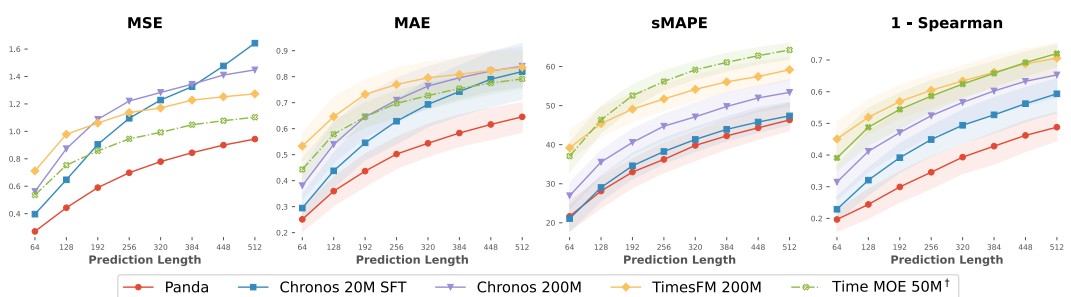

Figure 15: Zero-shot forecast metrics for our baselines, using *probabilistic* (10 samples) forecasts for the *Chronos* models. [†]Dash-dotted lines indicate presence of NaNs for some systems.

| Time per Forecast (s) | |
|---|---|
| Model | Time (mean $\pm$ std) |
| **Panda** | **$0.031 \pm 0.001$** |
| TimeMOE 50M | $0.336 \pm 0.060$ |
| TimesFM 200M | $0.605 \pm 0.032$ |
| Chronos 20M | $1.880 \pm 0.041$ |
| Chronos 200M | $4.233 \pm 0.121$ |

Table 7: Inference time per forecast ($L_{pred} = 512$), computed over $N = 1000$ calls to each model, on a single H100 GPU. Each model call uses context length 512 timesteps, from our multivariate data, which has variable number of channels (at least 3). The univariate models (Chronos, TimesFM, TimeMOE) treat the channels as batch dimension, for each call.

We also provide median forecast metrics with IQR for the metrics in Fig. 2 over multiple prediction horizons for the best baselines in Tables 8, 9.

| sMAPE Median [P25, P75] | | | |
|---|---|---|---|
| Model | $L_{pred} = 128$ | $L_{pred} = 256$ | $L_{pred} = 512$ |
| Panda | **27.6** [18.5, 39.3] | **36.7** [26.2, 47.6] | **46.3** [37.0, 57.0] |
| Chronos 20M SFT | 30.3 [21.9, 40.0] | 40.1 [30.3, 48.3] | 48.8 [37.3, 56.8] |
| Chronos 200M | 36.0 [26.4, 44.6] | 44.6 [34.6, 52.9] | 53.5 [42.8, 60.8] |
| Chronos 20M SFT Probabilistic | 29.7 [21.3, 40.7] | 39.4 [29.3, 48.7] | 48.3 [37.8, 57.1] |
| Chronos 200M Probabilistic | 36.4 [26.7, 44.7] | 45.0 [34.2, 53.1] | 53.8 [42.8, 60.6] |
| DynaMix | 47.1 [37.9, 57.9] | 54.7 [46.8, 62.6] | 60.8 [53.9, 65.8] |

Table 8: Median sMAPE and interquartile range [P25, P75]. Note: while we include DynaMix for completeness, it is a recurrent model (not a transformer) that benefits from longer context lengths than the short context settings of our experiments. Its performance improves with additional context. In contrast, most transformer-based models, including Panda, degrade when tested on longer context than they were trained on.

| MAE Median [P25, P75] | | | |
|---|---|---|---|
| Model | $L_{pred} = 128$ | $L_{pred} = 256$ | $L_{pred} = 512$ |
| Panda | **0.35** [0.22, 0.54] | **0.49** [0.35, 0.70] | **0.65** [0.48, 0.84] |
| Chronos 20M SFT | 0.56 [0.34, 0.80] | 0.85 [0.52, 1.26] | 1.25 [0.72, 2.14] |
| Chronos 200M | 0.61 [0.41, 0.80] | 0.86 [0.58, 1.18] | 1.07 [0.75, 1.83] |
| Chronos 20M SFT Probabilistic | 0.46 [0.29, 0.69] | 0.65 [0.43, 0.93] | 0.85 [0.57, 1.34] |
| Chronos 200M Probabilistic | 0.55 [0.38, 0.73] | 0.71 [0.51, 0.91] | 0.84 [0.63, 1.12] |
| DynaMix | 0.79 [0.60, 0.94] | 0.94 [0.76, 1.06] | 1.02 [0.88, 1.15] |

Table 9: Median MAE and interquartile range [P25, P75]. Note that the DynaMix comparison does not match the model's original experiment setting, see the note in Table 8

### D.0.1 ADDITIONAL METRICS FOR MODEL ABLATIONS

In Fig. 3 we presented a sMAPE comparison for several key ablations of our model. Here, we provide additional zero-shot forecast metrics for these ablations, supporting our conclusion that our dynamics embedding with polynomial features (PolyEmbed) is best for long-horizon forecasting via rollouts.

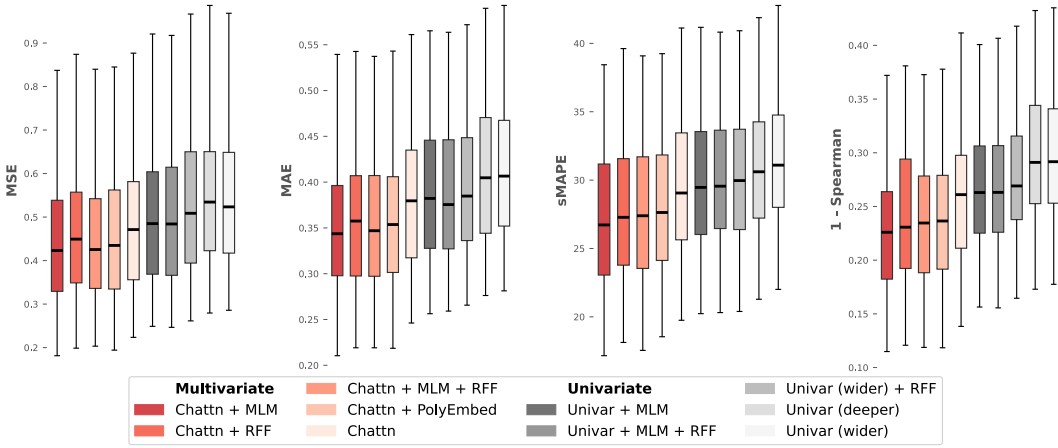

Figure 16: Zero-shot forecast metrics for our ablation experiments.

# E MLM COMPLETIONS

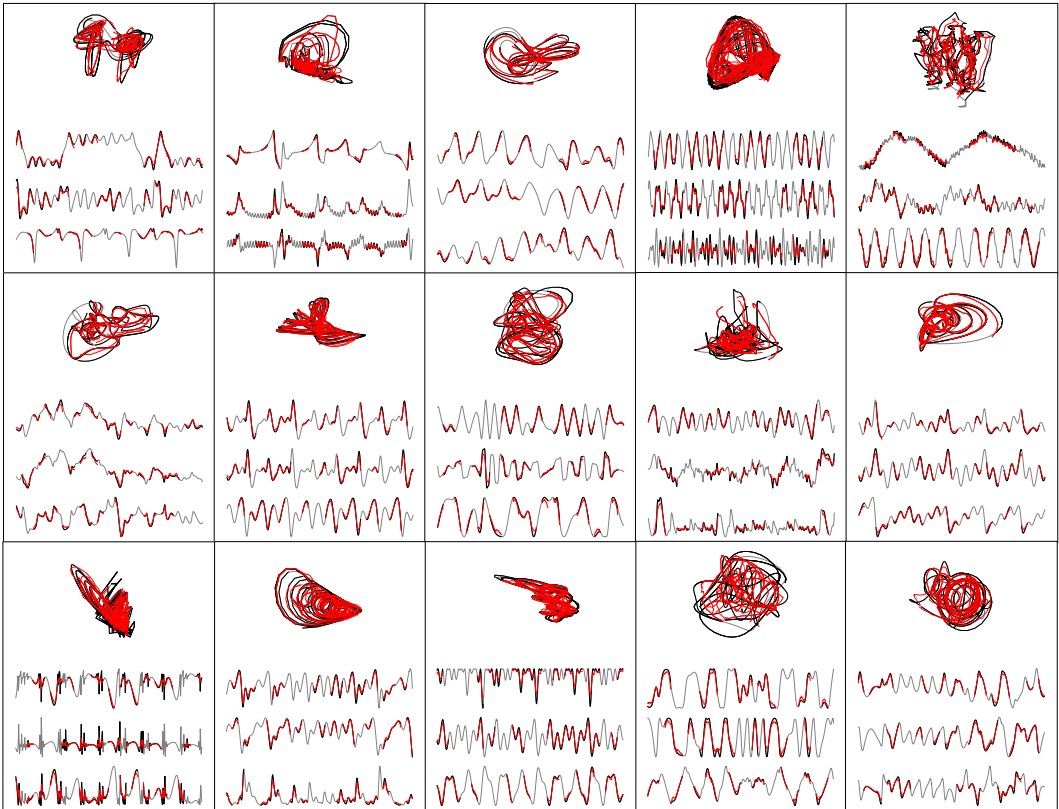

Figure 17: Examples of zero-shot completions on held-out chaotic dynamical systems. Each completion plotted was with a context length of 512 time points, with half the patches (patch length 16) randomly masked out in a channel-inconsistent manner. These plots show *Panda MLM*, our *20M* parameter checkpoint, completing the masked-out trajectories i.e. 256 time points. See Appendix N for more examples from *Panda MLM* and from our scaled-up model *Panda MLM-66M*.

We present examples of *Panda MLM* completions on our held-out test set in Fig. 17. For more examples of zero-shot completions, see Appendix N.

For the completions task, we randomly mask out half of patches for each coordinate dimension separately i.e. channel-independent masking. We trained *Panda MLM* with patch length 16 and context length 512, so each context window has 32 patches on the time axis. But we can generate completions with any context length. We refer to the masked-out portions of the trajectory as the erasures. We seek to measure how the model learns the cross-channel coupling relationships and statistical dependencies.

In future work, we hope to investigate more sophisticated masking strategies, such as masking out contiguous blocks of patches and investigating channel-dependent masking, which is closer to a forecasting task. Recall from our discussion of Fig. 3 that MLM pretraining reduces performance on autoregressive rollout (c.f. Fig. 16). Determining the optimal MLM pretraining objective for long horizon forecasts on autoregressive rollout remains an intriguing area to investigate.

To quantify the performance of our MLM checkpoint on the completion task, we compute the correlation dimension (Fig. 18) of completions versus ground truth trajectories using the Grassberger-Procaccia algorithm (Grassberger & Procaccia, 1983a;b). This algorithm was developed to quantify the strangeness (Lorenz, 1963; Ott, 1981; Ruelle & Takens, 1971) of chaotic attractors via a computable metric related to the fractal (Hausdorff) dimension and information entropy. For the result in Fig. 18, we take the entire length 4096 trajectory for each of our $9.3 \times 10^3$ held-out systems and we randomly mask out (erase) half of the patches (patch length 16) in a channel-inconsistent manner.

Let $\{x_i\}_{i=1}^T \subset \mathbb{R}^D$ be a time series of $T$ points in $D$ dimensions. First, we compute pairwise Euclidean distances (excluding $i = j$):

$$\mathcal{R} := \left\{ r_{ij} = \|x_i - x_j\|_2 \mid 1 \leq i, j \leq T, \ i \neq j \right\}$$

Next, we select the scaling region. Let $r_{(5\%)}$ and $r_{(50\%)}$ denote the empirical 5th and 50th percentiles of $\mathcal{R}$. Then truncate to:

$$\mathcal{R}^* = \left\{ r \in \mathcal{R} \mid r_{(5\%)} < r < r_{(50\%)} \right\}$$

Now denote $n := |\mathcal{R}^*|$ and $r_{\min} := \min_{r \in \mathcal{R}^*} r$.

Following Clauset, Shalizi, and Newman (Clauset et al., 2009), we identify a power law fit using maximum likelihood estimation (MLE). Assume for $r \geq r_{\min}$ that the distances follow $p(r) = Z\, r^{-\alpha}$, where $Z$ is the normalizing constant. Then,

$$\hat{\alpha} = 1 + \frac{n}{\sum\limits_{r \in \mathcal{R}^*} \ln\left(\frac{r}{r_{\min}}\right)}$$

In the Grassberger–Procaccia method one examines a correlation integral with unbiased estimator:

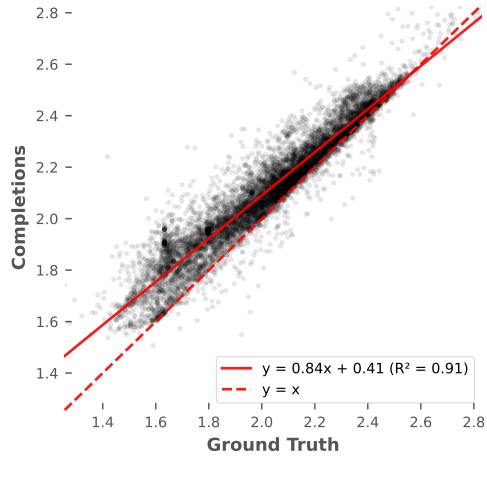

Figure 18: Correlation dimension comparison on held-out systems. Computed for ground truth and completions from *Panda MLM-66M* using the Grassberger-Procaccia method, averaged over 8 independent masks for each trajectory, using context length 4096 with half the patches (patch length 16) randomly masked out in a channel-inconsistent manner.

$$\mathcal{C}(r) = \frac{2}{T(T-1)} \sum_{i<j} H\big(r - \|x_i - x_j\|_2\big), \quad \mathcal{C}(r) \sim r^{D_2} \quad (r \to 0),$$

so that $D_2 = \frac{d \ln \mathcal{C}(r)}{d \ln r}$. Fitting $\mathcal{C}(r) \propto r^{D_2}$ is equivalent to fitting the distribution of pairwise distances to a power law, yielding $D_2 \approx \hat{\alpha}$ as the estimated correlation dimension.

*Panda MLM*, with *20M* parameters, shows promise in recovering the correlation dimension, a statistical invariant of the attractor, even when given much longer context (length 4096) than seen during traning (recall the context length for training was 512), and with half of the timesteps masked out (in patches) per dimension. We also trained a scaled-up checkpoint, *Panda MLM-66M* (with details in Appendix K), which demonstrates improved performance (Fig. 18).

In 10, we present the comparison against interpolation baselines. For polynomial interpolation, we use `numpy.polyfit` to fit a polynomial to the unmasked timesteps, and evaluate with `numpy.polyval` at masked positions. For linear interpolation, we use `scipy.interpolate.interp1d`, with extrapolation for timesteps outside the range of known values. For the piecewise cubic spline baseline, we use `scipy.interpolate.make_interp_spline` with $k = 3$ (cubic spline).

| Comparison with Baselines for Completions Task | |
|---|---|
| Method | $L_{context} = 4096$, with 50% Erasure (in patches) |
| **Panda MLM-66M** | **0.91** |
| Panda MLM | 0.78 |
| Piecewise Cubic Spline | 0.71 |
| Linear Interpolation | 0.61 |
| Polynomial Interpolation (Deg 3) | 0.21 |

Table 10: Coefficient of Determination ($R^2$) between the correlation dimension (via Grassberger-Procaccia) computed on the completions versus the full length 4096 ground truth trajectories. For each of our 9347 held-out test systems, we average across 8 random seeds, which determine the timestep masks for each trajectory (in patches of length 16). See Appendix N for further discussion.

# F   ADDITIONAL DISTRIBUTIONAL METRICS AND INVARIANT QUANTITIES

We compare long-term forecast metrics in Tables 11 and 12, we report mean $\pm$ std. dev. across all test systems, averaged over 5 context windows for prediction horizons $L_{\text{pred}} \in \{512, 1024, 2048, 3072\}$. Only one context window is available for evaluating $L_{\text{pred}} = 3584$, as our dataset contains trajectories of length 4096. Here, we include comparisons to *DynaMix* due to its strength in maintaining long term statistics during long rollouts, but note that differences in inference settings and our focus on shorter context lengths and rollouts compared to *DynaMix*, a fair comparison is not possible. We thus intend to primarily compare with other transformer-based foundation models.

$D_{KL}$ (**Ground Truth**$(L_{\text{pred}})$||**Model Prediction**$(L_{\text{pred}})$)

| Model | $L_{\text{pred}} = 512$ | $L_{\text{pred}} = 1024$ | $L_{\text{pred}} = 2048$ | $L_{\text{pred}} = 3072$ | $L_{\text{pred}} = 3584$ |
|---|---|---|---|---|---|
| **Panda** | $3.93 \pm 3.51$ | $4.72 \pm 3.64$ | $5.63 \pm 3.71$ | $6.14 \pm 3.68$ | $6.39 \pm 3.90$ |
| Chronos 20M SFT | $4.72 \pm 5.00$ | $5.09 \pm 4.90$ | $5.62 \pm 4.86$ | $5.93 \pm 4.84$ | $6.05 \pm 5.34$ |
| Chronos 20M | $5.99 \pm 5.07$ | $6.19 \pm 4.85$ | $6.51 \pm 4.76$ | $6.76 \pm 4.74$ | $6.94 \pm 5.41$ |
| Chronos 200M | $5.12 \pm 5.25$ | $5.49 \pm 5.22$ | $6.05 \pm 5.30$ | $6.36 \pm 5.28$ | $6.47 \pm 5.67$ |
| DynaMix | $4.75 \pm 5.70$ | $4.90 \pm 5.65$ | $5.22 \pm 5.72$ | $5.40 \pm 5.70$ | $5.51 \pm 6.13$ |

Table 11: KL divergence between ground truth and model predictions. See Table 13 for per-system differences, and Appendix F.1 for implementation details.

**Average** $H^2(S_{\text{Ground Truth }(L_{\text{pred}})}||S_{\text{Model Prediction }(L_{\text{pred}})})$

| Model | $L_{\text{pred}} = 512$ | $L_{\text{pred}} = 1024$ | $L_{\text{pred}} = 2048$ | $L_{\text{pred}} = 3072$ | $L_{\text{pred}} = 3584$ |
|---|---|---|---|---|---|
| **Panda** | $0.25 \pm 0.14$ | $0.25 \pm 0.12$ | $0.25 \pm 0.11$ | $0.25 \pm 0.11$ | $0.26 \pm 0.12$ |
| Chronos 20M SFT | $0.29 \pm 0.17$ | $0.29 \pm 0.16$ | $0.29 \pm 0.16$ | $0.30 \pm 0.16$ | $0.30 \pm 0.18$ |
| Chronos 20M | $0.37 \pm 0.16$ | $0.36 \pm 0.16$ | $0.37 \pm 0.16$ | $0.38 \pm 0.16$ | $0.38 \pm 0.17$ |
| Chronos 200M | $0.28 \pm 0.16$ | $0.28 \pm 0.15$ | $0.29 \pm 0.15$ | $0.30 \pm 0.15$ | $0.30 \pm 0.17$ |
| DynaMix | $0.36 \pm 0.19$ | $0.34 \pm 0.19$ | $0.33 \pm 0.19$ | $0.33 \pm 0.19$ | $0.32 \pm 0.21$ |

Table 12: Average per-dimension spectral Hellinger distance between ground truth and model predictions. See Table 14 for per-system differences. We use Welch's method for estimating the PSD. Note that the DynaMix comparison does not match that model's original experiment setting, see the note in Table 8

### F.1 IMPLEMENTATION DETAILS: KL DIVERGENCE VIA GMMS

Algorithm 1 presents our implementation of the Kullback Leibler divergence between ground truth and model predictions. This is the implementation we use for our main evaluations (Tables 11 and 13), although in Subsection F.3 we also present results using an alternative implementation found in the literature. In particular, we construct GMMs by fitting Gaussians to each point, with local scale parameter determined by the simplex neighbors algorithm.

---

**Algorithm 1** KL Divergence Estimation via Gaussian Mixture Models

---

**Require:** Ground Truth $\mathbf{X} = \{\mathbf{x}_t\}_{t=1}^T$, Generated Predictions $\mathbf{Y} = \{\mathbf{y}_t\}_{t=1}^T$, Number of Monte Carlo Samples $n_s$, small $\varepsilon > 0$ and $\mathtt{tol} > 0$

1: **Function** ESTIMATEKLDIVERGENCE($\mathbf{X}, \mathbf{Y}, n_s, \varepsilon$)
 **// Step 1: Local bandwidth (scale) estimation**
2: $\boldsymbol{\sigma}^X \leftarrow$ SIMPLEXNEIGHBORS($\mathbf{X}, k = 10$)
3: $\boldsymbol{\sigma}^Y \leftarrow$ SIMPLEXNEIGHBORS($\mathbf{Y}, k = 10$)
 **// Step 2: Construct Gaussian Mixture Models**
4: $p \leftarrow$ GAUSSIANMIXTURE(means $= \mathbf{X}$, covariances $= \boldsymbol{\sigma}^X$)
5: $q \leftarrow$ GAUSSIANMIXTURE(means $= \mathbf{Y}$, covariances $= \boldsymbol{\sigma}^Y$)
 **// Step 3: Monte Carlo KLD Estimate**
6: $\{\mathbf{z}_i\}_{i=1}^{n_s} \leftarrow p.\text{SAMPLE}(n_s)$
7: **for** $i = 1$ to $n_s$ **do**
8: $\quad p_i \leftarrow p(\mathbf{z}_i)$
9: $\quad q_i \leftarrow q(\mathbf{z}_i)$
10: $\quad q_i \leftarrow \max(q_i, \varepsilon)$
11: $\quad r_i \leftarrow \log(p_i/q_i)$
12: **end for**
13: **return** $\widehat{\text{KLD}} = \dfrac{1}{n_s} \sum_{i=1}^{n_s} r_i$

14: **Function** SIMPLEXNEIGHBORS($\mathbf{Z}, k$)
15: Let $\mathbf{Z} = \{\mathbf{z}_i\}_{i=1}^n$ with $\mathbf{z}_i \in \mathbb{R}^d$
16: Build a $(k+1)$-nearest neighbor search structure on $\mathbf{Z}$ (e.g., using Euclidean distance)
17: **for** $i = 1$ to $n$ **do**
18: $\quad$ Query $(k+1)$ nearest neighbors of $\mathbf{z}_i$, including itself
19: $\quad$ Discard the self-neighbor to obtain neighbors $\{\mathbf{z}_{i,j}\}_{j=1}^k$ with distances $d_{i,j}$
20: $\quad \sigma_i \leftarrow$ FINDSIGMA($(d_{i,1}, \ldots, d_{i,k}), k$) $\qquad\qquad$ ▷ Estimate local scale parameter
21: $\quad$ Let $\rho_i \leftarrow \min_j d_{i,j}$
22: **end for**
23: **return** $\boldsymbol{\sigma} = (\sigma_i)_{i=1}^n$

24: **Function** FINDSIGMA($\mathbf{d}, k$) $\qquad$ ▷ $\mathbf{d} = (d_1, \ldots, d_k)$ are distances to $k$ nearest neighbors
25: $\rho \leftarrow \min_j d_j$
26: Define $\Delta_j \leftarrow \max(d_j - \rho, 0)$ for $j = 1, \ldots, k$ $\qquad$ ▷ ReLU on shifted distances
27: Define objective

$$\phi(\sigma) = \left( \sum_{j=1}^k \exp(-\Delta_j/(\sigma + \mathtt{tol})) - \log_2(k) \right)^2, \quad \sigma > 0$$

28: Minimize $\phi(\sigma)$ using 1D optimization (e.g., root-finding in $\log \sigma$ with initial guess $\rho$).
29: Let $\sigma^*$ be the resulting positive solution
30: **return** $\sigma^*$

---

## F.2 Per-System Differences in Distributional Metrics

We report mean $\pm$ std. dev. of per-system differences across all test systems, averaged over 5 context windows for prediction horizons $L_{\text{pred}} \in \{512, 1024, 2048, 3072\}$.

| Per-system Difference in $D_{KL}$ (**Ground Truth**($L_{\text{pred}}$)\|\|**Model Prediction**($L_{\text{pred}}$)) between Baselines | | | | | |
|---|---|---|---|---|---|
| Comparison | $L_{\text{pred}} = 512$ | $L_{\text{pred}} = 1024$ | $L_{\text{pred}} = 2048$ | $L_{\text{pred}} = 3072$ | $L_{\text{pred}} = 3584$ |
| **Chronos 20M SFT – Panda** | $\mathbf{0.79} \pm 4.60$ | $\mathbf{0.36} \pm 5.03$ | $\mathbf{-0.01} \pm 5.14$ | $\mathbf{-0.22} \pm 5.09$ | $\mathbf{-0.33} \pm 5.84$ |
| DynaMix – Panda | $0.81 \pm 5.12$ | $0.18 \pm 5.58$ | $-0.42 \pm 5.59$ | $-0.75 \pm 5.52$ | $-0.88 \pm 6.07$ |
| Chronos 20M – Chronos 20M SFT | $1.27 \pm 4.64$ | $1.09 \pm 5.39$ | $0.89 \pm 5.50$ | $0.84 \pm 5.53$ | $0.89 \pm 6.62$ |
| Chronos 200M – Chronos 20M SFT | $0.39 \pm 4.35$ | $0.41 \pm 5.36$ | $0.43 \pm 5.61$ | $0.44 \pm 5.73$ | $0.42 \pm 6.59$ |

Table 13: The (mean $\pm$ std) of per-system diff. in KL divergence between models, a fine-grained view of Table 11. *DynaMix* and *Chronos 20M SFT* outperform *Panda* on very long prediction horizons.

| Per-System Difference in **Average** $H^2(S_{\text{Ground Truth }(L_{\text{pred}})}\|\|S_{\text{Model Prediction }(L_{\text{pred}})})$ between Baselines | | | | | |
|---|---|---|---|---|---|
| Comparison | $L_{\text{pred}} = 512$ | $L_{\text{pred}} = 1024$ | $L_{\text{pred}} = 2048$ | $L_{\text{pred}} = 3072$ | $L_{\text{pred}} = 3584$ |
| Chronos 20M SFT – **Panda** | $0.04 \pm 0.19$ | $0.04 \pm 0.18$ | $0.04 \pm 0.18$ | $0.04 \pm 0.18$ | $0.04 \pm 0.20$ |
| DynaMix – **Panda** | $0.11 \pm 0.22$ | $0.09 \pm 0.22$ | $0.08 \pm 0.21$ | $0.07 \pm 0.21$ | $0.07 \pm 0.24$ |
| Chronos 20M – Chronos 20M SFT | $0.08 \pm 0.20$ | $0.08 \pm 0.20$ | $0.08 \pm 0.20$ | $0.08 \pm 0.20$ | $0.08 \pm 0.23$ |
| Chronos 200M – Chronos 20M SFT | $0.00 \pm 0.20$ | $0.00 \pm 0.20$ | $0.00 \pm 0.20$ | $0.00 \pm 0.20$ | $0.00 \pm 0.23$ |

Table 14: The (mean $\pm$ std) of per-system differences in average spectral Hellinger distance between models, a fine-grained view of Table 12 showing that *Panda* outperforms the baselines.

## F.3 An Alternative KL Divergence Implementation (Geometric Misalignment)

In addition to our GMM-based KL divergence implementation (Tables 11 and 13), we also use the implementation of (Hemmer & Durstewitz, 2025) based on geometric misalignment.

| $D_{KL}$ (**Ground Truth**($L_{\text{pred}}$)\|\|**Model Prediction**($L_{\text{pred}}$)) via Geometric Misalignment | | | | | |
|---|---|---|---|---|---|
| Model | $L_{\text{pred}} = 512$ | $L_{\text{pred}} = 1024$ | $L_{\text{pred}} = 2048$ | $L_{\text{pred}} = 3072$ | $L_{\text{pred}} = 3584$ |
| Panda | $2.82 \pm 2.67$ | $3.29 \pm 2.79$ | $3.88 \pm 2.85$ | $4.26 \pm 2.88$ | $4.44 \pm 3.14$ |
| **Chronos 20M SFT** | $\mathbf{2.52 \pm 2.63}$ | $\mathbf{2.81 \pm 2.94}$ | $\mathbf{3.09 \pm 3.16}$ | $\mathbf{3.25 \pm 3.28}$ | $\mathbf{3.34 \pm 3.72}$ |
| Chronos 20M | $4.33 \pm 3.20$ | $4.67 \pm 3.53$ | $5.03 \pm 3.76$ | $5.24 \pm 3.88$ | $5.37 \pm 4.40$ |
| Chronos 200M | $2.96 \pm 2.86$ | $3.19 \pm 3.15$ | $3.47 \pm 3.37$ | $3.64 \pm 3.49$ | $3.73 \pm 4.01$ |
| DynaMix | $3.06 \pm 4.07$ | $3.15 \pm 4.42$ | $3.24 \pm 4.68$ | $3.30 \pm 4.81$ | $3.37 \pm 5.40$ |
| $\Delta\% (\uparrow)$ | $-11.9\%$ | $-17.1\%$ | $-25.6\%$ | $-31.1\%$ | $-32.9\%$ |

Table 15: KL divergence between the ground truth and model predictions. $\Delta\%$ denotes percentage gain of *Panda* over the best baseline. See Table 16 for per-system differences. Here, we use the implementation of (Hess et al., 2023). However, note that our experiment setting does not match the DynaMix's original experiment setting, see the note in Table 8

| Per-system Difference in $D_{KL}$ (**Ground Truth**($L_{\text{pred}}$)\|\|**Model Prediction**($L_{\text{pred}}$)) between Baselines | | | | | |
|---|---|---|---|---|---|
| Comparison | $L_{\text{pred}} = 512$ | $L_{\text{pred}} = 1024$ | $L_{\text{pred}} = 2048$ | $L_{\text{pred}} = 3072$ | $L_{\text{pred}} = 3584$ |
| **Chronos 20M SFT** – Panda | $-0.30 \pm 3.50$ | $-0.49 \pm 3.75$ | $-0.79 \pm 3.95$ | $-1.00 \pm 4.01$ | $-1.09 \pm 4.54$ |
| DynaMix – Panda | $0.23 \pm 4.71$ | $-0.14 \pm 5.06$ | $-0.64 \pm 5.25$ | $-0.96 \pm 5.38$ | $-1.07 \pm 6.01$ |
| Chronos 20M – Chronos 20M SFT | $1.82 \pm 3.60$ | $1.86 \pm 3.93$ | $1.94 \pm 4.25$ | $1.99 \pm 4.33$ | $2.03 \pm 5.14$ |
| Chronos 200M – Chronos 20M SFT | $0.44 \pm 3.33$ | $0.39 \pm 3.66$ | $0.38 \pm 4.01$ | $0.39 \pm 4.10$ | $0.38 \pm 4.88$ |

Table 16: The (mean $\pm$ std) of per-system diff. in KL divergence between models, a fine-grained view of Table 15. *DynaMix* and *Chronos 20M SFT* outperform *Panda* on very long prediction horizons.

## F.4 VISUALIZATION OF METRICS DISTRIBUTION ACROSS ALL TEST SYSTEMS

We visualize distributional metrics of different models in Figure 19 and 20, which quantify how well the forecasts align with the global invariant statistics of the underlying dynamical systems. We emphasize that a fair comparison to *Dynamix* is not entirely feasible due to the fact that it requires a much longer context length during inference and is only trained on 3 dimensional dynamical systems. We hypothesize that the latter fact explains the anomalous mode in Figure 19.

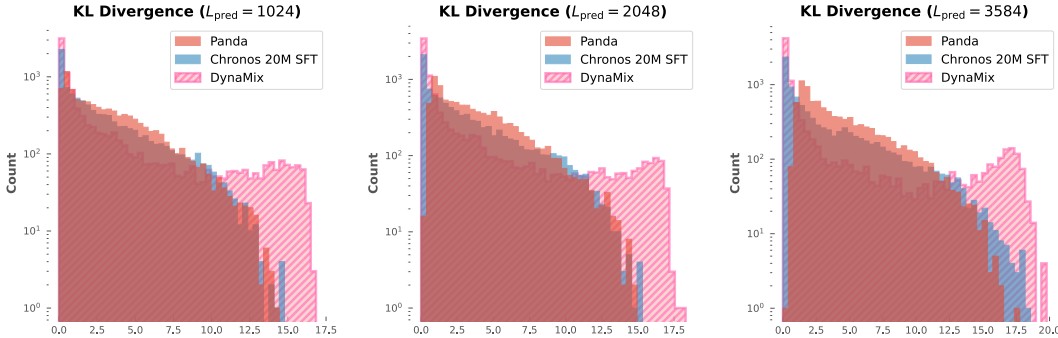

Figure 19: KL divergence (via geometric misalignment) between ground truth ($L_{\text{pred}}$) and model predictions ($L_{\text{pred}}$). See Table 15 for aggregate values. Note that $L_{\text{pred}} = 3584$ is 28x the prediction length used in training *Panda*.

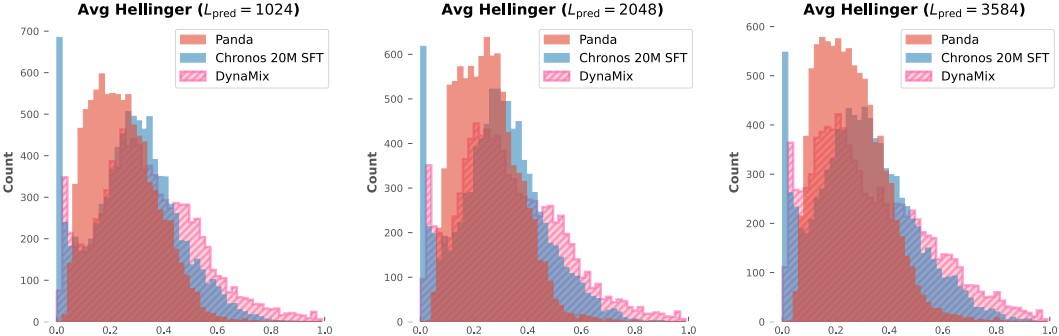

Figure 20: Average spectral Hellinger distance, between the power spectra of ground truth ($L_{\text{pred}}$) and model predictions ($L_{\text{pred}}$). See Table 12 for aggregate values.

## F.5 QUANTIFYING MEAN REGRESSION ON VERY LONG HORIZONS

Mean regression is a common failure mode for TSFMs on very long prediction horizons. To quantify this failure mode, we compute the distributional metrics at $L_{\text{pred}} = 3584$, which is the longest possible horizon for evaluation, since our dataset contains trajectories of 4096 timepoints. However, *we cut off* the first $N_{\text{cutoff}} = 1536$ timepoints of model predictions and ground truth, to compute the metrics on the last 2048 timepoints - solidly within the mean regression regime.

| Metrics on $t_{\text{pred}} = [1536, 3584]$ **(Cut Off First 1536 Timepoints)** | | |
|---|---|---|
| Model | KL Divergence (Geometric Misalignment) | Spectral Hellinger Distance |
| Panda | $15.25 \pm 2.46$ | $0.49 \pm 0.11$ |
| Chronos 20M SFT | $7.00 \pm 5.63$ | $\mathbf{0.34 \pm 0.18}$ |
| Chronos 20M | $9.68 \pm 6.12$ | $0.48 \pm 0.18$ |
| Chronos 200M | $7.37 \pm 6.05$ | $0.36 \pm 0.19$ |
| DynaMix | $\mathbf{3.50 \pm 5.44}$ | $0.35 \pm 0.22$ |

Table 17: Metrics between ground truth and model predictions *after cutting off* the first 1536 timepoints of $L_{\text{pred}} = 3584$ (keeping only the last 2048). We present (mean $\pm$ std) across test systems.

## F.6 MAXIMUM LYAPUNOV EXPONENT

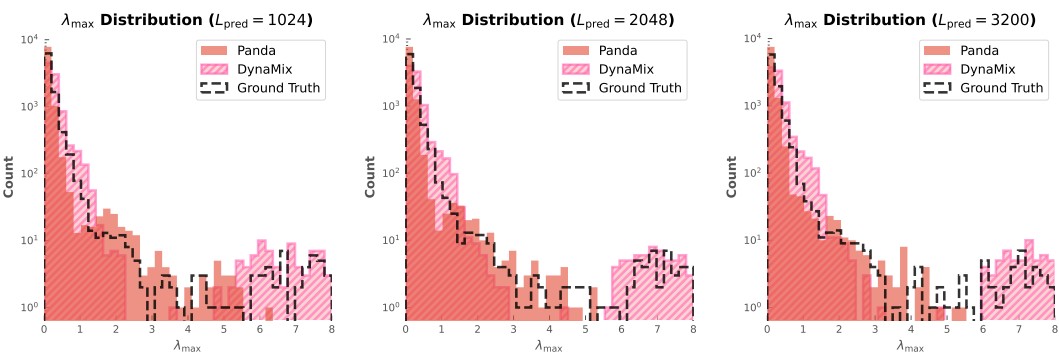

Figure 21: Distributional comparison of the maximum Lyapunov exponents estimated from the ground truth ($L_{\text{pred}}$) versus estimates from the model predictions ($L_{\text{pred}}$) of *Panda* and of *DynaMix*. Note that the y-axis is on a log scale.

We compute the maximum Lyapunov exponents for long prediction horizons, using the data-driven Rosenstein estimator. In Fig. 21, we compare the distribution of estimated ($\lambda_{\text{max}}$) for *Panda* versus that of *DynaMix* at prediction lengths $L_{\text{pred}} = 1024, 2048, 3200$. Note that *DynaMix* was trained pointwise autoregressively for a prediction horizon of 550 points. Despite *Panda* being trained non-autoregressively for a $4\times$ shorter prediction horizon, it is capable of producing forecasts which maintain the characteristic Lyapunov exponent out to $25\times$ the prediction horizon it was trained on.

We do observe that *Panda* struggles to capture systems with $\lambda_{max} > 6$ in Fig. 21. This is likely due to the failure mode of mean regression over long enough prediction horizons. As a recurrent model, DynaMix avoids this limitation at long horizons.

## G IMPLICIT SPATIO-TEMPORAL COUPLING

Temporal attention and channel attention layers independently mix information along the patch and channel dimensions. For a system like the Lorenz attractor with coupled phase coordinates $[x, y, z]$, we would ideally want information to mix across space *and* time. We will show that by composing temporal and channel attention in sequence, *Panda* implicitly performs spatio-temporal coupling.

Let $W_Q, W_K, W_V$ denote the learned projections for temporal attention and $\overline{W}_Q, \overline{W}_K, \overline{W}_V$ for channel attention. For simplicity, we will focus on the linear attention setting without the row-wise softmax. Let $P \in \mathbb{R}^{N \times C \times d_{\text{model}}}$ be a stack of $N$, $d_{\text{model}}$-dimensional patch embeddings with $C$ channels, and $\mathsf{p}_i^{(c)} \in \mathbb{R}^{d_{\text{model}}}$ an individual patch embedding for patch $i$ and channel $c$. The linear attention output is $(PW_Q W_K^\top P) PW_V$. In vector form,

$$(\textbf{TA}): \phi_i^{(\cdot)} = \sum_{j=1}^{T} \left\langle W_Q^\top \mathsf{p}_i^{(\cdot)}, W_K^\top \mathsf{p}_j^{(\cdot)} \right\rangle W_V^\top \mathsf{p}_j^{(\cdot)} = \sum_{j=1}^{T} \left\langle \mathsf{p}_i^{(\cdot)}, A_{\textbf{TA}}, \mathsf{p}_j^{(\cdot)} \right\rangle W_V^\top \mathsf{p}_j^{(\cdot)} \qquad (3)$$

$$(\mathbf{CA}) : \overline{\phi}_i^{(k)} = \sum_{\ell=1}^{c} \left\langle \overline{W}_Q^\top \phi_i^{(k)}, \overline{W}_K^\top \phi_i^{(\ell)} \right\rangle \overline{W}_V^\top \phi_i^{(\ell)} = \sum_{\ell=1}^{c} \underbrace{\left\langle \phi_i^{(k)}, A_{\mathbf{CA}}, \phi_i^{(\ell)} \right\rangle}_{M_i^{k\ell}} \overline{W}_V^\top \phi_i^{(\ell)} \tag{4}$$

Where **TA** denotes temporal attention and **CA** channel attention, and $A_{\mathbf{TA}} \coloneqq W_Q W_K^\top$ and $A_{\mathbf{CA}} \coloneqq \overline{W}_Q \overline{W}_K^\top$. Looking at an element of the 3-tensor $M_i^{k\ell}$ we see that:

$$M_{k\ell} = \left\langle \sum_{j=1}^{T} \left\langle \mathsf{p}_i^{(k)}, A_{\mathbf{TA}} \mathsf{p}_j^{(k)} \right\rangle W_V^\top \mathsf{p}_j^{(k)}, A_{\mathbf{CA}} \sum_{j'=1}^{T} \left\langle \mathsf{p}_i^{(\ell)}, A_{\mathbf{TA}} \mathsf{p}_{j'}^{(\ell)} \right\rangle W_V^\top \mathsf{p}_{j'}^{(\ell)} \right\rangle \tag{5}$$

$$= \sum_{j,j'=1}^{T} \left\langle \left\langle \mathsf{p}_i^{(k)}, A_{\mathbf{TA}} \mathsf{p}_j^{(k)} \right\rangle W_V^\top \mathsf{p}_j^{(k)}, \left\langle \mathsf{p}_i^{(\ell)}, A_{\mathbf{TA}} \mathsf{p}_{j'}^{(\ell)} \right\rangle A_{\mathbf{CA}} W_V^\top \mathsf{p}_{j'}^{(\ell)} \right\rangle \tag{6}$$

$$= \sum_{j,j'=1}^{T} \left\langle \mathsf{p}_i^{(k)}, A_{\mathbf{TA}} \mathsf{p}_j^{(k)} \right\rangle \left\langle \mathsf{p}_i^{(\ell)}, A_{\mathbf{TA}} \mathsf{p}_{j'}^{(\ell)} \right\rangle \left\langle \mathsf{p}_j^{(k)}, \underbrace{\left( W_V A_{\mathbf{CA}} W_V^T \right)}_{\tilde{A}_{\mathbf{CA}}} \mathsf{p}_{j'}^{(\ell)} \right\rangle \tag{7}$$

Where $\tilde{A}_{\mathbf{CA}}$ prescribes how patches from different channels attend to each other. In matrix form,

$$M_i^{k\ell} = (\mathsf{p}_{\mathbf{TA}}^{(k)})^\top \underbrace{\begin{bmatrix} \left\langle \mathsf{p}_1^{(k)}, \tilde{A}_{\mathbf{CA}} \mathsf{p}_1^{(\ell)} \right\rangle & \cdots & \left\langle \mathsf{p}_1^{(k)}, \tilde{A}_{\mathbf{CA}} \mathsf{p}_T^{(\ell)} \right\rangle \\ \vdots & \ddots & \vdots \\ \left\langle \mathsf{p}_T^{(k)}, \tilde{A}_{\mathbf{CA}} \mathsf{p}_1^{(\ell)} \right\rangle & \cdots & \left\langle \mathsf{p}_T^{(k)}, \tilde{A}_{\mathbf{CA}} \mathsf{p}_T^{(\ell)} \right\rangle \end{bmatrix}}_{\text{Cross-Channel Mixing Map}} \mathsf{p}_{\mathbf{TA}}^{(\ell)}, \quad \mathsf{p}_{\mathbf{TA}}^{(k)} \coloneqq \begin{bmatrix} \left\langle \mathsf{p}_i^{(k)}, A_{\mathbf{TA}} \mathsf{p}_1^{(k)} \right\rangle \\ \vdots \\ \left\langle \mathsf{p}_i^{(k)}, A_{\mathbf{TA}} \mathsf{p}_T^{(k)} \right\rangle \end{bmatrix}$$

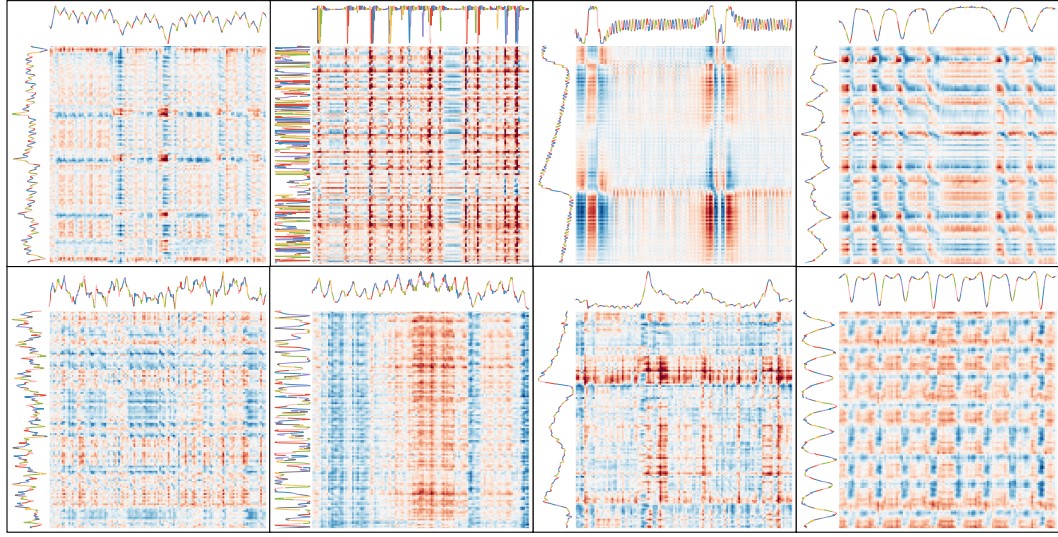

Figure 22: Cross-channel mixing maps across patches for different channels from different held-out systems. Each mixing map is max-scaled to the range $[-1, 1]$.

## H  FORECASTS ON PDEs

Other than foundation models, we include a Fourier neural operator (FNO) baseline that was trained using the `neuraloperator` framework (Kossaifi et al., 2024; Kovachki et al., 2021), and a DeepONet baseline trained with the `deepxde` framework (Lu et al., 2021). Unless otherwise specified, experiment parameters follow the default values in these libraries. For both operator learning baselines, we tune the parameters for each PDE and use the best training checkpoint according to the validation loss. Both operator learning baselines are trained (on a single MI100X GPU) for one-step-ahead prediction on length 512 context windows and rolled out for 512 prediction steps for each evaluation window in Fig. 6c.

For the Kuramoto-Shivashinsky (KS) PDE, we integrate the equations pseudospectrally with 64 Fourier modes and the spatial length parameter $L = 100$. We use an explicit eighth-order Dormand-Prince scheme (DOP853) to integrate the discretized PDE with a relative and absolute tolerance of 1e-8 from $t = 0$ to $t = 100$ save the trajectory at 4096 uniformly spaced timepoints. We sample 40 initial conditions from $u_i \sim \mathcal{N}(0, \varepsilon^2 \mathbb{I}_{64 \times 64})$ where we choose $\varepsilon = 0.1$ and use the length 512 context window starting at the 1024-th timepoint for training and the following 512 for prediction/rollout for each sample to produce the error bars in Fig. 6c. See Tables 18 for comprehensive details.

For the Von-Karman vortex street (VKVS) data, we use 4600 timepoints of velocity field data in the domain $\Omega = [0, 2] \times [0, 1]$ on a $256 \times 128$ grid simulated via a Lattice Boltzmann simulation at a Reynolds number of 450. We then compute the vorticity field via second-order finite difference and reduce the dimensionality by keeping the top 512 principal components. For evaluation in Fig. 6c, we train on length 512 training context windows starting at the time indices $\{0, 1024, 2048, 3072\}$ and cross-validate on the length 512 prediction windows starting at the time indices $\{512, 1536, 2560, 3584\}$ (avoiding train-set leakage) to produce the error bars in Fig. 6c. See Tables 19 for comprehensive details on the operator learning baselines.

| Component | Specification |
|---|---|
| Model | Fourier Neural Operator (FNO) |
| Modes | 256 |
| Hidden Channels | 256 |
| Layers | 6 |
| Activation | GELU |
| Optimizer | AdamW |
| Learning Rate | $1 \times 10^{-3}$ |
| LR Scheduler | Cosine decay |
| Epochs | 5000 |
| Batch Size | 512 |
| Loss Function | $L_2$ |

(a) FNO configuration

| Component | Specification |
|---|---|
| Model | DeepONet |
| Branch Net | $[128, 256 \times 6]$ |
| Trunk Net | $[1, 256 \times 6]$ |
| Activation | tanh |
| Initializer | He normal |
| Optimizer | AdamW |
| Learning Rate | $1 \times 10^{-3}$ |
| LR Scheduler | Cosine decay |
| Iterations | $2 \times 10^6$ |
| Batch Size | 512 |
| Metric | Mean relative $L^2$ error |

(b) DeepONet configuration

Table 18: Kuramoto-Shivashinsky PDE Operator Learning Configurations.

| Component | Specification |
|---|---|
| Model | Fourier Neural Operator (FNO) |
| Modes | 512 |
| Hidden Channels | 256 |
| Layers | 5 |
| Activation | GELU |
| Optimizer | AdamW |
| Learning Rate | $1 \times 10^{-3}$ |
| LR Scheduler | Cosine decay |
| Epochs | 5000 |
| Batch Size | 512 |
| Loss Function | $L_2$ |

(a) FNO configuration

| Component | Specification |
|---|---|
| Model | DeepONet |
| Branch Net | $[512, 512 \times 5]$ |
| Trunk Net | $[1, 512 \times 5]$ |
| Activation | tanh |
| Initializer | He normal |
| Optimizer | AdamW |
| Learning Rate | $1 \times 10^{-3}$ |
| LR Scheduler | Cosine decay |
| Iterations | $1 \times 10^6$ |
| Batch Size | 512 |
| Metric | Mean relative $L^2$ error |

(b) DeepONet configuration

Table 19: Von-Karman PDE Operator Learning Configurations.

Compared to the foundation models, the operator learning baselines under-perform mostly since they are limited to a context and prediction length of 1 for one-step-ahead prediction in contrast to the much larger context and prediction lengths of foundation models. We do not claim that foundation models are superior operator learning methods, but merely aim to provide a baseline for the PDE problems. The dash-dotted lines in Fig. 6 indicate that these methods are not directly comparable.

## I  COMPUTING AND HARDWARE REQUIREMENTS

All training runs were conducted on 4× AMD MI100X GPUs, each with 32 GB of HBM2 memory. Inference was performed on a single AMD MI100X GPU.

## J  EFFECT OF PATCH LENGTH

To investigate the effect of patch length on our model's performance, we conduct an ablation study in which we train a version of our model with various patch lengths. To isolate the effect of patch size, we **remove** our dynamics embedding for these ablations. This is because each patch gets embedded to dimension $d_{\text{model}}$, making the dynamics embedding incomparable between models using different patch lengths. Keeping a fixed compute budget, we also must halve the batch size every time we halve the patch length, as a tradeoff patch length $\propto 1/\text{batch size}$ exists between the two quantities: half the patch length implies twice as many patches, all embedded to dimension $d_{\text{model}}$. In the tables below, $\Delta\%$ denotes percentage improvement of the best ablation over the next closest.

| **sMAPE Median [P25, P75]** | | | |
|---|---|---|---|
| Model | $L_{\text{pred}} = 128$ | $L_{\text{pred}} = 256$ | $L_{\text{pred}} = 512$ |
| Patch 4 | **26.6** [17.5, 37.4] | **36.2** [26.5, 47.5] | 48.1 [39.0, 58.4] |
| Patch 8 | 28.6 [19.6, 40.2] | 37.7 [27.7, 48.8] | 48.2 [39.9, 58.7] |
| Patch 12 | 28.9 [19.4, 40.6] | 37.8 [27.9, 49.5] | 47.9 [39.6, 59.0] |
| Patch 16 | 29.1 [19.7, 41.1] | 37.6 [27.8, 49.4] | 47.7 [38.5, 58.7] |
| Patch 24 | 30.1 [20.6, 41.4] | 37.9 [27.9, 49.5] | 47.0 [38.7, 58.3] |
| Patch 32 | 30.1 [20.3, 42.2] | 37.7 [28.2, 49.3] | **46.6** [38.1, 57.4] |
| $\Delta\% (\uparrow)$ | +7.0% | +3.7% | +0.9% |

Table 20: Median sMAPE and interquartile range [P25, P75] for various patch lengths.

| **MAE Median [P25, P75]** | | | |
|---|---|---|---|
| Model | $L_{\text{pred}} = 128$ | $L_{\text{pred}} = 256$ | $L_{\text{pred}} = 512$ |
| Patch 4 | **0.321** [0.198, 0.509] | **0.472** [0.329, 0.685] | **0.642** [0.481, 0.824] |
| Patch 8 | 0.359 [0.224, 0.543] | 0.498 [0.359, 0.710] | 0.668 [0.502, 0.860] |
| Patch 12 | 0.364 [0.234, 0.559] | 0.501 [0.366, 0.711] | 0.662 [0.490, 0.859] |
| Patch 16 | 0.380 [0.246, 0.561] | 0.507 [0.365, 0.724] | 0.656 [0.486, 0.846] |
| Patch 24 | 0.382 [0.245, 0.578] | 0.519 [0.375, 0.729] | 0.668 [0.495, 0.857] |
| Patch 32 | 0.382 [0.248, 0.584] | 0.519 [0.376, 0.722] | 0.664 [0.492, 0.852] |
| $\Delta\% (\uparrow)$ | +10.7% | +5.2% | +2.1% |

Table 21: Median MAE and interquartile range [P25, P75] for various patch lengths.

| $1 - \rho$ **(Spearman distance) Median [P25, P75]** | | | |
|---|---|---|---|
| Model | $L_{\text{pred}} = 128$ | $L_{\text{pred}} = 256$ | $L_{\text{pred}} = 512$ |
| Patch 4 | **0.219** [0.117, 0.349] | **0.322** [0.197, 0.464] | **0.480** [0.326, 0.612] |
| Patch 8 | 0.244 [0.138, 0.376] | 0.350 [0.229, 0.485] | 0.495 [0.350, 0.637] |
| Patch 12 | 0.246 [0.139, 0.384] | 0.357 [0.225, 0.507] | 0.496 [0.338, 0.640] |
| Patch 16 | 0.261 [0.138, 0.412] | 0.360 [0.221, 0.504] | 0.486 [0.341, 0.644] |
| Patch 24 | 0.264 [0.150, 0.408] | 0.361 [0.235, 0.512] | 0.492 [0.348, 0.643] |
| Patch 32 | 0.268 [0.150, 0.413] | 0.364 [0.230, 0.513] | 0.485 [0.342, 0.644] |
| $\Delta\% (\uparrow)$ | +10.2% | +8.0% | +1.0% |

Table 22: Median $1 - \rho$ and interquartile range [P25, P75] for various patch lengths.

## K    SCALING UP

We scale up our model parameters and training to investigate the improvement in performance. Our 21M, 42M, and 72M parameter models have values of $(n_{\text{heads}}, n_{\text{layers}}, d_{\text{model}})$ set to (8, 8, 512), (10, 10, 640), and (12, 12, 768) respectively. For the scaled-up training, we had $(N_{iters}$, batch size per device, number of GPUs) set to (400K, 1024, 4), (400K, 512, 6), and (800K, 384, 6) respectively.

We also scaled up our MLM checkpoint to create *Panda MLM-66M* with $(n_{\text{heads}}, n_{\text{layers}}, d_{\text{model}})$ set to (12, 12, 768), trained for 800K iterations, with batch size 192, and on 6 GPUs.

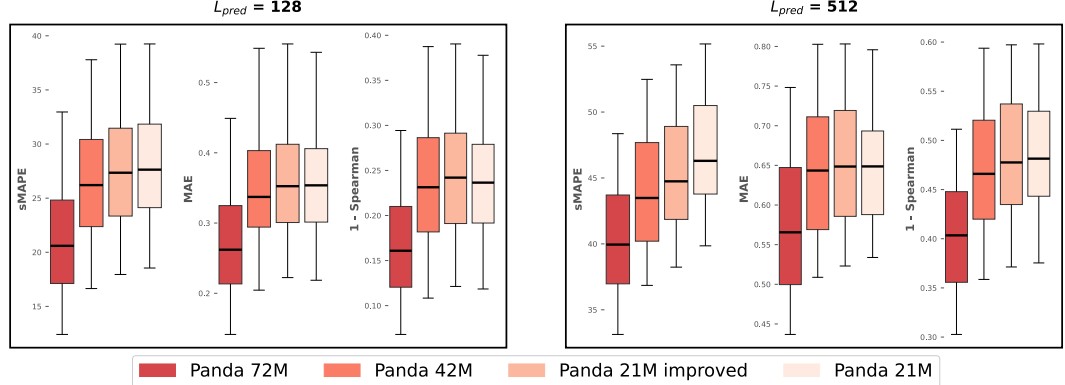

Figure 23: Zero-shot metrics for scaled-up checkpoints of *Panda* with increasing number of parameters. Here, Panda 21M is our original model presented in the main body. For Panda 21M *improved*, we trained for 400K iterations (compared to the 100K iterations for our original Panda 21M), and on an improved dataset with $\approx 20\%$ more systems, which we also use for the Panda 42M training. For Panda 72M, we trained on a larger version of our improved dataset with 8 initial conditions per system and with mixed periods. For presentation, bars show a semi-interquartile range (40th to 60th percentile); for numerical values of medians and interquartile ranges, see Tables 23, 24, 25.

### K.1    SCALED-UP BASELINES

We also scaled up the model parameters for Chronos as well as the training for the Chronos SFT baseline. Hardware limitations prevented us from fine-tuning Chronos 200M and larger model classes. We observe that our model continues to beat the baselines.

For our scaled-up training of Chronos 46M SFT, we used ($N_{iters} = 400K$, batch size per device = 100, number of GPUs = 6).

For Panda 72M and for Panda MLM-66M, we trained on a larger dataset with 8 initial conditions per system and with mixed periods.

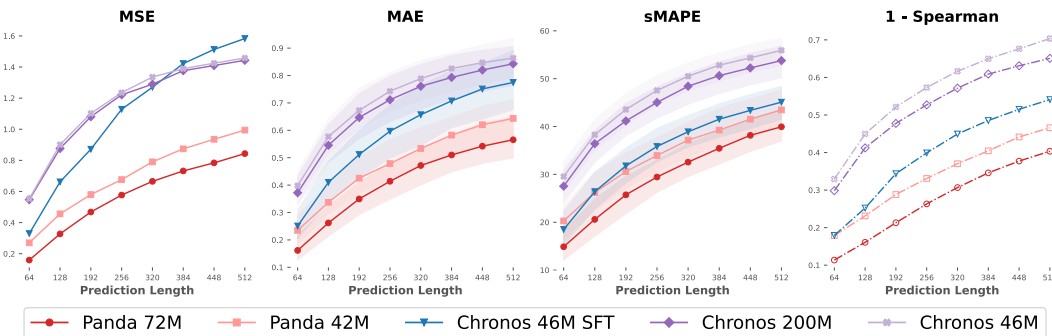

Figure 26: Zero-shot forecast metrics for scaled-up baselines, using *probabilistic* forecasts for Chronos and Chronos SFT. Dash-dotted lines indicate presence of NaNs for some systems (4% of systems for Spearman).

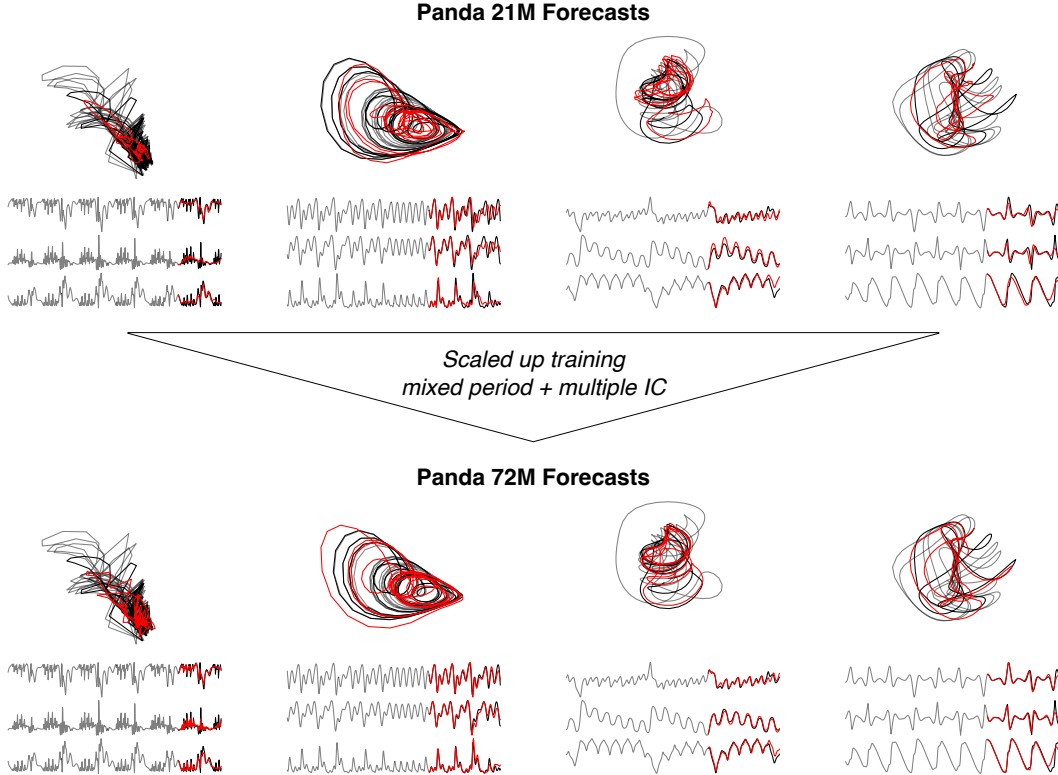

Figure 24: Comparison of sample zero-shot forecasts between the Panda 21M model (8 heads, 8 layers), and the Panda 72M model (12 heads, 12 layers), with the latter trained on a larger dataset with 8 initial conditions and mixed periods. As reflected in the metrics of Fig. 23, the scaled-up model forecasts appear to decrease error and capture higher-frequency details.

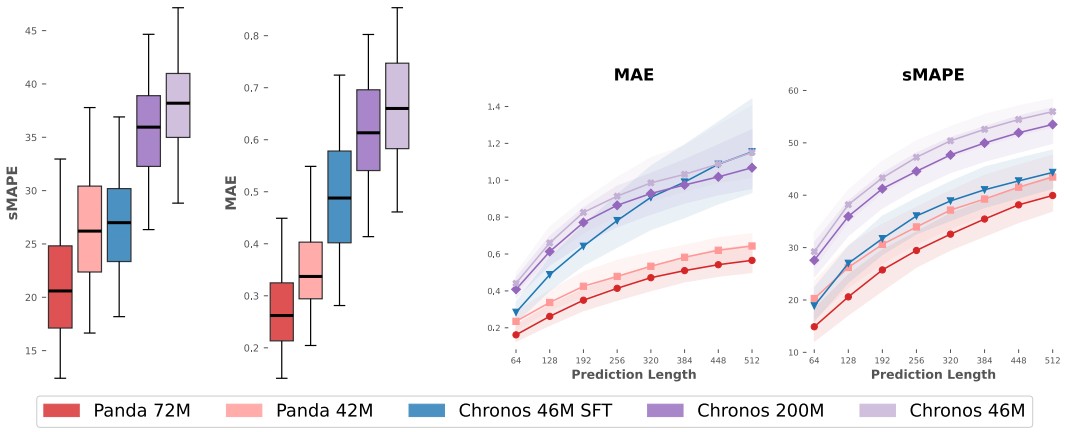

Figure 25: Zero-shot forecast metrics for scaled-up baselines, using deterministic forecasts for Chronos and Chronos SFT. For Panda 72M, we trained on a larger dataset with 8 initial conditions per system and with mixed periods. Spearman correlation is not shown because of the high proportion of NaNs for the Chronos deterministic forecasts, which we attribute to mean regression.

We present the metrics shown in Fig. 25 and Fig. 26 in tabular form in Tables 23, 24, 25.

### sMAPE Median [P25, P75]

| Model | $L_{\text{pred}} = 128$ | $L_{\text{pred}} = 256$ | $L_{\text{pred}} = 512$ |
|---|---|---|---|
| Panda 72M | **20.6** [12.4, 32.9] | **29.4** [20.0, 41.4] | **39.9** [30.9, 50.6] |
| Panda 42M | 26.2 [16.6, 37.8] | 33.9 [24.4, 45.7] | 43.5 [35.0, 54.9] |
| Chronos 46M SFT | 27.0 [18.2, 36.9] | 36.0 [26.2, 45.5] | 44.4 [32.7, 54.3] |
| Chronos 200M | 36.0 [26.3, 44.7] | 44.6 [34.6, 52.9] | 53.5 [42.7, 60.8] |
| Chronos 46M | 38.2 [28.8, 47.1] | 47.2 [38.6, 54.6] | 56.0 [46.9, 61.8] |
| Chronos 46M SFT Probabilistic | 26.4 [18.0, 37.3] | 35.8 [26.0, 46.0] | 45.1 [33.8, 54.1] |
| Chronos 200M Probabilistic | 36.4 [26.7, 44.7] | 45.0 [34.2, 53.1] | 53.8 [42.8, 60.6] |
| Chronos 46M Probabilistic | 38.3 [28.6, 46.6] | 47.5 [37.9, 54.7] | 55.9 [47.2, 61.6] |

Table 23: Median sMAPE and interquartile range [P25, P75] for scaled-up models.

### MAE Median [P25, P75]

| Model | $L_{\text{pred}} = 128$ | $L_{\text{pred}} = 256$ | $L_{\text{pred}} = 512$ |
|---|---|---|---|
| Panda 72M | **0.26** [0.14, 0.45] | **0.41** [0.26, 0.60] | **0.57** [0.41, 0.78] |
| Panda 42M | 0.34 [0.20, 0.55] | 0.48 [0.33, 0.71] | 0.64 [0.46, 0.85] |
| Chronos 46M SFT | 0.49 [0.28, 0.72] | 0.78 [0.45, 1.17] | 1.15 [0.61, 1.95] |
| Chronos 200M | 0.61 [0.41, 0.80] | 0.86 [0.58, 1.18] | 1.07 [0.75, 1.83] |
| Chronos 46M | 0.66 [0.46, 0.85] | 0.91 [0.63, 1.23] | 1.15 [0.81, 1.95] |
| Chronos 46M SFT Probabilistic | 0.41 [0.25, 0.64] | 0.60 [0.37, 0.89] | 0.77 [0.50, 1.21] |
| Chronos 200M Probabilistic | 0.55 [0.38, 0.73] | 0.71 [0.51, 0.91] | 0.84 [0.63, 1.12] |
| Chronos 46M Probabilistic | 0.58 [0.41, 0.75] | 0.74 [0.54, 0.93] | 0.86 [0.69, 1.17] |

Table 24: Median MAE and interquartile range [P25, P75] for scaled-up models.

### $1 - \rho$ (Spearman distance) Median [P25, P75]

| Model | $L_{\text{pred}} = 128$ | $L_{\text{pred}} = 256$ | $L_{\text{pred}} = 512$ |
|---|---|---|---|
| Panda 72M | **0.16** [0.07, 0.29] | **0.26** [0.14, 0.40] | **0.40** [0.26, 0.56] |
| Panda 42M | 0.23 [0.11, 0.39] | 0.33 [0.20, 0.48] | 0.47 [0.32, 0.62] |
| Chronos 46M SFT Probabilistic | 0.25 [0.14, 0.40] | 0.40 [0.25, 0.54] | 0.54 [0.36, 0.68] |
| Chronos 200M Probabilistic | 0.41 [0.25, 0.55] | 0.53 [0.35, 0.67] | 0.65 [0.48, 0.79] |
| Chronos 46M Probabilistic | 0.45 [0.28, 0.60] | 0.57 [0.40, 0.71] | 0.70 [0.55, 0.82] |

Table 25: Median $1 - \rho$ and interquartile range [P25, P75] for scaled-up models.

## K.2 DATASET WITH MULTIPLE INITIAL CONDITIONS AND MIXED PERIODS

For our scaled-up training, we used larger dataset of multiple initial conditions and mixed periods. We present a sample of this dataset in Fig. 27. We vary the number of periods (on Fourier timescale), from 20 to 100 to produce multiple periods, and carry out the numerical integration with up to 16 different initial conditions (although we only use 8 initial conditions per system for training, due to compute budget restrictions). We integrate the same set of $2 \times 10^4$ systems used in our training set. The scaled-up training thus allows us to assess the effect of varying the timescales present in our training data.

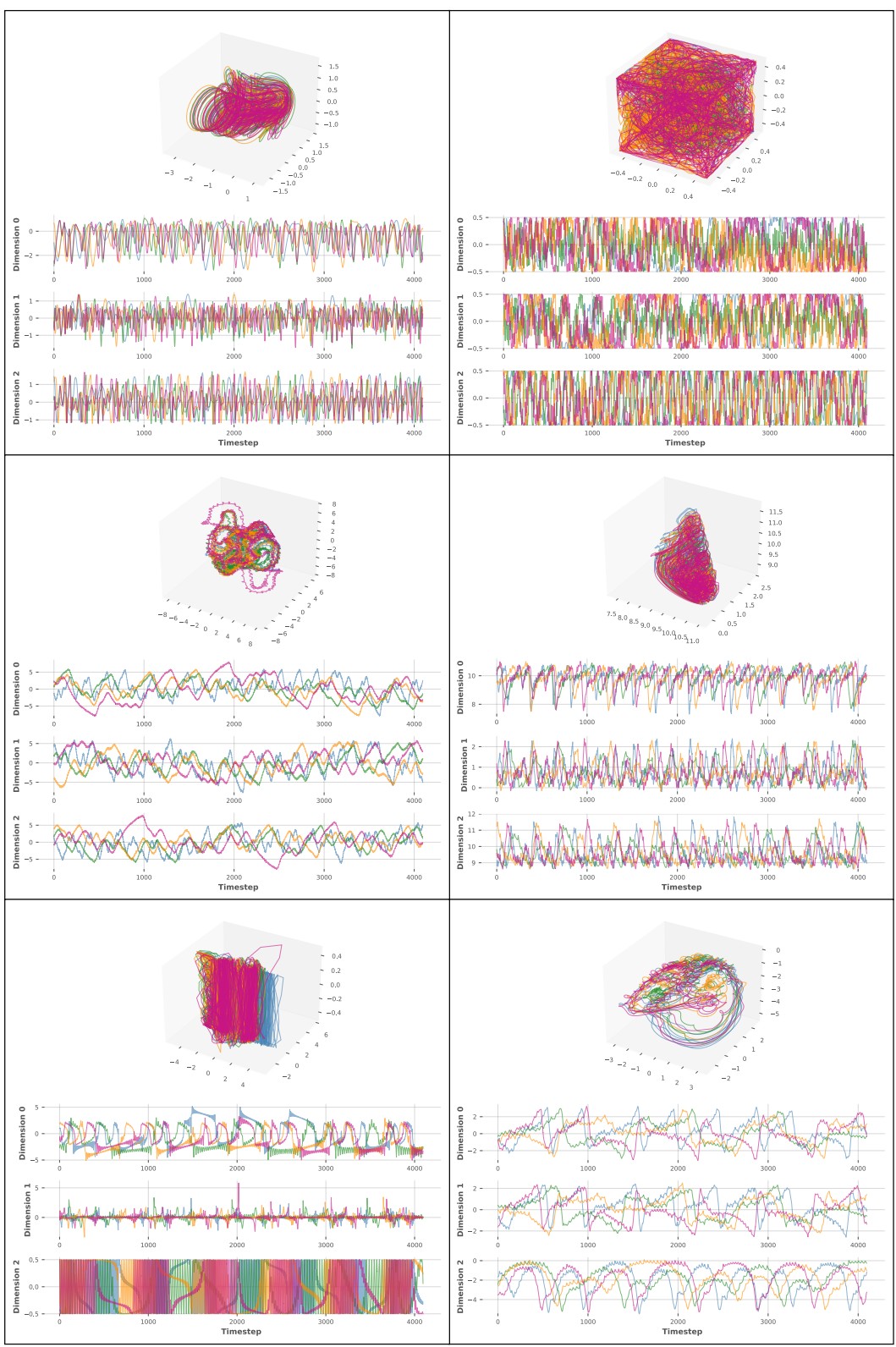

Figure 27: Examples of systems from our mixed period multi-IC training dataset. Each subplot shows multiple (4) initial conditions for a single system (integrated with different timescale).

## L    ADDITIONAL FORECASTS

In Appendix C, we presented a sample of forecasts from *Panda* on our held-out test set. Here, we provide more forecasts. As done previously, we keep the prediction length fixed at $L_{pred} = 256$ for consistency and clearer visibility. Our model was trained with $L_{pred} = 128$, so these forecasts include an autoregressive rollout. Fig. 30 presents more forecasts, and Appendix Section M presents failure modes and comparison against baseline models.

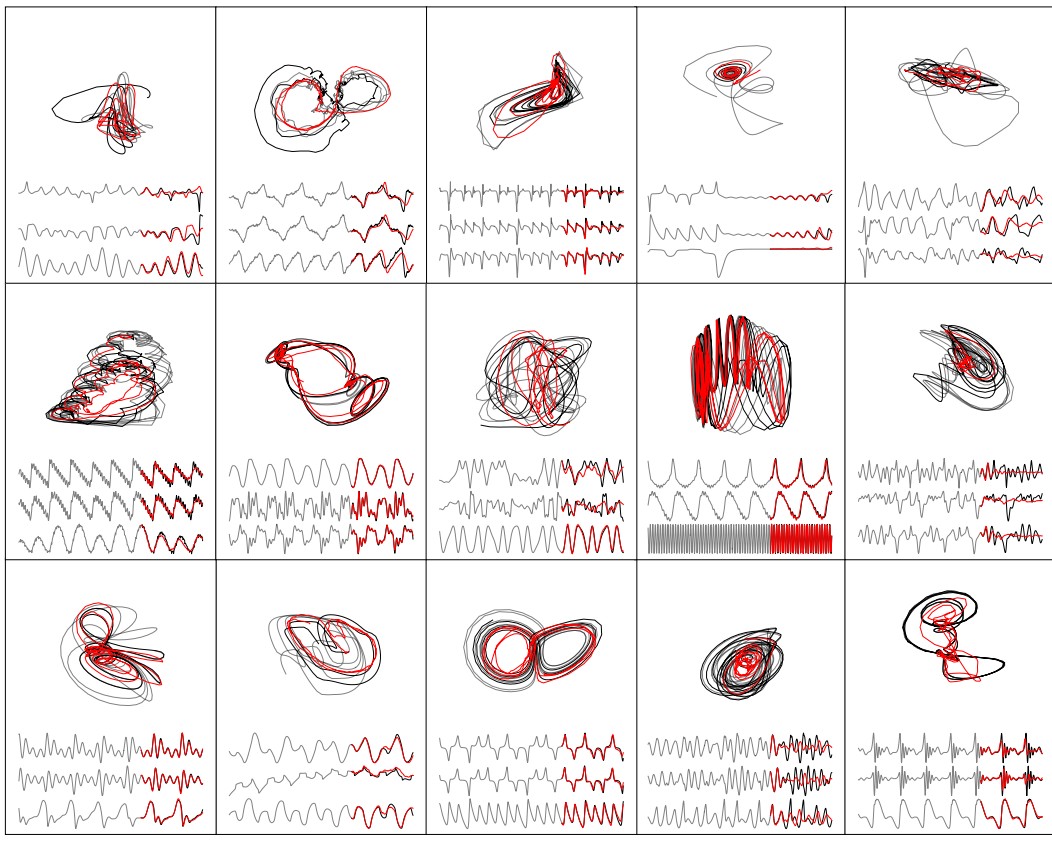

Figure 28: Examples of zero-shot forecasts ($L_{pred} = 256$) on held-out chaotic dynamical systems.

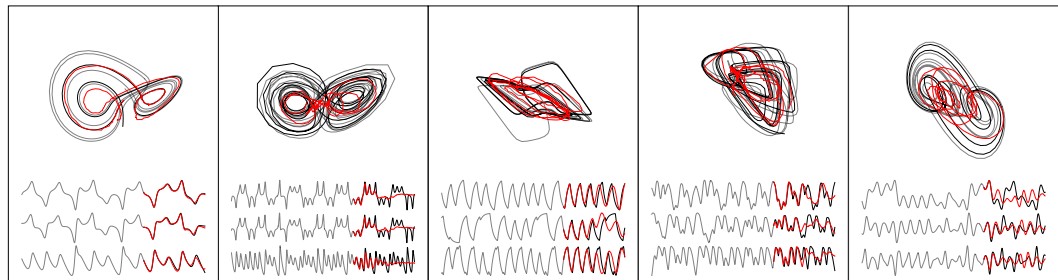

Figure 29: Examples of zero-shot forecasts ($L_{pred} = 256$) on held-out base systems (from the founder pool, parents of the skew-product systems).

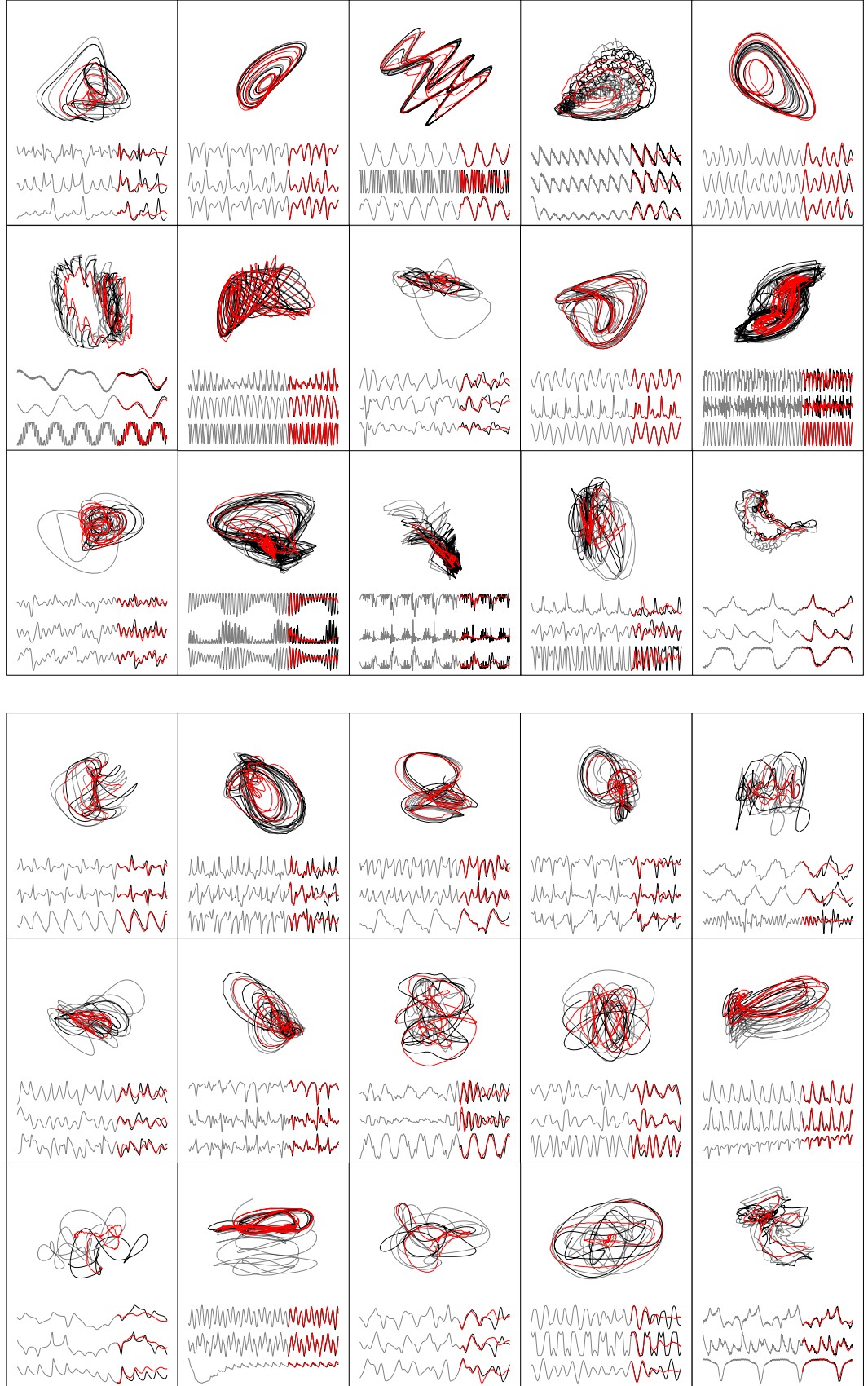

Figure 30: Examples of zero-shot forecasts ($L_{\text{pred}} = 256$) on held-out chaotic dynamical systems.

# M COMPARISON WITH BASELINE MODEL FORECASTS AND FAILURE MODES

We compare long-term ($L_{\text{pred}} = 512$) forecasts between *Panda* and the *Chronos SFT* and *Chronos* baselines. The following plots highlight some failure modes of each model, and also emphasize the advantage of our multivariate approach. Clearly, a univariate model can do well on a single channel (dimension) but fail to respect the attractor geometry. The coupling between channels encodes important information.

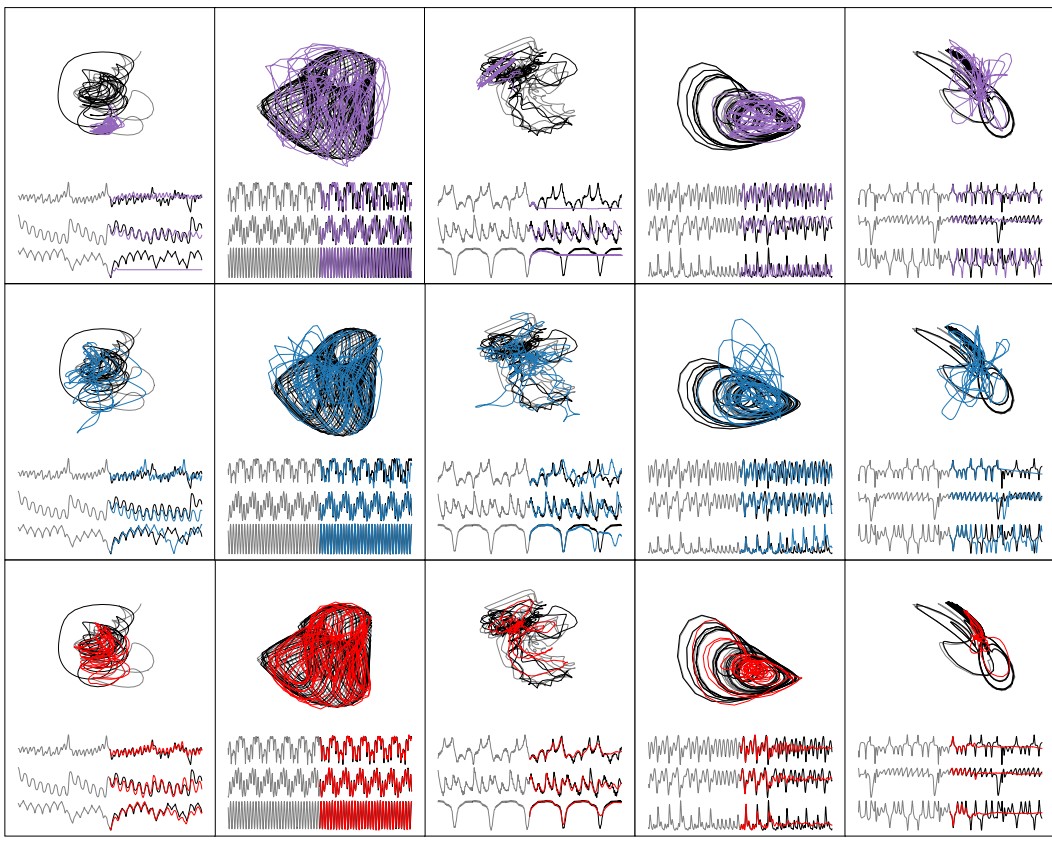

Figure 31: Long-term zero-shot forecasts ($L_{\text{pred}} = 512$) on held-out chaotic dynamical systems. Comparison between *Panda* (Red), *Chronos SFT* (Blue), and *Chronos* (Purple).

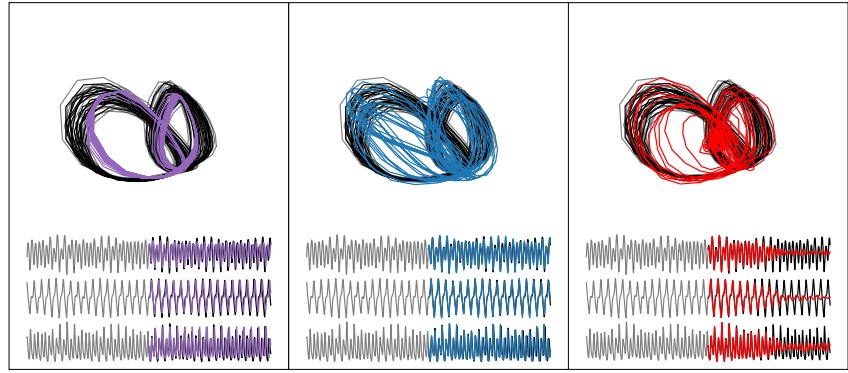

Figure 32: Comparison ($L_{\text{pred}} = 512$) between *Panda* (Red), *Chronos SFT* (Blue), and *Chronos* (Purple). An illustrative example of a held-out system where *Chronos* appears to parrot (limit cycle), *Chronos SFT* does not respect the attractor geometry, and *Panda* mean regresses.

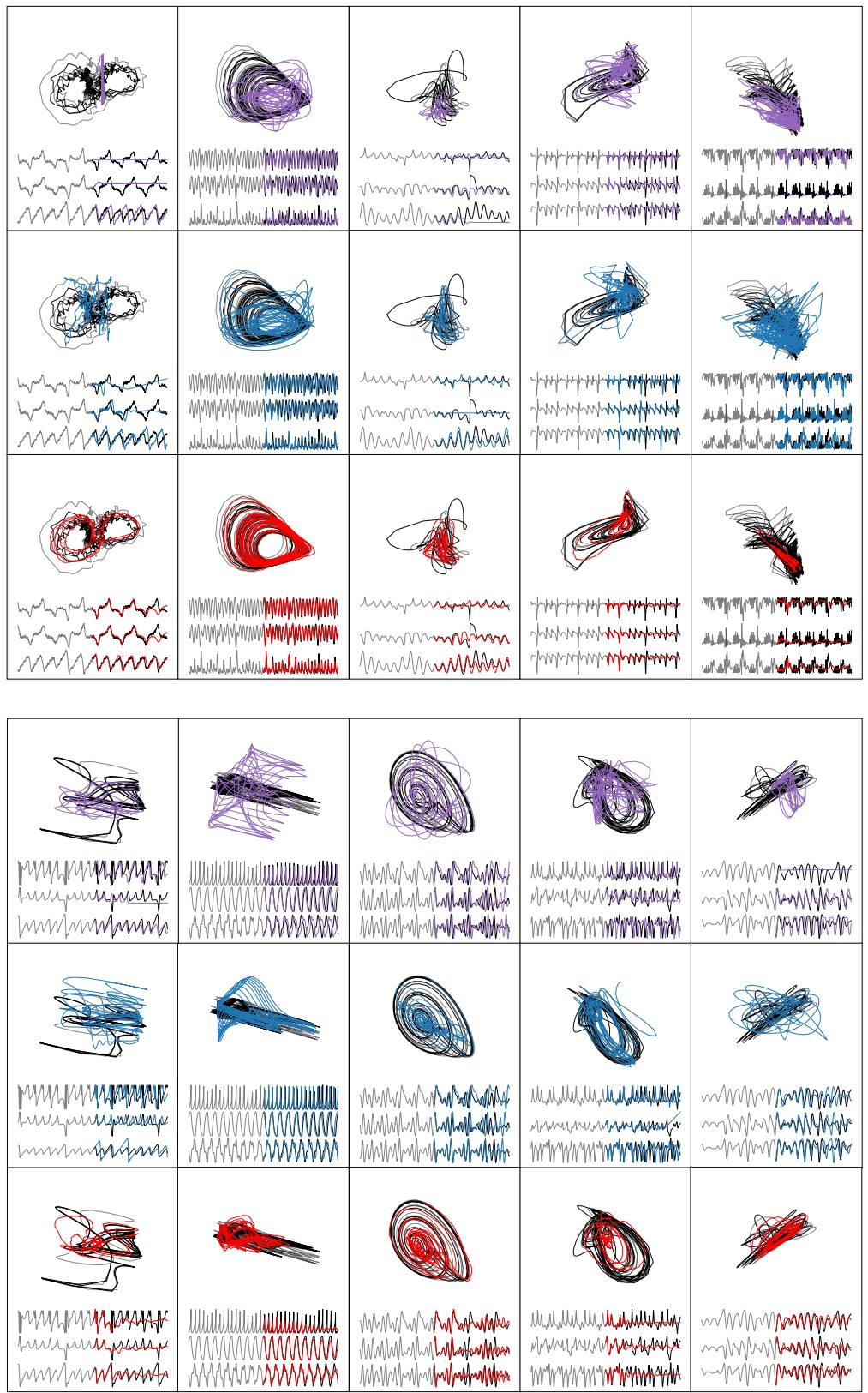

Figure 33: Long-term zero-shot forecasts ($L_{\text{pred}} = 512$) on held-out chaotic dynamical systems. Comparison between *Panda* (Red), *Chronos SFT* (Blue), and *Chronos* (Purple).

## N   ADDITIONAL COMPLETIONS

In Appendix E, we presented a sample of completions from our *Panda MLM* checkpoint on our held-out test set. Here, we provide more completions from *Panda MLM* (Fig. 34) and long-context completions from our scaled-up checkpoint *Panda MLM-66M* (Fig. 36). We also provide qualitative comparison between *Panda MLM* completions and piecewise cubic spline interpolation (Fig. 35) to further demonstrate the advantage of our method.

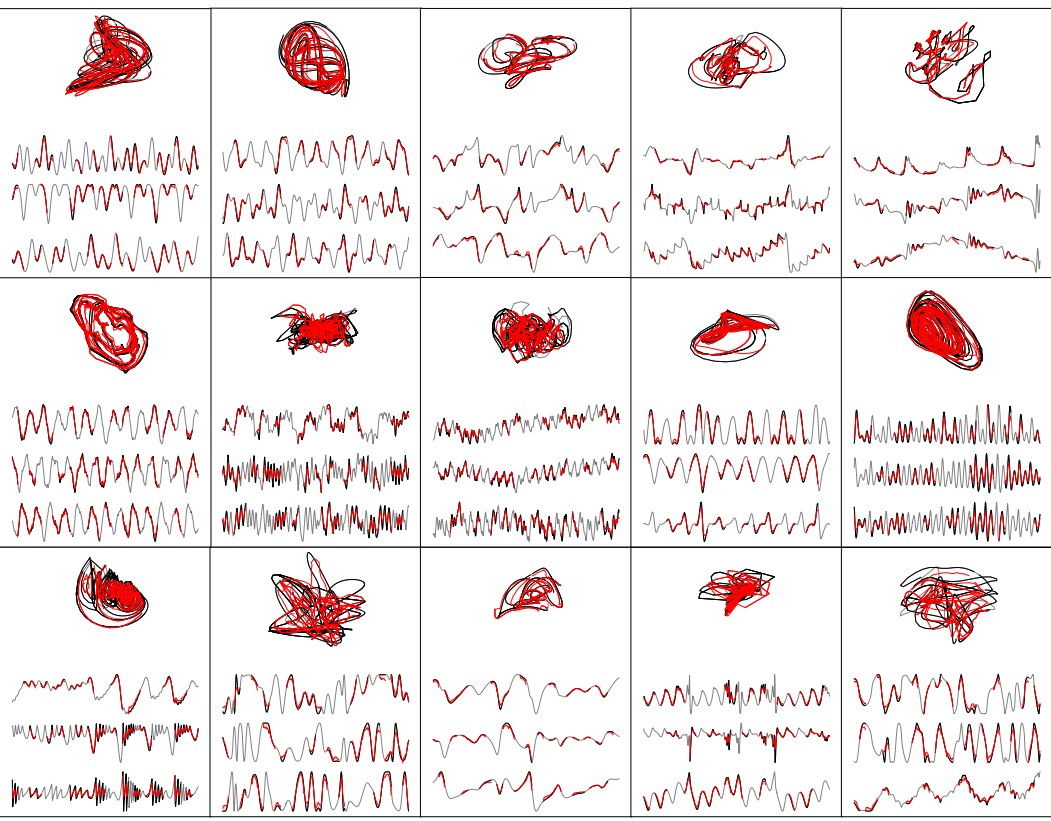

Figure 34: Examples of zero-shot completions on held-out chaotic dynamical systems. Each completion plotted was with a context length of 512 time points, with half the patches (patch length 16) randomly masked out in a channel-inconsistent manner. These plots show *Panda MLM*, our *20M* parameter checkpoint, completing the masked-out trajectories i.e. 256 time points.

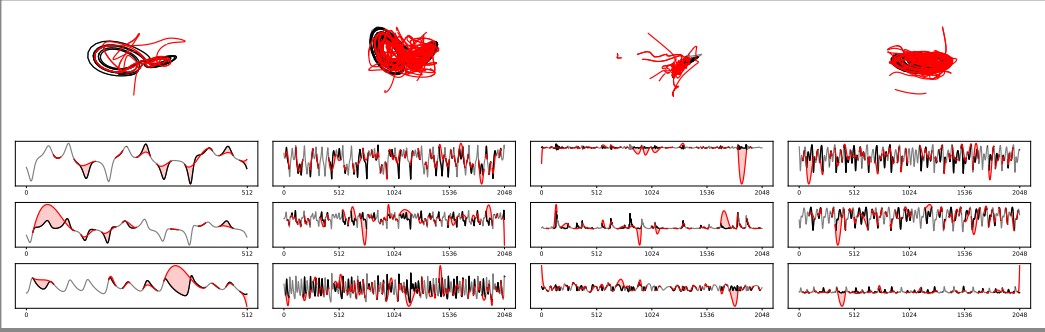

Figure 35: Qualitative comparison between completions generated by *Panda MLM* (with 20M parameters) and by piecewise cubic splines. First (leftmost) panel provides an example with context length 512 for clearer presentation; all other panels show context length 2048. Shaded red regions show the difference from the ground truth. Piecewise cubic spline interpolation is the most successful naive baseline, and although it achieves near competitive performance on preserving the correlation dimension (Table 10), it is not competitive with respect to pointwise error or attractor geometry.

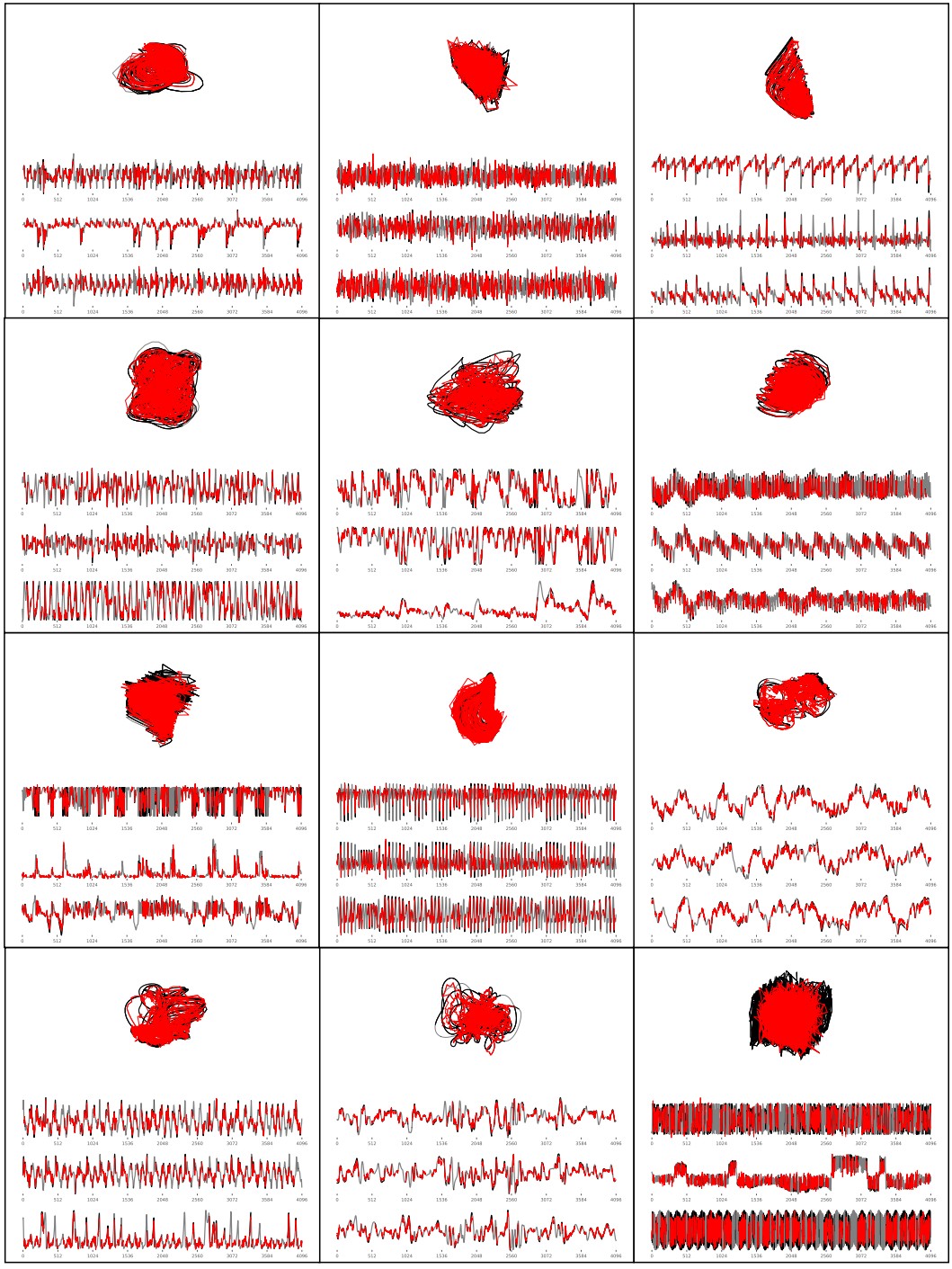

Figure 36: Examples of zero-shot completions on held-out chaotic dynamical systems. Each completion plotted was with a context length of 4096 time points, with half the patches (patch length 16) randomly masked out in a channel-inconsistent manner. These plots show *Panda MLM-66M*, completing the masked-out trajectories i.e. 2048 time points, despite only being trained on context length 512.

