# OpenReview forum: "Panda: A pretrained forecast model for chaotic dynamics"
_ICLR.cc/2026/Conference — ICLR 2026 Poster_

### Official Review · Reviewer_bbv2 · 2025-10-27

**Soundness:** 3
**Presentation:** 3
**Contribution:** 2
**Rating:** 6
**Confidence:** 4

**Summary:**

This paper introduces  PANDA (Patched Attention for Nonlinear DynAmics), a pretrained transformer model designed to forecast chaotic dynamical systems. The model provides the results of the model to forecast unseen chaotic systems and experiment data snippets in the zero-shot manner; the paper explores the instrinsic properties of Panda in Scaling law, long term statistical property, and some ability in PDE forecasting.

**Strengths:**

The manuscript shows the following strengths worth credits:

1. the idea of new chaotic trajectories dataset creation is novel;
2. the emprical visualization of the scaling law Panda Performance;
3. key indicators in long-term properties preserving and patch-attention maps are included

**Weaknesses:**

1. Buzz words cause overclaims, such as 'a universal model ...'. The model does not specify whether the model learns from one modality in the system and is able to forecast other modalities in the real-world systems and PDEs.
2. The emergent PDE performance is less solid, how is the long term statistical property evaluated?
3. Though the results presentation is extensive, the model layers, param and inference computing details are not found.

**Questions:**

1. As the authors claimed 'a framework to create the dataset of 2e4 ODEs' in the contribution 1, the framework would be a great contribution if the authors can clarify a) what algorithmic evolutionary method is applied; b) how to demonstrate the recombined thing is equivalent to physic solvers and simulators? c) What the phyics meanings of the recombined systems and how the community can access and benefit?
2. The emergent PDE performance is baseline-limited, how the performance is comparable specific surrogate models?
3. The emergent PDE performance is case-limited, e.g., how the performance is in classic benchmarks of 2D/3D Kolmogrov Flows?
4. Why Panda uses this set of attention design? What can be expected if other attention types (e.g Fourier Attention, fast attention, attention as gating filters) are applied?

---

> ### Author Response · Authors · 2025-11-26
> **Response to reviewer bvv2 (1/2)**
>
> We thank the referee for their review and constructive feedback. We have posted a revision and have aimed to address all of your points:
>
> **[Weakness 1 Claims]** Here, zero-shot forecasting is possible exactly because the model is able to learn modalities purely from low-dimensional simulated systems, and transfer at test to real-world data and PDEs. This is why we argue that **Figures 4 and 6** are so compelling---the model, at no point during training, saw those real time series or PDE simulations. We've revised the text to make this point clearer, and we have also removed the "universal" descriptions throughout the text.
>
> **[Weakness 2 PDE long-term]** We do not evaluate long term statistical properties of the PDEs as this would be out of scope for this work. However, we have added operator learning baselines tuned to the context windows we evaluate on to our PDE results for comparison. We would like to emphasize that we are not claiming *Panda* to be a suitable replacement for PDEs: the key finding here is that *Panda* can generalize at all to forecast high dimensional PDEs, despite only being trained on 3-dimensional subsets of dynamical systems.
>
> **[Weakness 3 Computing Details].** Regarding the training setup, those details are now in newly extended **Appendix B: Training Details**. We have also extended our inference time comparison in our new **Table 8** to include all baselines, showing that *Panda* has an order of magnitude faster inference than the next closest baseline. The GitHub and HuggingFace checkpoints are online, and we will post a non-anonymized link once the anonymity period ends. However, if the referee needs earlier access, please let us know and we will work on making an anonymized Huggingface. The model file is too large to upload directly to OpenReview. The Huggingface repository also contains json files with full training and model config parameters, and we also have logged all of our runs on wandb, which we can also make publicly available after the anonymity period ends.
>
> **[Q1a Evolutionary Algorithm].** Please see our extended **Appendix A: Generation of a Novel Chaotic Systems Dataset**, which describes each step of the evolutionary algorithm in detail. The stages are summarized at a high level in **Figure 1**, and **Appendix A** gives extensive detail about each step.
>
> **[Q1b Equivalence to Physics Solvers].** The generated dataset is exactly equivalent to a set of physical simulations. Each system is generated by integrating the right hand side of a set of analytic ODE. The individual terms in each ODE, like sin(x), etc, are what we discover via evolutionary search. Therefore, we can guarantee that the dataset matches the physics of known dynamical systems. To show this, we have added a new **Figure 11** in the **Appendix A.5: Dataset Properties** in which use 700+ standard time series features to embed each time series from one of the evolved ODE in the same space as the "parent" (human-discovered) ODE, and find overlapping regions. This means that the discovered ODE have consistent physical properties with the parent ODEs. We also measured several key properties characterizing dynamical systems: the Lyapunov times and the fractal dimension of the attractor (**Figures 12 and 13**). In both, we find that the discovered systems have a diversity of invariant properties, consistent with those of the upstream systems. We also have extended the description of the evolutionary algorithm in the supplementary material to clarify that it's the actual terms in the ODE that we are modifying, and so each system physically exists as a set of ODE.

---

> ### Author Response · Authors · 2025-11-26
> **Response to reviewer bvv2 (2/2)**
>
> **[Q1c Community Access].** The described dataset, along with an extended version with multiple initial conditions and different timescales per system, is publicly available to download on Huggingface. We cannot upload it here because it's a giant dataset, and there is not an easy way to anonymize HuggingFace, but the dataset is available and, if the referee asks, we will provide the AC a link to it to verify that this will be released with the paper.
>
> **[Q2 Surrogate Models].** We have now added two new baseline surrogate models in **Figure 6**: Fourier Neural Operators and DeepONet. Importantly, these are both fully-trained models: we train their weights on the same data that we give to *Panda* and *Chronos SFT* as *context only*. Surprisingly, both of the neural operators underperform the zero-shot models in this setting. We believe that this is because we in a low-data regime; short context isn't enough for the neural operators to converge, and so the pretraining of TSFM give them an advantage here.
>
> **[Q3 Case-Limited PDE performance].** These two systems, VKVS and KS, cover two major PDE used for benchmarking, particularly for weakly-chaotic (intermittent) dynamics. We don't expect a pretrained model to be a good choice for fully-turbulent 2D/3D Kolmogorov flow, which is challenging even for fully-trained neural operators (see Li, Liu-Schiaffini, et al.) [1]. We will note this in the paper, however, we are hesitant (and don't have enough time in the 2-week review period) to perform extensive additional benchmarks on PDEs. If this is critical for the referee please let us know and we will do what we can to meet this request after the review period.
>
> **[Q4 Other attention types].** *Panda* uses temporal and channel attention because it scales well and our focus was to demonstrate the efficacy of channel attention as a simple mechanism to improve forecast performance on multivariate dynamics. We tried several options before settling on the final architecture. We believe that trying other attention variants could improve forecasting results of different types of systems and produce different scaling laws. Although it is an interesting direction for future research, it is outside the current scope of our work.
>
> **References**
>
> [1] Li, Zongyi and Liu-Schiaffini, Miguel and Kovachki, Nikola and Liu, Burigede and Azizzadenesheli, Kamyar and Bhattacharya, Kaushik and Stuart, Andrew and Anandkumar, Anima, "Learning dissipative dynamics in chaotic systems", *Neurips 36*, 2022.

---

### Official Review · Reviewer_SotH · 2025-10-31

**Soundness:** 4
**Presentation:** 3
**Contribution:** 4
**Rating:** 8
**Confidence:** 4

**Summary:**

This paper introduces Panda, a pretrained transformer model for forecasting chaotic dynamical systems. The key contributions are: (1) an evolutionary algorithm that discovers ~20,000 novel chaotic ODE systems through mutation and recombination of 129 base systems; (2) a multivariate patch-based architecture incorporating dynamics-inspired features. in particular, channel attention; (3) demonstration of zero-shot forecasting capabilities on held-out chaotic systems, real-world experimental data, and emergent PDE forecasting; (4) a scaling law showing that diversity of dynamical systems, rather than data volume, drives generalization performance.

**Strengths:**

1. **Synthetic dataset generation**: The evolutionary discovery framework is interesting and well-executed. I think this is a valuable contribution.
2. **Scaling law for dynamics**: The finding that diversity of unique systems matters more than total data volume is very interesting scientifically.
3. **Architectural choices**: Channel attention for coupled systems and dynamics embeddings are good.
4. **Experimental results**: The evaluations are comprehensive and the results strong compared to baselines.
5. **Emergent PDE forecasting**: The zero-shot ability to forecast PDEs after training only on low-dimensional ODEs is impressive.
6. **Computational efficiency**: Table 8 shows significant speedup over Chronos, the nearest baseline.

**Weaknesses:**

1. **Clarity**: While the paper is generally very well presented, there are a few places where clarity could be improved. For example, the fact that training uses only 3 randomly sampled channels while evaluating on arbitrary dimensions is buried in Appendix B, but for me this an important detail. In addition, the paper doesn't explicitly state how the univariate baselines are applied to multivariate data—do they process channels independently and concatenate forecasts? How does Chronos-SFT handle multivariate data during fine-tuning?
2. **Mean regression problem**: Section 5.4 reveals that Panda regresses to the mean on long rollouts, yet the paper claims superior distributional metrics compared to Chronos, which "parrots" the input. How can mean regression preserve attractor statistics at all? I think this issues deserves closer study. The claims about attractor statistics would be strengthened if, for example, the model was able to reproduce Lyapunov spectra.
3. **Training details**: While Table 8 reports inference time showing favorable comparison to Chronos, the paper provides no analysis of training setup or timing.

**Questions:**

1. Please explicitly describe the inference procedure for univariate baselines on multivariate data. Do they process each channel independently and concatenate forecasts? How does Chronos-SFT handle multivariate data during fine-tuning?
2. Why does MLM pretraining help at training horizon but hurt rollout? Do you have some intuition here?
3. How can mean regression achieve better distributional metrics than "parroting"?
4. What are the training costs?
5. Have you tried computing Lyapunov spectra?

---

> ### Author Response · Authors · 2025-11-26
> **Response to reviewer SotH (1/2)**
>
> We thank the referee for their positive comments and support for our paper. We have aimed to address all of your points in our response:
>
> **[Weakness 1 Clarity].**
>
> We have modified the main text to clarify that univariate models are trained and evaluated separately on each channel. We have also extended Appendix B to provide more information about hyperparameter ranges, and attention architectures, and other details.
>
> **[Weakness 2 Mean regression].** We have performed comparisons with additional metrics, and we now have results using KL divergence, Hellinger distance, and Lyapunov exponents at a variety of longer horizons (see our revised **Appendix F**). Models like *Chronos* and many other TSFMs are univariate models, and parroting on a channel separately (without mixing information from the other channels/dimensions) is not at all guaranteed to preserve the geometry of the attractor or long-term statistical properties. To see this, we refer the reviewer to **Figures 30, 31, and 32** in **Appendix M**. We also thank the referee for the suggestion to compute the Lyapunov spectra; see our response to **Q5** below.
>
> **[Weakness 3 Training details].** We agree, and we thank the referee for this suggestion. Our newly-revised **Appendix B: Training Details** describes the attention layers more explicitly and all hyperparameters.  We added the training time costs in Appendix B. Training the *Panda* forecasting model required $\sim$104 GPU hours and finetuning the size-matched *Chronos* model required $\sim$192 GPU hours. Overall, Panda is much cheaper to train than existing time series foundation models since *Chronos* and *TimeMOE*, which require pretraining the base model on massive collections of real time series data. We also have publicly accessible (but non-anonymized) Huggingface repositories that host all of our model checkpoints; we will release these links after the anonymity period ends. These repos also contain json files with full training and model config parameters, and we have also logged all of our runs on wandb. We can provide those links as well after the anonymity period.
>
> **[Question 1 Inference].** We have added this information to the main text. Forecasting with univariate models is performed on each channel independently for training and inference - this is also how univariate time series foundation models are applied to multivariate time series in practice. We include timing experiments for inference in our newly revised Appendix (**Table 8**). Panda is by far the fastest in inference time by an order of magnitude.
>
> **[Question 2 MLM].** MLM pretraining helps the model learn the distribution of dynamics that are bounded to an attractor since the model can leverage surrounding context (even from the future) to infill missing patches. However for rollout, the model can only use a fixed history of *true* context and autoregressively compounds on its own error. We believe it is an interesting open research question to see how MLM can be better used to control the open-loop rollout error. We also believe an interesting future direction is to investigate how the timescale set by the patches (patch size 16) and random masking strategy could impact the performance on rollout. The current random masking results in a geometric distribution over the run lengths i.e. the probability of $k$ contiguous masked out patches is $(1-p)p^{k-1}$, where $p=0.5$ for our setting (half the patches are masked out randomly). Thus, the average run length is $1/(1-p)$ contiguous patches masked, corresponding to blocks of 2 patches in a row i.e. blocks of 32 timesteps in a row masked out in between ground truth points, on average, in our setting. There are also more sophisticated masking strategies to explore, in future work.
>
> **[Question 3 Regression Distributional Metrics].** We believe this depends on the metric, but crucially we want to emphasize that parroting on a channel without mixing across channels can easily result in predictions that go wildly off attractor (**Figures 30, 31, 32**). As seen from the example forecasts we provide throughout our appendix, in addition to the qualitative comparison against the baselines in **Appendix M**, *Panda* tends to stay on attractor, respecting the attractor geometry, despite mean regressing for some systems at much longer prediction horizons (compared with the prediction horizon 128 we used for training). We hope that these comparisons provide an intuitive view for the distributional metrics results, and we are happy to extend this section to include more examples as well, if the Referee deems it illuminating. We thank the referee for bringing up this point for us to clarify, as we also believe it ties into a broader discussion on the potential utility of multivariate TSFMs (leading TSFMs are almost all univariate as of the time of this work).

---

> ### Author Response · Authors · 2025-11-26
> **Response to reviewer SotH (2/2)**
>
> **[Question 4 Training Costs].** Thank you, we have added the training time costs in **Appendix B**. Training the *Panda* forecasting model requires $\sim$104 GPU hours, while finetuning (SFT) the size-matched *Chronos 20M* model required $\sim$192 GPU hours. Overall, Panda is much cheaper to train than existing time series foundation models since Chronos and TimeMOE.
>
> **[Question 5 Lyapunov Spectra].** See **Appendix F: More Distributional Metrics and Invariant Quantities** subsection **F.1: Leading Lyapunov Exponent** **Figure 20**, where we show the distributional comparison against the leading Lyapunov exponents estimated from the ground truth of the same prediction lengths. This shows that at long prediction lengths, *Panda* matches the leading Lyapunov exponent for the bulk of our test systems, up to some systems with very high leading Lyapunov exponent estimates of the ground truth, for which *DynaMix* does well. We emphasize that the data-driven estimation of Lyapunov exponents suffer from brittleness and sensitivity to hyperparameters and are thus not fully reliable for estimating invariant measures over finite horizons, compared to other metrics like KL divergence and Hellinger distance.

---

### Official Review · Reviewer_tpp7 · 2025-10-31

**Soundness:** 2
**Presentation:** 3
**Contribution:** 3
**Rating:** 2
**Confidence:** 4

**Summary:**

This work presents Panda, an attention-based foundation model (FM) dedicated to zero-shot forecasting chaotic dynamics. Using a novel synthetic dataset constructed from a library of known chaotic systems using principles from dynamical systems theory (DST), Panda is trained to forecast trajectories based on limited dynamical context. The model demonstrates robust generalization, successfully providing zero-shot forecasts for entirely novel chaotic systems, diverse real-world datasets, and even PDE problems. Panda is compared to other existing time series FMs in the field and demonstrates superior performance.

**Strengths:**

1. Panda shows great scaling behavior both in the amount of data as well as number of parameters.
2. Great to see DST motivated architectural design and the focus on setting a suiting inductive bias, which other FMs in the field lack.
3. The manuscript is well written and easy to follow.
4. Generating new chaotic systems using skew-product coupling and evolutionary search is a great idea and powerful method to generate vast amounts of valid data for this class of FMs. I think this greatly benefits research in the entire field.

**Weaknesses:**

My two main concerns are the following:
1. The paper misses relevant literature which introduces another DS FM model: DynaMix [1]. Both methods train on a synthetic dataset comprised of low-d chaotic DS, both methods have the aim of zero-shot forecasting and both question the efficacy of existing TS foundation models. I think it would be highly valuable to the SciML community if Panda is compared to such a similarly specialized model, which seems to perform much better than context parroting architectures such as Chronos.
2. “Panda exhibits emergent properties: zero-shot forecasting of unseen chaotic systems preserving both short-term accuracy and long-term statistics.” I find the claim that Panda preserves long-term statistics not sufficiently backed up by evidence. The main metrics employed to verify this statement are comparison of the fairly short prediction lengths (up to 512) w.r.t. the ground truth attractor (4096 time steps) in Section 5.4. On the same note, the authors identify regression-to-the-mean (or convergence to fixed point dynamics) as a failure mode of Panda. Proper evaluation of long-term dynamics would consist in performing *longer autoregressive roll-outs* in which the model, ideally, converges to the “climate” hinted by the context (see e.g. [1]). In this case, according to the argumentation of the authors that Chronos exhibits the more suiting inductive bias for climate forecasting (parroting), Chronos should actually perform better in both Hellinger distance as well as KL measure. The way these measures are currently used only strengthens the point that Panda is superior in preserving e.g. geometric features only in the *short-term*, also indicated by the consistently rising mean measure values with prediction length (see Table 1, 2, 10). A SciML motivated architecture should in the end aim to preserve the physics of the underlying system in the long-term, i.e. $T \rightarrow \infty$.

Other concerns:
1. Since the emergent zero-shot PDE inference is highlighted as a major contribution of the paper, the authors should include comparisons to baselines given by domain specific models trained on the context window (e.g. Neural Operator based architectures such as DeepONets [2] or FNOs [3])
2. (Appendix E / Figure 15) ground-truth + completions to ground truth using the correlation dimension lacks baselines: As a reader I can not quantify how well naive imputation methods would perform in estimating the correlation dimension.

Minor comments:
- Fig. 2 misses labels A, B, C mentioned in the caption
- Abbrv. “MLM” is first used in Section 5.1 but hasn’t been introduced in the main text before
- Double citation in References: “Jonas Mikhaeil, Zahra Monfared, and Daniel Durstewitz. On the difficulty of learning chaotic dynamics with rnns. Advances in neural information processing systems, 35:11297–11312, 2022a.”  and “Jonas M. Mikhaeil, Zahra Monfared, and Daniel Durstewitz. On the difficulty of learning chaotic dynamics with rnns. In Proceedings of the 36th International Conference on Neural Information Processing Systems, NIPS ’22, Red Hook, NY, USA, 2022b. Curran Associates Inc. ISBN 9781713871088.”

**References**:

[1] Hemmer, Christoph Jürgen, and Daniel Durstewitz. "True zero-shot inference of dynamical systems preserving long-term statistics." arXiv preprint arXiv:2505.13192 (2025).

[2] Lu, Lu, et al. "Learning nonlinear operators via DeepONet based on the universal approximation theorem of operators." Nature Machine Intelligence, vol. 3, no. 3, Mar. 2021. https://doi.org/10.1038/s42256-021-00302-5

[3] Liu-Schiaffini, Miguel, et al. "Neural Operators with Localized Integral and Differential Kernels." International Conference on Machine Learning. PMLR, 2024.

**Questions:**

1. What is the rationale for comparing context + ground truth to context + prediction for several experiments and measures throughout the paper? Why not simply ground truth vs. prediction? Doesn’t inclusion of the context heavily skew the outcome of empirical estimation of the Lyapunov exponent, for example? Especially in cases where $L_{pred} = 128$, where the estimated value is dominated by the context which is $4 \times L_{pred} =  512$?

---

> ### Author Response · Authors · 2025-11-12
> **Clarifying question at start of discussion period: DynaMix code was released 3 weeks ago**
>
> Thanks very much for your review. To start the discussion, we would like to clarify that **the code for DynaMix was not public at the time of the ICLR submission deadline.** The DynaMix code was only released 3 weeks ago (for the NeurIPS camera-ready).
>
> We are happy to include this new result in our benchmark and discussion (it will take some time to run this), but we would like to further clarify that DynaMix is *contemporaneous*, not prior work, to ours. We can’t provide direct evidence of this here due to the anonymity requirements, but we are willing to provide preprint dates or other information to the AC if the referee needs verification.
>
> We are now preparing new experiments in consideration of your other comments. However, we wanted to provide this information early in the discussion period, in case this was a deciding factor in your score.

---

> > ### Comment · Reviewer_tpp7 · 2025-11-17
> >
> > I thank the authors for the clarification, and I of course see the argument of contemporaneous work. I also wasn't aware that the code has only been online for a few weeks now. Nevertheless, I think including and discussing DynaMix in this work would be highly valuable for the community and I'm more than happy that the authors started experiments with DynaMix.
> >
> > This point is indeed a significant factor in my score, and I want to make clear that I'm more than happy to increase it to an accept if my concerns are addressed adequately.

---

> ### Author Response · Authors · 2025-11-25
> **Rebuttal to reviewer tpp7 (1/3)**
>
> We thank the referee for their review and constructive feedback. Our substantial revision includes all of the referee's suggested experiments. We hope the referee will find that their comments were substantively addressed:
>
> *EDIT*: we have updated the Figure numbers we refer to in the responses, to match our latest revision
>
> **[Weakness 1 DynaMix]**
>
> DynaMix is contemporaneous to our paper, and their code was released *after* the ICLR deadline. We have now added new benchmarks against DynaMix. The new benchmarks show that DynaMix does not beat Panda on our metrics. However, we understand that our study considers an evaluation setting slightly different than theirs, because we focus on short-term forecast accuracy in both training and evaluation, and we match our context/horizon to those used in time series foundation model (TSFM) studies.
>
> We now write, in our related works section, that the main differences between our work and DynaMix are:
>     1. Our new and much larger training dataset. DynaMix is pretrained on our founder pool of 135 systems published in an earlier study, rather than our enlarged dataset of $2 \times 10^4$ newly-discovered systems.
>     2. Our architecture uses channel and temporal attention, and allows for in-context transfer to different dimensionalities. Many general-purpose TSFMs like Chronos are univariate.
>     3. Our results, such as zero-shot PDE generalization, neural scaling laws in the number of dynamical systems, and resonances in the attention heads, all of which are consequences of our channel-attention approach.
>
> We now feature our metrics (sMAPE, MAE, MSE, and Spearman distance) evaluations for:
>     1. *DynaMix*, and
>     2. *Chronos 200M* instead of *Chronos 20M*,
> and we have updated **Figure 2** of the main text to replace *Chronos 20M* with *Chronos 200M*. Moreover, we have updated **Appendix D: Additional Forecast Metrics** to include in **Table 7** the statistical significance tests (Wilcoxon signed ranked test with Holm-Sidák adjusted p-values) of these new baselines for the aforementioned metrics. We have also updated 1) **Figure 15** to show the metrics for the new baselines on rollout, and 2) **Tables 8 and 9** to present the numerical values.
>
> And we also include the evaluation of *DynaMix* and *Chronos 200M* for the distributional metrics. Please see our response to the next point regarding long-term statistics and geometric structure. Note: in our new figures we use "DynaMix" instead of "DynaMix" but we will fix this in the next revision.

---

> ### Author Response · Authors · 2025-11-25
> **Rebuttal to reviewer tpp7 (2/3)**
>
> **[Weakness 2 Long-term statistics and geometric structure]**
>
> Please see our newly-extended **Appendix F: More Distributional Metrics and Invariant Quantities**, in which we compute the average spectral Hellinger distances, the KL divergences, and the leading Lyapunov exponents against *DynaMix* and our baselines, for predictions up to prediction horizon 3200, which is 25 times longer than the prediction length (128) over which *Panda* was trained. *Chronos* performs poorly at these very long horizons, because its parroting strategy eventually accrues error over such long horizons. *DynaMix* is competitive in this setting due to the fact that it is a purpose-built dynamical systems architecture which is trained autoregressively (like Chronos and unlike Panda) for a *very*  long prediction horizon of 10K time points or $\sim78\times$ longer than Panda's prediction horizon. Despite this, it performs worse in average spectral Hellinger distance, for which *Panda* beats all the baselines for all the prediction horizons we evaluated, from length 128 to 3200. We note that this is surprising since *Panda* is trained to minimize pointwise error over a short horizon non-autoregressively. *Panda* also outperforms the baselines on KL divergence *until* we get to very long prediction horizons, where *DynaMix* excels. In the next four paragraphs, we provide details on these results, and we take care to highlight them in our main text and Appendix sections.
>
>
> **KL divergence:** We have updated our **Table 1** to incorporate:
>     1. Longer prediction horizons: 1024 and 3200; which correspond to 8x and 25x the prediction length (128) we trained on; and
>     2. Evaluation results for *DynaMix* and *Chronos 200M*.
> For this main text table, we now present the KL divergence between the ground truth and model predictions for each prediction length. The main finding here is that at very long prediction lengths, *DynaMix* beats *Panda* in KL divergence. In **Appendix F: More Distributional Metrics and Invariant Quantities**, we present **Table 12** to show the aggregated per-system differences in KL divergence between *Panda* and the baselines, to provide a more fine-grained evaluation. In this table, we clearly mark the point at which *DynaMix* outperforms.
>
> **Hellinger Distance:** We have also updated our **Table 2** to incorporate:
>     1) Longer prediction lengths: 1024 and 3200; and
>     2) Evaluation results for *DynaMix* and *Chronos 200M*.
> We find that *Panda* outperforms all baselines on the average spectral Hellinger distance for prediction length 128 to 3200, although our advantage decreases with very long prediction lengths. In **Appendix F: More Distributional Metrics and Invariant Quantities**, we present **Table 13** to show the aggregated per-system differences in average spectral Hellinger distsance between *Panda* and the baselines. And in **Figure 19** we show the distribution of computed values corresponding to **Table 2** as an alternative view, for the longer prediction horizons. We have also improved the computation of the spectral Hellinger distance to make it more numerically stable, thus mitigating NaN values.
>
> **Lyapunov Exponents:** We now compute the leading Lyapunov exponents for much longer prediction horizons (1024, 2048, and 3200), for predictions from *Panda* and for *DynaMix*. We present these results and discussion in **Appendix F: More Distributional Metrics and Invariant Quantities** subsection **F.1: Leading Lyapunov Exponent** **Figure 20**, where we show the distributional comparison against the leading Lyapunov exponents estimated from the ground truth of the same prediction lengths. This shows that at long prediction lengths, *Panda* matches the leading Lyapunov exponent for the bulk of our test systems, up to some systems with very high leading Lyapunov exponent estimates of the ground truth, for which *DynaMix* does well. We emphasize that the data-driven estimation of Lyapunov exponents suffer from large errors and sensitivity to hyperparameters and are thus not fully reliable for estimating invariant measures over finite horizons, but we believe that these estimates still provide useful signal. We have also removed our previous two tables on the Lyapunov estimates from this Appendix section, as they were estimates for much shorter horizons (128, 256, 512), but we can add it back if the Referee thinks it necessary.

---

> ### Author Response · Authors · 2025-11-25
> **Rebuttal to reviewer tpp7 (3/3)**
>
> **[Other Concern 1 PDE ability].**
>
> We have added baselines for both FNO and DeepONets tuned on the evaluation context windows in Figure 6 to present **preliminary** results for the von Karmán and the Kuramoto Sivashinsky PDEs. Note that these comparisons are not entirely apples-to-apples since the foundation models see the full context while the neural operators only see the current state. We are still optimizing the baselines, as these are nontrivial PDEs and require a fair amount of tuning. We will prioritize getting these results as soon as we can and we will update the figure accordingly. Moreover, it is not possible to train the operator learning baselines with the same context window as the foundation models in this experiment setup. We do not think that a full benchmark against a large suite of fully-trained PDE models is within scope for our paper, and we do not argue that we think our zero-shot model is a substitute for Neural Operators when sufficient data is available. Details on the operator learning baselines tuned to the context window are provided in Appendix H.
>
> **[Other Concern 2 Ground Truth + Completions].**
>
> In **Appendix E: MLM Completions** we have added results to show that *Panda MLM* outperforms three interpolation baselines:
>     1. Piecewise cubic splines;
>     2. Linear interpolation; and
>     3. Polynomial (degree 3) interpolation.
> Please see our new **Table 11** in **Appendix E: MLM Completions**. We have re-run all the MLM evaluations for double the number (now 8) of random seeds, which determine the random timestep maskings per trajectory. We now use as context the entire length 4096 trajectories per system. We have also trained, released, and evaluated a scaled-up model, *Panda MLM-66M*, which achieves a significant improvement, and we have updated **Figure 18** for results on this model. We have also updated the captions of **Figures 17 and 33**, which show examples of completions, to correct an error in the context length for those visualizations: from 2048 (erroneous) to 512. In **Appendix N: Additional Completions** we present in **Figure 35** example completions with context length 4096 from *Panda MLM-66M*. Lastly, in **Appendix N: Additional Completions** we also show in **Figure 34** a qualitative comparison of the completions generated by *Panda MLM* versus those from piecewise cubic splines, which were the best-performing interpolation baseline. This figure highlights that there are more illuminating metrics to compute at a future date that will even further emphasize the advantage of our approach
>
> **[Minor Concerns]**
>
> Thank you for identifying these. We have removed the repeated reference and defined the acronym. **Figure 2** now includes labels for each sub-panel.
>
> **[Question 1 Context + Predictions].**
>
> We want to clarify that, in the main text, we are comparing *only* the predictions against the ground truth. Importantly, we are **not** appending the true context to our predictions, which would inflate the metrics. In the referenced SI Tables, we also computed the metrics with the context because, at a previous point, we were asked to perform this comparison. In no main text results do we mix true context with predictions. To avoid confusion, we have removed from **Appendix F: More Distributional Metrics and Invariant Quantities** all of our previous tables with the metrics computed on (context + predictions) vs. (context + ground truth), as these were vestigial from a previous request and we feel that these are no longer needed.
> We also make sure that every table with these distributional metrics clearly has the number of timesteps (prediction horizon, $L_{\text{pred}}$) of the comparisons marked in the table.

---

> > ### Comment · Reviewer_tpp7 · 2025-11-26
> >
> > **[Re: Weakness 1 & 2: DynaMix, long-term statistics and geometric structure]**
> >
> > I thank the authors for including DynaMix as a comparison model. I had a deeper look back at the DynaMix paper, and I’m very confused and puzzled by the substantial contradictions between the papers.
> >
> > 1) As per my original review, KL and Hellinger distance are supposed to be measures that evaluate the long-term behavior of the systems in the ergodic limit. Even 3200 time steps of forward prediction may be too short for that, but using short transients of as few as just 128 or 512 time steps is certainly not appropriate, and pretending that these assessments are on equal ground with the correct application of these measures I find highly misleading. I therefore think the table needs to be corrected and reported only with the numbers that best reflect this long-term limit. Even then, if the system didn’t reach the limit set yet, these numbers are to be interpreted with caution, and in my mind this should be clearly acknowledged in the paper.
> > 2) That essentially all models tested appear to perform better than DynaMix in both KL and Hellinger distance is even more puzzling, as the authors stated themselves that Chronos and Panda usually converge to fixed points (regression-to-the-mean) or cycles in the limit, very unlike the results for DynaMix reported in the other paper. Thus, Panda and Chronos should have flat or strongly peaked power spectra in the limit. How could these possibly have better agreement in power spectra therefore than a system that approximates the ground truth limiting behavior well? This is hard to understand, and I suspect it may be related to the limitations stated in 1) and the issue of obtaining any reasonable estimate of these quantities from less than a few thousand data points.
> > 3) Another point could be that DynaMix seems to benefit from more context (see Fig. 3c in [1]), as more context provides more information on the invariant measure of the underlying attractor. It would hence be great to see how the different models compare for increasing context lengths (e.g. up to 2048).
> >
> > 4) The manuscript currently states that DynaMix was trained on long autoregressive roll-outs up to 10k time steps, which is incorrect. While evaluation was performed on long autoregressive roll-outs, DynaMix was only trained using sequences of length 550 (see Appx. A.1 of [1]).
> >
> > I would appreciate if the authors could please show actual figures of a) longer rollouts of the systems and Panda predictions shown in Fig. 12 to qualitatively assess the long-term limit  b) the corresponding state spaces, and c) the actual power spectra. Additionally I would be interested in both metrics (KL and hellinger distance) for >10k time steps for both Panda and DynaMix. Moreover, the authors should add more details on the computation of the measures (KL and hellinger distance) including code (these measures are currently not included in the provided codebase) to rule out any possible errors. Otherwise I remain unconvinced.
> >
> > I stress that I’m not opposed to acceptance of this work but I would like to see a bit more fair evaluation of the models and presentation of the results. As it stands, this is just highly misleading and confusing in my opinion.
> >
> > **[Re: PDE ability]**
> >
> > I thank the authors for including custom-trained NOs as PDE baselines. I agree with the authors here, that this comparison is sufficient for the manuscript.
> >
> > **[Re: Concern 2 Ground-truth + Completions]**
> >
> > Thanks, this clearly shows that Panda indeed provides superior completions!
> >
> > **[Re: Question 1 Context + Prediction]**
> >
> > I see, thanks for the clarification!

---

> ### Author Response · Authors · 2025-11-26
> **Question for the referee: would the following new experiments be sufficient?**
>
> Thank you for considering our reply.
>
> First off, we apologize for the incorrect statement that *DynaMix* is trained for 10K timepoints - this was a misinterpretation of the training script in the DynaMix codebase. We have corrected this mistake in our revision.
>
> We are looking into the *DynaMix* issue. It's a much smaller model ($\sim$10k params) trained on a much smaller and less diverse dataset (34 systems instead of 20,000). Moreover, *DynaMix* can only effectively handle 3D dynamical systems - the codebase opts to handle each channel independently when forecasting systems >3D - and changing the *DynaMix* code to accommodate forecasting >3D systems differently is out of scope. Given its small size and pretraining set, we think it's very impressive that it's competitive at all with *Chronos* on our test set.
>
> It may take a few days for us to understand why it’s not performing well here, but these are our current hypotheses. However, we disagree with the referee's characterization of our current results as misleading. We implemented all of our metrics and chose our context lengths and prediction horizons for experiments well before the ICLR deadline and before *DynaMix* was even available. Additionally, the code for the metrics was not implemented here in this study, it is from the existing open source `dysts` library under the `metrics` submodule (this library is also where the *DynaMix* authors got their 34 pretraining systems).
>
> Our experiment settings were chosen to match the typical scales of standard foundation model papers that focus on short-term rollouts, while the *DynaMix* paper instead focuses on a very different setting than other TSFM work (thousands of points and many multiples of the context, longer than would be typical for *Chronos* et al.). In fact, the only reason we include long-term metrics is because we have previously been asked to include them, despite our paper’s focus being very different.
>
> Nevertheless, we agree that long term properties should be more informative at longer horizons, and that *DynaMix* may perform better in this setting. But we are hesitant to *post hoc* shift the focus of our experiments towards this regime due to *DynaMix* is not performing well on our test set. Please note that all systems in the train and test set ($\sim$10K systems) were only integrated for 4096 timepoints each which is why we rollout to 3200 timepoints for the long term metrics. It is not feasible for us to generate and quality check a new dataset within this rebuttal window to match *DynaMix*'s later choice to use 10,000 timepoints for their rollout evaluations.
>
> To this end, we’d prefer to run narrower supplementary experiments as well as edit the main text to clearly state that we understand that *DynaMix* is optimized for a different setting than the standard TSFMs we benchmark. We propose to sample a smaller subset of our test set systems and integrate them for at least 10550 timepoints (cutting off any transients, which is something we already do for all generated systems) and then rerun the long term metrics evaluation. **Would this targeted experiment be sufficient for the reviewer to reconsider their evaluation?**

---

> > ### Comment · Reviewer_tpp7 · 2025-11-27
> >
> > My point is that you are not using the long-term metrics correctly as *long-term* metrics that assess the behavior on the attractor, and therefore your assessment in the provided comparison table in my mind is strongly misleading. The table gives the wrong impression that you were assessing *long-term* behavior in the ergodic limit, while truly you were not, but mostly transients (hence not really different from the sMAPE values already reported in Fig. 1). KL is supposed to be used as a measure that tests agreement of the attractors, it becomes completely meaningless if you assess it on just short transients. Even purely statistically, computing power spectra and KL on just a few hundred data points is just not sensible in my opinion and experience. *To use these measures properly, ideally you would need really long roll-outs (say ~10k), from which you cut off the first ~2000 time points reflecting transients.* And just for clarity, with transients here I mean transients in your *model forecasts* (not in the true systems used as benchmarks). Besides this flaw in the evaluation, as I suggested in my previous comment, it might also have played a role that they used longer context signals in the DynaMix paper (e.g. 2000 context length for results in Fig. 4 of [1]).
> >
> >
> > This being said, I'm not asking to scale up DynaMix to >3D. I think the paper has already improved a lot, and I understand that Panda was conceived for a different (shorter-term) application scenario, so I'm actually not expecting Panda to outperform DynaMix (or other models for that matter) on every single measure. But I would like the authors to do a *fair* comparison that honestly acknowledges current limitations, pros and cons, and uses statistics that have been suggested in the literature in an appropriate manner. I thus maintain my opinion that the current tables are strongly misleading for the reasons discussed.
> >
> >
> > However, from what the authors described, I understand that it will be difficult to obtain full comparisons on 10k time points for the time being. I'd suggest some middle ground: Evaluating KL and Hellinger distance on the *last 2000 time points only* of the longest rollouts the authors have.
> >
> >
> > Regarding Fig. 12, although the authors have integrated their benchmark systems only for 4096 time points, it should still be possible to produce longer roll-outs for Panda to get an impression of how it performs in the long-term limit?

---

> > > ### Author Response · Authors · 2025-12-03
> > > **Final Response to Reviewer tpp7**
> > >
> > > We thank the referee for clarifying their requests. We have now addressed all of these concerns with new experiments and substantial re-writing of the long-term metrics section.
> > >
> > > 1. In the new main text **Section 5.6 Limitation: Regression to the mean** we transparently describe the limitations of Panda in producing long-term forecasts, and its tendency to mean-regress over long horizons instead of context parroting. In recognition of the referee's comments, we have also revised all other discussions of "long-term" metrics to instead describe "distributional metrics," because times series foundation models (TSFM) like Chronos traditionally target a regime of forecasting of hundreds of timepoints, not tens of thousands. We have modified the abstract and introduction as well to reflect this.
> > >
> > > 2. We recomputed the KL divergence metric using the same implementation of the KL divergence (called geometric misalignment) in the DynaMix repo (instead of the Gaussian mixture in the dysts repo, which uses a fuzzy simplex method to fit the width of each Gaussian locally). See the newly-added **[F.3 An Alternative KL Divergence Implementation (Geometric Misalignment)]** and **Figure 19**. DynaMix fares slightly worse in geometric misalignment at longer prediction horizons of $3584$.
> > >
> > > 3. We now calculate all metrics at a range of different prediction horizons, including the longest available $L_\text{pred} = 3584$ given our length 4096 trajectories and 512 context length. See newly-added **[Appendix F: More Distributional Metrics and Invariant Quantities]**, particularly **[Tables 12 -- 16]** and **[Figures 19 --21]**,.
> > >
> > > 4. See newly-added **[Table 16]**, which shows distributional metrics on the last 2048 forecast points (cutting off the first 1536 forecast points) compared to the full forecast, illustrating the mean regression phenomenon raised by the referee and now described in the main text.
> > >
> > > 5. We provide state space plots of long-term rollouts of Panda on select systems, to provide a qualitative view of the model's long-term behavior in **[Appendix M, Figures 31, 32, and 33]**.

---

### Official Review · Reviewer_t2gr · 2025-11-01

**Soundness:** 4
**Presentation:** 4
**Contribution:** 4
**Rating:** 10
**Confidence:** 4

**Summary:**

This paper introduces Panda, which is a large foundation model for chaotic time series data. To generate the training dataset, the authors introduce a novel, genetic algorithm-like method for discovering new chaotic systems. The model architecture is an extension of the PatchTST model, utilizing patching along the time dimension, feature augmentation with random polynomial and Fourier features, and the use of both temporal attention and channel attention. Comparing the trained model to previous time series foundation models of similar size, the authors find that Pandas gives more accurate predictions, both in the short-term exact trajectory prediction and in the long term predictions of the statistical properties of the attractor. Furthermore, Pandas was able to perform zero-shot forecasts for both experimental data (which contains imperfections not presented in the training dataset) and for PDE systems (which were not seen by the model since it is only trained on ODEs). The paper contains lots of additional results, such as the scaling law with respect to the diversity of the training dataset, and analyses of the attention maps learned by the model.

**Strengths:**

1.	The proposed model shows excellent performance in both short term trajectory predictions and long term statistical property predictions. It is carefully designed to predict chaotic systems in particular and contain lots of architectural considerations that will be useful to researchers working on similar problems.

2. The automatized dataset generation process is also novel, and the suite of criteria used to automatically sift for chaotic systems seems quite useful. The final dataset of 20000 new chaotic ODEs is also, to my knowledge, first of its kind. Such large scale dataset will also help the further development of foundational models for chaos.

3. Lots of experiments were performed to demonstrate the model's performance. The out-of-distribution results for experimental data and the PDE is quite impressive too, and suggests that indeed the model is able to properly forecast chaos, instead of just having memorized motifs in the training dataset.

**Weaknesses:**

1.	This is a very well written paper, and there seem to be no significant weaknesses.

**Questions:**

1.	I am curious about the additive skew-product coupling used in the recombination step for the data generation process. From equations 1 and 2, it seems that recombination will lead to a ODE of larger dimensionality than its parent, since given ODEs for x and y, recombination gives a ODE for the concatenated state [x, y]. If so, I would imagine that the dimensions of the ODEs in the dataset will grow over iterations due to the repeated recombination process. Is this indeed the case?

2. The authors mention that the integration horizon and the granularity (which I assume indicates the sampling period of the trajectories) is standardized using the dominant timescale. How was this timescale determined? Are the authors looking at the dominant frequency in the frequency spectrum, or using the inverse of the maximum Lyapunov exponent?

3. Looking the generated chaotic systems in A.1 and A.2, it seems that loosely there are families of related systems (single, double, triple scrolls, etc). I wonder if it would be possible to do a low dimensional embedding of the generated dataset (or if it is too expensive, some subset of it) to create a 2D plot that shows how different systems in the dataset are positioned (in the chaotic behavior space) with respect to one another, as done in Figure 1 of [1].

4. How long does it take to train the Panda model? How does this compare to the competitors considered such as Chronos and Time MOE?

5. On a similar note, how long does the data generation process take?

[1] W. Gilpin. Chaos as an interpretable benchmark for forecasting and data-driven modelling. NeurIPS (2021).

---

> ### Author Response · Authors · 2025-11-26
> **response to reviewer t2gr**
>
> We very much thank the referee for their review and strong support of our paper.
>
> **[Q1 Coupling].** We agree, this is a good point---our training setting has a bias towards dynamical dimensionality slightly higher than 3, because all of the systems are descendants of named systems (and 3 is the minimum number of dynamical dimensions to support chaos in continuous time). For our dataset, we are only using the first offspring generation, and so we don't see dimensionality drift (a sort of "outcross" effect in the genetic algorithm). However, we agree that this effect should appear if we were to run the procedure for additional generations, and we are actually currently studying the "heredity" of various chaotic properties. We do see that the increased effective dimensionality of later generations has some effect on invariant properties (see **Figures 12 and 13** and **Table 4** in the Appendix of the current manuscript), but chaoticity itself does not appear to be suppressed, and so we don't think it's occurring in the first generation we consider here.
>
> **[Q2 Timescale].** The timescale is determined from a procedure that extracts the average period from the dominant frequencies of the power spectral density across each channel. In practice, we find this timescale to produce trajectories that make a sufficient amount of "round trips" around the attractor. It is also worth noting that this timescale is often larger than the Lyapunov time (**Figure 12**). We favor this timescale over the Lyapunov time for technical reasons: prior works find that it is more stable as granularity or data volume vary, and it can be estimated from time series without knowing the underlying dynamical system.
>
> **[Q3 Embedding].** We agree, thank you for this suggestion. Per the referee's suggestion, we have added a **Figure 11** in **Appendix A.5: Dataset Properties**, that shows a low-dimensional embedding of the $2 \times 10^4$ skew-product systems in the pretraining dataset, with the "founding generation" of 135 parents overlaid (Note that we actually use 129 of these parent systems as parents, since we cut out the delay systems, and we'll make a note of that in the next revision). The two distributions overlap substantially, precluding strong founder effects or mode collapse in this first generation.
>
> **[Q4 Training time]** We added the training time costs in **Appendix B**. Training the *Panda* forecasting model required $\sim$104 GPU hours and finetuning the size-matched *Chronos* model required $\sim$192 GPU hours. Overall, Panda is much cheaper to train than existing time series foundation models since *Chronos* and *TimeMOE*, which require pretraining the base model on massive collections of real time series data.
>
> **[Q5 Data Generation].** The data generation process can run on CPU, and each recombination and selection sweep can be parallelized. Generating the initial set of 20k systems took about two days on 64 EPYC 7V13 CPUs. We have since improved the efficiency of our attractor selection computation, and we are able to create much bigger datasets, which are planning to make public after the anonymity period ends. We have also logged many of our data runs on wandb, which we can also release to show full hardware system  metrics like CPU utilization, memory usage, disk utilization, etc.

---

### Meta-Review · Area_Chair_SFHM · 2026-01-06

**Summary:**

The reviewers generally indicated that:
1. The paper is well written and contains clear exposition.
2. The new dataset is generated in a novel way and may be useful to other researchers in the field.
3. The methods generalizes well to unseen dynamical systems, at least, in relatively short-term predictions.
4. The presented scaling laws demonstrate well the utility of more data coming from related problems.
5. The work misses a comparison to DynaMix and pre-trained surrogates in the PDE setting.
6. The work does not do an adequate job of showing that long-term statistics of the dynamics are captured.

**Reviewer Concerns:**

1. The authors have adequately addressed issues regarding baseline comparisons. In particular, they have shown results for DynaMix (a contemporaneous work) and added two PDE surrogates, DeepONet and FNO.
2. The authors have added statistical metrics for the long-term dynamics and increased the time intervals on which they are computed. This does not still fully resolve the issue as capturing properties of the attractor requires very long roll-outs and is not reconciled with their observation that the method eventually reverts to the mean.

**Reviewer Scores:**

I believe reviewer tpp7 could have increased their score given that the authors provided a substantial amount of new results to resolve their concerns. All other reviews were quite positive towards the work, and the authors did a good job of addressing their concerns.

---

### Decision · Program_Chairs · 2026-01-26

Accept (Poster)